# Transcriptional repression facilitates RNA:DNA hybrid accumulation at DNA double-strand breaks

Florian Saur[1,3], Emma Lesage [1,3], Lea Pradel[1], Sarah Collins [1], Anne-Laure Finoux[1], Emile Alghoul [1], Benjamin Le Bozec[1], Vincent Rocher[1], Romane Carette[1], Nadine Puget[1], Marie Couralet[1], Melanie Petiot[1], Thomas Clouaire [1,2] ✉, Aline Marnef [1] ✉ & Gaëlle Legube [1] ✉

RNA:DNA hybrids accumulate at DNA double-strand breaks (DSBs) and were shown to regulate homologous recombination repair. The mechanism responsible for the formation of these non-canonical RNA:DNA structures remains unclear although they were proposed to arise consequently to RNA polymerase II or III loading followed by DSB-induced de novo transcription at the break site. Here, we found no evidence of RNA polymerase recruitment at DSBs. Rather, strand-specific R-loop mapping revealed that RNA:DNA hybrids are mainly generated at DSBs occurring in transcribing loci, from the hybridization of pre-existing RNA to the 3′ overhang left by DNA end resection. We further identified the H3K4me3 reader spindlin 1 and the transcriptional regulator PAF1 as factors promoting RNA:DNA hybrid accumulation at DSBs, through their role in mediating transcriptional repression in *cis* to DSBs. Altogether, we provide evidence that RNA:DNA hybrids accumulate at DSBs occurring in transcribing loci as a result of DSB-induced transcriptional shut down.

DNA DSBs are toxic lesions that are highly detrimental for the maintenance of genomic integrity, holding the potential to generate mutations, copy number alterations and translocations. DSBs are generally repaired by two groups of pathways: homologous recombination (HR) and non-homologous end joining (C-NHEJ and Alt-NHEJ). While C-NHEJ directly ligates the broken ends, HR starts by a process known as resection, which generates single-stranded DNA necessary for the invasion of a homologous intact DNA copy of the damaged locus that is used as a template for DNA synthesis[1].

In the past decade, transcription machineries and RNAs have emerged as key contributors during DSB repair (reviewed previously[2–4]). Damage-induced noncoding RNAs (ncRNAs) have been identified in many experimental conditions, mostly using microscopy-based approaches. RNA molecules generated at the vicinity of the broken

ends have the ability to hybridize to their complementary DNA strands leading to the formation of RNA:DNA hybrids (two-stranded structures) or R-loops (three-stranded structures with a displaced DNA strand). Indeed, several reports subsequently identified the presence of RNA:DNA hybrids or R-loops at sites of damage (also called break-induced RNA-DNA hybrids (BIRDHs), reviewed in refs. 3,4). The use of dedicated tools to probe R-loops and RNA:DNA hybrids (for example, RNase H hybrid binding domain fused to a fluorescent protein or the S9.6 antibody directed against RNA:DNA hybrids) clearly established that RNA:DNA hybrids (or R-loops) accumulate at DSBs induced by laser micro-irradiation and nucleases such as HO, I-SceI, AsiSI or I-PpoI, in various organisms ranging from yeast to human[4]. Such damage-induced ncRNAs and RNA:DNA hybrids were proposed to function in DNA damage signalling and in HR, by regulating

[1]MCD, Centre de Biologie Intégrative (CBI), CNRS, Université de Toulouse UT, Toulouse, France. [2]Institut National de la Santé et de la Recherche Médicale (INSERM), Paris, France. [3]These authors contributed equally: Florian Saur, Emma Lesage. ✉e-mail: thomas.clouaire@univ-tlse3.fr; aline.marnef@univ-tlse3.fr; gaelle.legube@univ-tlse3.fr

resection[5,6], the stability of the resected end[7], the formation of the RAD51 nucleofilament[8] and the strand invasion step[9]. Additionally, proteins involved in alt-NHEJ (PARP1, Lig3 and XRCC1) were also found to interact with RNA:DNA hybrids[10,11] indicating that DSB-induced RNA:DNA hybrids also likely contribute to other DSB repair pathways. Furthermore, increasing evidence support a potential function of RNA as a template for DSB repair[4]. Altogether, this gave rise to the concept of RNA-dependent DNA repair[2].

Despite the large number of studies highlighting the crosstalk between transcription and DSB repair in recent years, the precise nature and biogenesis of damage-induced ncRNAs and subsequent RNA:DNA hybrids is not fully understood. Previous work reported a direct recruitment of RNA polymerase II (RNAPII) itself and the preinitiation complex (PIC) at sites of damage[7,12–15], while a latter study suggested that RNA polymerase III (RNAPIII), and not RNAPII, is targeted to DSBs[16]. Irrespective of the identity of the RNA polymerase, these reports raised the possibility that damage-induced ncRNAs, responsible for RNA:DNA hybrid formation in *cis* to DSBs, arise from bidirectional de novo transcription mediated through a non-canonical recruitment of transcription machineries at DNA ends. This has been further supported by in vitro studies, showing that the purified MRN complex, a key sensor for DSB, considerably fosters RNAPII loading and transcription arising from DNA ends[17], and by microscopy-based, in vivo studies showing nascent RNA at the vicinity of DSBs[18,19].

Yet, genome-wide, sequencing-based studies globally failed to identify a general accumulation of ncRNAs and RNA:DNA hybrids at all types of DSBs. Rather, ncRNAs and RNA:DNA hybrids were mainly observed at breaks occurring within or at the immediate vicinity of a transcribed unit, or/and repeated sequences, known to be heavily transcribed (for example, ribosomal DNA), and barely detectable at DSBs induced in intergenic and non-transcribed loci[8,20,21]. This raised the question as to whether RNA-dependent DNA repair may not be a general feature of DSB repair but rather correspond to a specialized pathway dedicated to handle transcription-coupled DSBs (TC-DSBs for DSBs occurring in actively transcribed loci). Moreover, several studies established that DSB-induced RNA:DNA hybrids need to be further removed to allow for faithful repair processes[7,8,22–27], raising the question as to whether DSB-induced RNA:DNA hybrids are bona fide repair intermediates or impediments that shall be eliminated[4,22].

On another side, it is also widely accepted that upon DSB formation, the transcriptional activity of nearby genes is shut down (reviewed previously[28,29]). Intriguingly, in-*cis* transcriptional repression is tightly linked to the completion of HR, though the precise mechanism remains unclear[29]. DSB-induced transcriptional repression depends on the proteasome[30], ATM, DNA-PK and DYRK1B kinases activity and also relies on multiple chromatin modifiers such as the NuRD (nucleosome remodelling and deacetylase) complex, the SWI–SNF chromatin remodelling complex, the histone H3 lysine 4 demethylase KDM5A or the Polycomb repressive complex for instance[28,29]. Such in-*cis* transcriptional repression has been difficult to reconcile with DSB-induced de novo transcriptional activity. Of interest, low RNAPII elongation rate, pausing and premature termination are well established as able to induce R-loop (for instance, ref. 31) raising the interesting possibility that DSB-induced R-loops/RNA:DNA hybrids may in fact rather represent by-product of transcriptional repression at TC-DSBs (reviewed previously[4,29]).

Here, we undertook a careful examination of RNAPII and RNAPIII recruitment, as well as of the distribution and strand specificity of RNA:DNA hybrids on a genome-wide scale following the induction of multiple DSBs in both active and intergenic loci. We provide evidence that RNA:DNA hybrids arise at sites of breaks from pre-existing RNA that hybridize with the single-stranded DNA (ssDNA) overhang left by end resection. The formation of these hybrids depends on the transcriptional repression established in *cis* to DSB following ATM activation, as well as spindlin 1 (SPIN1), a reader of H3K4me3, and the

transcription regulator PAF1 (polymerase-associated factor 1) that we identify as important for both transcriptional shut down and RNA:DNA hybrid accumulation. We propose a model whereby DSB-induced RNA:DNA hybrids only accumulate at TC-DSBs (DSBs that occur in transcribing RNAPII-enriched loci) as a consequence of the DSB-induced ATM/SPIN1/PAF1-dependent transcriptional shut down combined with DNA end resection. These RNA:DNA hybrids further foster the initiation of resection and thus promote RPA and RAD51 assembly at TC-DSBs. However, these repression and resection-induced by-products are globally detrimental and RNA:DNA hybrids need to be further eliminated to ensure cell survival and genome stability.

## Results

### No evidence of RNAPII and RNAPIII loading at DSBs

To gain a global insight into RNA polymerases dynamics at DSBs, we took advantage of the DIvA cell system allowing for the induction of ~80 DSBs across the genome[32], a large fraction of which are located within promoters and genes bodies, including exons and introns (Extended Data Fig. 1a)[33]. For this study, DSBs were further categorized as being either induced in a transcribed locus ('TC-DSBs', $n = 65$) or in an intergenic or silent gene ('silent DSBs' $n = 15$; Supplementary Table 1 and Extended Data Fig. 1a,b). Notably, TC-DSBs and silent DSBs are cleaved to an equivalent level (Extended Data Fig. 1b). TC-DSBs located within genes bodies can fall in exons and introns, and most (88%) are located within 1 kb around the transcription start site (TSS) (Extended Data Fig. 1a).

To establish whether de novo recruitment of RNAPII takes place at DSBs, we performed chromatin immunoprecipitation sequencing (ChIP-seq) against total RNAPII in DIvA cells using two different antibodies. To compensate for a potential global transcriptional repression upon damage as described previously[34], RNAPII ChIP-seq data were scaled to account for potential systematic differences across all genes between conditions, allowing us to focus on the specific changes occurring at DSBs upon damage (Methods). We first validated our RNAPII ChIP-seq datasets on all genes (Extended Data Fig. 1c) and confirmed our capacity to visualize RNAPII recruitment following DSB induction on a DNA damage-responsive gene (*CDKN1A* encoding p21) (Extended Data Fig. 1d). However, RNAPII recruitment was not observed at DSBs after damage induction (Fig. 1a–c). This absence of RNAPII accrual holds true for both TC-DSBs and silent DSBs (Fig. 1b,c and Extended Data Fig. 1e) and was observed irrespectively of the initial amount of RNAPII before DSB induction (Extended Data Fig. 1f). Furthermore, TC-DSBs induced in exons or introns of genes seemed equally unable to recruit RNAPII (Fig. 1b and Extended Data Fig. 1g). Rather, DSB induction triggered a reduction of RNAPII occupancy at DSBs, in agreement with the transcriptional repression observed in *cis* to DSBs (for instance on DSB 526; Fig. 1b and Extended Data Fig. 1h). To rule out that ChIP-seq may not be sensitive enough to detect low level of RNAPII accumulation at DSB, we also analysed ChIP experiments using quantitative PCR. As expected, before DSB induction, RNAPII was enriched on transcribed loci (*RBMXL1*, *KDELR*3, *TRIM37* and *CDKN1A*) but not at an intergenic locus (Ctrl2) or on a silent gene (*LUZP2*) (Extended Data Fig. 2a). Yet, RNAPII accrual was neither detected at TC-DSBs (DSB 526, 657 and 379) nor at a DSB induced in a silent locus (DSB 72) (Extended Data Fig. 2a). In contrast, and in agreement with DSB-induced RNAPII eviction[30,35,36], RNAPII levels were decreased at DSBs induced in *RBMXL1* (DSB 526), *KDELR*3 (DSB 657) and *TRIM37* (DSB 379) (Extended Data Fig. 2a). To assess whether these results could be recapitulated with another DSB induction method, we used the CRIPSR-Cas9 system to target the active *RBMXL1* gene promoter (sgRNA1), first intron (sgRNA2) or an intergenic locus (sgRNA3) (Fig. 1d). We did not detect RNAPII recruitment by ChIP–qPCR at any of these Cas9-induced DSBs for which efficient cleavage was confirmed by ChIP–qPCR against γH2AX (Fig. 1e, right panels). We also set to analyse RNAPII level at DSBs induced by the topoisomerase II inhibitor etoposide that we identified using END-seq (Extended Data

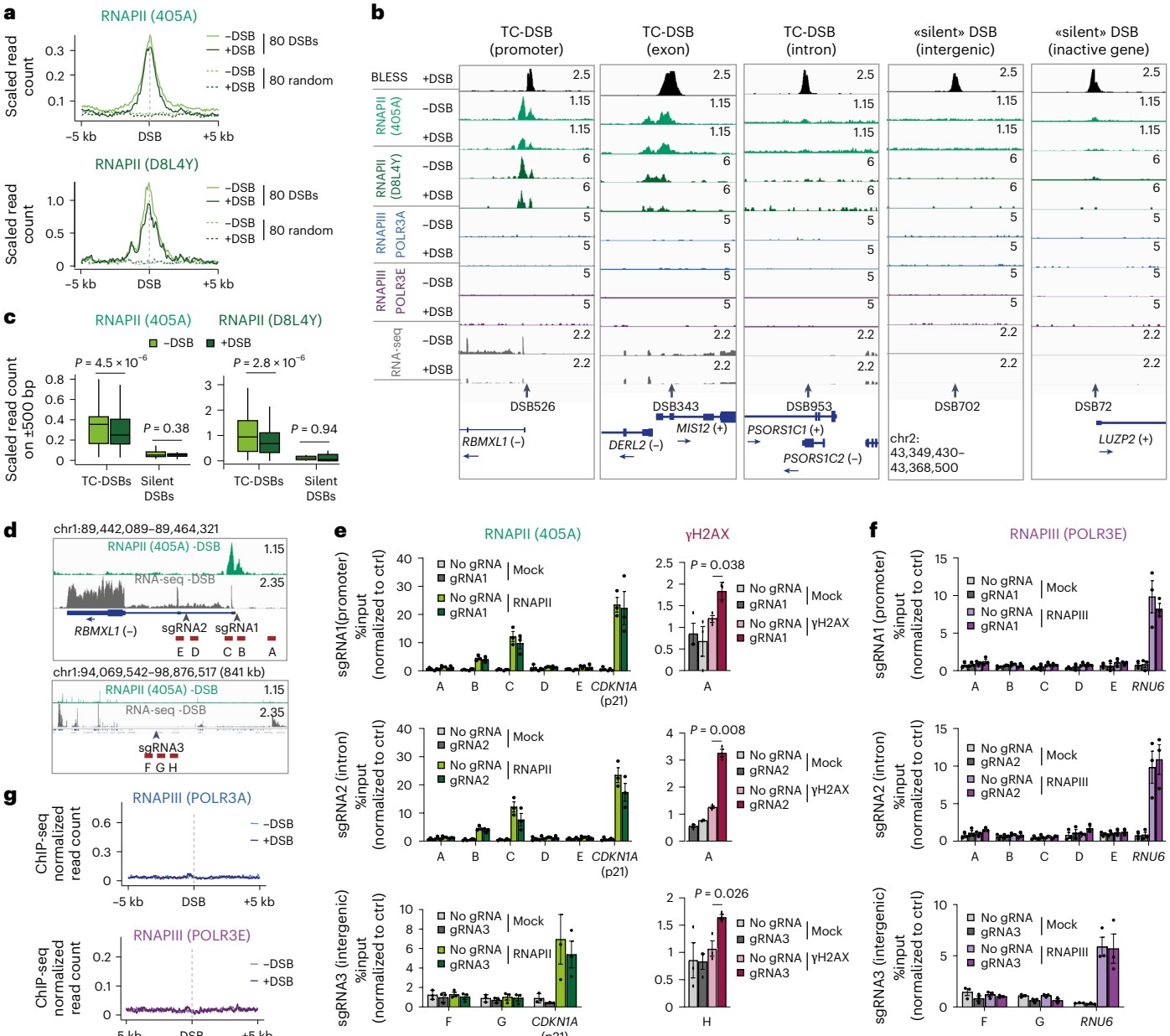

**Fig. 1 | RNA polymerases II and III are not recruited at breaks post-DSB induction. a**, Average profiles of RNAPII ChIP-seq signals using two different antibodies (405A and D8L4Y) before (−DSB) or after DSB induction (+DSB) around 80 best-cleaved DSBs and around 80 random sites. **b**, Genomic tracks (hg19) of BLESS, RNAPII, RNAPIII (POLR3A and POLR3E) ChIP-seq and RNA-seq around five DSBs located either in the promoter, exon, or intron of transcriptionally active genes, within an intergenic region or within a transcriptionally silent locus in the absence (−DSB) or presence of DSB (+DSB) induction. **c**, Quantification of RNAPII ChIP-seq signal on TC-DSBs ($n = 65$) and silent DSBs ($n = 15$) on a ±500-bp window around DSBs. Centre line shows the median; box limits show first and third quartiles; whiskers show maximum and minimum without outliers; points indicate outliers. $P$ values were calculated using paired two-sided nonparametric Wilcoxon tests. **d**, Genomic tracks (hg19) of RNAPII ChIP-seq (405A) and RNA-seq on the *RBMXL1* gene showing the position of sgRNA1 (promoter) and sgRNA2 (intron) (top) and on an intergenic

locus (bottom) with the position of sgRNA3. The position of primers for ChIP–qPCR in **e** and **g** is represented with red bars (A–E for sgRNA1 and sgRNA2, F–H for sgRNA3). **e**, RNAPII (405A) and γH2AX ChIP–qPCR before (no gRNA) or after DSBs induction with either sgRNA1, sgRNA2 or sgRNA3 normalized by Ctrl2. Mean and s.e.m. of three independent biological experiments are shown. $P$ values calculated from paired two-sided $t$-tests comparing no sgRNA with sgRNA conditions are only indicated if significant ($P < 0.05$). **f**, POLR3E ChIP–qPCR before (no gRNA) or after CRISPR-generated DSBs with either sgRNA1, sgRNA2 or sgRNA3 normalized by Ctrl2. Mean and s.e.m. of $n = 3$ independent biological experiments are shown. $P$ values from paired two-sided $t$-tests comparing no sgRNA with sgRNA conditions are only indicated if significant ($P < 0.05$). **g**, Average profiles of ChIP-seq signals for two different subunits of RNAPIII (POLR3A and POLR3E) before (−DSB) and after DSB induction (+DSB) around 80 best-cleaved DSBs in DIvA cells. Source numerical data are available in Source data.

Fig. 2b). ChIP–qPCR against RNAPII at different etoposide-induced DSBs in genes or intergenic loci confirmed the lack of RNAPII accumulation (Extended Data Fig. 2c). Given the previously reported RNAPIII recruitment at DSB[16], we also performed ChIP against RNAPIII. RNAPIII recruitment was not detected at Cas9-induced DSBs (Fig. 1f), nor at

etoposide-induced DSBs (Extended Data Fig. 2c). In DIvA cells, ChIP-seq against two subunits of the RNAPIII machinery showed a strong enrichment of both POLR3A (the catalytic subunit, also named RPC1) and POLR3E (peripheric subunit) at annotated loci encoding for transfer RNAs (Extended Data Fig. 2d,e). However, none of these subunits of

RNAPIII displayed a recruitment at DSBs (Fig. 1g and Extended Data Fig. 2f; the latter includes three annotated AsiSI sequences previously shown to accumulate RNAPIII using ChIP–qPCR[16]). Thus, using high-resolution ChIP-seq and highly sensitive ChIP–qPCR combined with three independent DSB induction methods, we find no evidence of de novo targeting of RNAPII and RNAPIII to DSBs induced in human cells.

## No evidence of de novo transcription at DSBs

To further deepen our analysis of RNAPII behaviour at DSBs, we profiled the different phosphorylated forms of the C-terminal repeat domain (CTD) of RNAPII (serine 2, 5 and 7, and tyrosine 1 phosphorylation) by ChIP-seq. Indeed, the transcription cycle comes along with notable changes in the RNAPII CTD phosphorylation status, from initiating RNAPII, phosphorylated on S5/7 and enriched at 5′ ends, to the elongating form of RNAPII, phosphorylated on S2 and accumulating on genes bodies and 3′ ends. Given that DSB induction was shown to alter RNAPII modifications genome-wide[34], we scaled our ChIP-seq data (Methods and above) to focus on the specific changes occurring at DSBs upon damage. ChIP-seq profiles for RNAPII CTD phosphorylation displayed the expected patterns across all human genes (Extended Data Fig. 3a) and phosphorylated RNAPII also accumulated at the DNA damage-responsive gene *CDKN1A* (p21) upon DSB induction (Extended Data Fig. 3b), validating our ChIP-seq datasets. As observed with total RNAPII, none of these phosphorylated forms of RNAPII showed accrual at TC-DSBs or DSBs induced in silent loci (see examples in Fig. 2a–c and Extended Data Fig. 3c). We next measured de novo transcription using transient transcriptome sequencing (TT$_{chem}$-seq), a powerful method for studying nascent transcription, especially that of short-lived RNA species[37]. Nascent transcription was readily observed when averaging the signal across all genes (Extended Data Fig. 3d). TT$_{chem}$-seq also efficiently detected nascent transcription at the lowly expressed *DICER1-AS* locus (Extended Data Fig. 3e) and the transcriptional increase at *CDKN1A* upon DSB induction (Extended Data Fig. 3e), further validating our dataset. Yet, TT$_{chem}$-seq did not show increased nascent transcription initiated from DSBs induced in silent or active loci (Fig. 2d–f and Extended Data Fig. 3f). Rather, and consistent with RNA sequencing (RNA-seq) and RNAPII ChIP-seq data (Extended Data Fig. 1h), DSB induction triggered a decrease of nascent RNA levels at DSBs sites (Fig. 2d–f and Extended Data Fig. 3f). This lack of DSB-induced increase in the TT$_{chem}$-seq signal holds true regardless of the position of the TC-DSB (in promoters, introns or exons) (Extended Data Fig. 3g). Furthermore, TT$_{chem}$-seq profiles indicated a reduction of transcription elongation across damaged genes compared to a control gene set (Fig. 2g,h), which is in agreement with a DSB-induced transcriptional repression. This elongation defect was accompanied by a reduction of SPT5, a subunit of the DSIF complex, which regulates RNAPII promoter-proximal pause (PPP) release and elongation[38,39] (Fig. 2i,j and Extended Data Fig. 3h,i). Overall, our data show no evidence that DSB induction triggers de novo transcription initiated at DSBs via the recruitment of either RNAPII or RNAPIII.

## RNA:DNA hybrids form on the single-stranded DNA after resection

The data above failed to detect RNAPII or RNAPIII recruitment as well as de novo transcription at DSBs. Nevertheless, RNA:DNA hybrids were repeatedly identified at sites of damage[4]. Thus, to further investigate the mechanisms that trigger the formation of RNA:DNA hybrids at DSBs, we established their strand specificity by performing quantitative differential DNA-RNA immunoprecipitation sequencing (qDRIP-seq)[40] (Extended Data Fig. 4a), before and after DSB induction in DIvA cells. In undamaged cells, strand-specific qDRIP-seq was consistent with previously published qDRIP-seq in HeLa cells[40] with R-loops generated during transcription accumulating on the Crick strand for genes in the forward orientation (+) (Extended Data Fig. 4b). We also confirmed our capacity to detect increased qDRIP-signal post-DSB induction on the *GADD45A*

gene, which is subjected to a strong transcriptional activation after damage in DIvA cells (Extended Data Fig. 4c). As previously reported using less-resolutive classical DRIP-seq[8], the qDRIP-seq signal was increased post-DSB induction when averaged over the 80 best-induced DSBs (Fig. 3a and Extended Data Fig. 4d). Of note, RNA:DNA hybrids were not detected at all DSBs, which is not explained by strong differences in cleavage efficiency (Extended Data Fig. 4e). Moreover, qDRIP-seq and DRIP–qPCR indicate that DSB-induced hybrids accumulate at levels similar to the R-loops detected on *RPL13A* and *RPS24* (Extended Data Fig. 4f,g), for which SMRF-seq (nondenaturing bisulfite R-loop footprinting followed by high-throughput sequencing) revealed that they form with a 5–10% frequency[41]. Considering that DSB 526 is cleaved in at least 10–20% of cells as detected by droplet digital PCR (ddPCR) assay (Extended Data Fig. 4h), this would indicate that RNA:DNA hybrids form at more than 50% of the damaged copies for this site. Importantly, DSB-induced RNA:DNA hybrids display two striking features: (1) a strong strand specificity, and (2) an asymmetry with respect to the DSB (Fig. 3b).

As end resection is a strand-specific process, we first assessed whether the strand specificity of our qDRIP-seq signals could be linked with resection. For this, we separated the qDRIP signals from the 5′-terminated DNA strands (which can undergo 5′ to 3′ resection, represented in brown on Fig. 3c) and 3′-terminated DNA strands (which corresponds to the 3′ overhang, represented in blue on Fig. 3c). For convenience, we summed the signals for each strand upstream and downstream of DSBs. Our data indicate that RNA:DNA hybrids are formed mainly on the non-resected strand rather than on the resected strand (Fig. 3c,d) (the DNA moiety of RNA:DNA hybrids is preferentially the 3′ terminated single-stranded DNA overhang left after 5′–3′ end resection). This result suggests that resection is a major determinant of RNA:DNA hybrid accumulation. In such a case, impairing resection initiation should negatively impact DSB-induced RNA:DNA hybrids. Indeed, we found that CtIP depletion (Extended Data Fig. 4i) or inhibition of MRE11 endo- or exonuclease activity by PFM01 and mirin respectively decreased RNA:DNA hybrids at DSBs (Fig. 3e). Altogether, these data show that RNA:DNA hybrids form through the hybridization of an RNA to the ssDNA 3′ overhang generated by end resection.

To further investigate the relationship between RNA:DNA hybrid formation and resection, we performed strand-specific replication protein A (RPA) ChIP-seq, END-seq and qDRIP-seq at both 4 and 24 h after DSB induction. As expected from previous studies, END-seq detected bidirectional resection 4 h after DSB induction[42] and END-seq signals strongly expanded at 24 h after DSB induction in agreement with extended resection at late time points[43] (Fig. 3f,g and Extended Data Fig. 5a). Similarly, RPA ChIP-seq also revealed extended, bidirectional and strand-specific binding of RPA at 24 h compared to 4 h after DSB, in a manner that correlated well with END-seq profiles (Fig. 3f,g and Extended Data Fig. 5b), which is in agreement with RPA being recruited on ssDNA following resection. By contrast, RNA:DNA hybrid distribution only moderately resembled RPA and END-seq profiles at 4 h and 24 h (Fig. 3f), although we could detect mild spreading of RNA:DNA hybrid accumulation at 24 h (Fig. 3f,g). This indicates that, while RPA binding strictly follows resection, in agreement with its capacity to bind ssDNA, RNA:DNA hybrids do not accumulate over the entire length of ssDNA. Moreover, the orientation of the RNA:DNA hybrid has little impact on extended bidirectional resection tracts (or long-range resection) as measured by END-seq and RPA binding (Fig. 3f,g and Extended Data Fig. 5c).

Notably, however, individual DSB inspection revealed that RPA profiles resembled qDRIP signals at the immediate vicinity of the broken end (Fig. 3g,h). This was more obvious when focusing on the 20 DSBs with the strongest enrichment in RNA:DNA hybrids for which DSB-proximal RPA accumulation (<500 bp) strongly overlapped with RNA:DNA hybrid levels (Fig. 3i; note that DSBs were oriented so that accumulation of RNA:DNA hybrids appears on the right side of the plot), suggesting that resection initiation events at the immediate vicinity of the break (or short-range resection) can be influenced by the formation of RNA:DNA

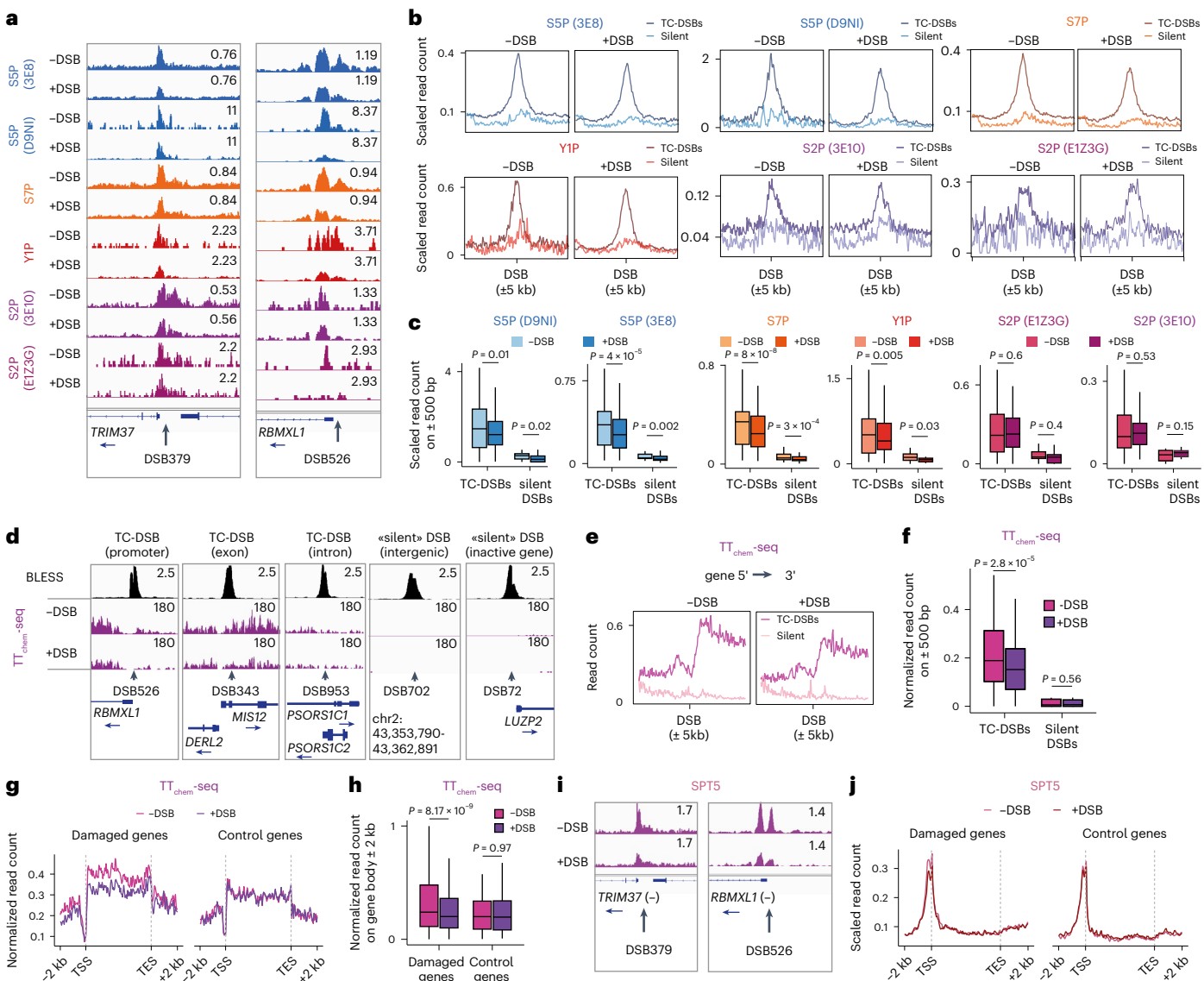

**Fig. 2 | No evidence of de novo transcription at DSBs. a**, Genomic tracks (hg19) of ChIP-seq against RNAPII CTD phosphorylation on serine 5 (S5P) (3E8 or D9NI antibodies), serine 7 (S7P), tyrosine 1 (Y1P) and serine 2 (S2P) (3E10 or E1Z3G antibodies) before and after DSB induction on two TC-DSBs (DSB 379 and DSB 526). **b**, Average profile of ChIP-seq against RNAPII phosphorylated forms at silent and TC-DSBs. **c**, Quantification of ChIP-seq signal of RNAPII CTD phosphorylations at silent ($n=15$) and TC-DSBs ($n=65$) on a ±500 bp window around DSBs. Centre line shows the median; box limits show first and third quartiles; whiskers show maximum and minimum without outliers; points indicate outliers. *P* values were calculated using paired two-sided nonparametric Wilcoxon tests. **d**, Genomic tracks (hg19) of BLESS and TT$_{chem}$-seq signal at silent and TC-DSBs before (−DSB) and after DSB induction. **e**, Average profile of TT$_{chem}$-seq signal at silent and TC-DSBs. The signal was oriented according to gene directionality. **f**, Quantification of TT$_{chem}$-seq on a ±500-bp window around silent ($n=15$) and TC-DSBs ($n=65$). Centre line shows median; box limits show first and third quartiles; whiskers show maximum and minimum without outliers; points indicate outliers. *P* values were calculated using paired two-sided nonparametric Wilcoxon tests. **g**, Average profile of TT$_{chem}$-seq signal on damaged and control genes in −DSB and +DSB conditions. **h**, Quantification of TT$_{chem}$-seq signal at damaged ($n=76$) and control ($n=73$) genes. Centre line shows median; box limits show first and third quartiles; whiskers show maximum and minimum without outliers; points indicate outliers. *P* values were calculated using paired two-sided nonparametric Wilcoxon tests. **i**, Genomic tracks (hg19) of ChIP-seq against SPT5 in −DSB and +DSB conditions at two TC-DSBs as in **a**. **j**, Average profile of SPT5 ChIP-seq signal on damaged and control genes in −DSB and +DSB conditions.

hybrids. To further evaluate the impact of RNA:DNA hybrids on end resection, we performed END-seq upon increased RNA:DNA hybrid levels thanks to the depletion of senataxin (SETX), one of the main RNA:DNA helicase involved in resolving RNA:DNA hybrids at DSBs[8,43] (Extended Data Fig. 5e). Notably, SETX depletion did not trigger an increase in the length of resection tracts (Fig. 3j, see arrows), indicating that RNA:DNA hybrids do not favour resection processivity nor speed. However, SETX depletion triggered an overall increase in resection signals (more cells in the cell population undergo DSB resection) (Fig. 3j), suggesting that RNA:DNA hybrids may potentiate resection initiation.

Altogether, our data indicate that (1) it is not R-loops that are detected at DSBs but rather RNA:DNA hybrids that are composed of an RNA hybridized to the ssDNA overhang generated by end resection, and (2) RNA:DNA hybrids likely potentiate short-range resection.

### DSB-induced RNA:DNA hybrids result from the hybridization of pre-existing RNA to ssDNA

So far, our observations revealed that while resection is clearly a bidirectional process (see symmetrical spreading of END-seq and

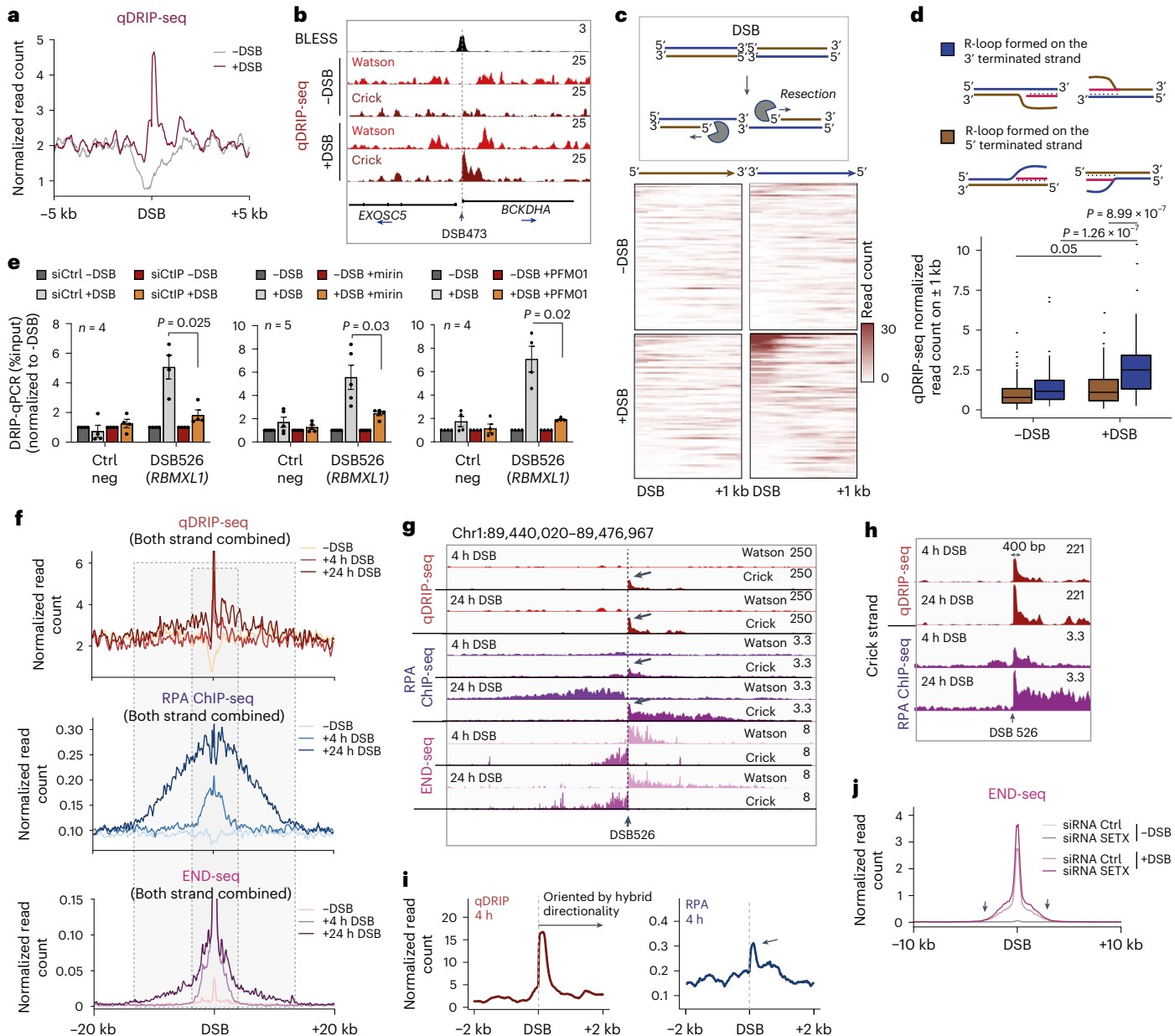

**Fig. 3 | Strand-specific accumulation of RNA:DNA hybrids on single-stranded DNA resulting from DNA end resection. a**, Average qDRIP-seq profiles on ±5 kb around the 80 best-cleaved DSBs. **b**, Genomic tracks (hg19) of BLESS and strand-specific qDRIP-seq at a TC-DSB (DSB 473). **c**, Top: schematic representation of DNA end resection at DSBs: resection triggers the elimination of the 5′-terminated strand (brown) leaving the 3′-terminated strand (blue) intact. Heatmaps of qDRIP-seq signal on 5′-terminated strands (brown, left, both sides of the DSB combined) and on 3′-terminated strand (blue, right, both sides of the DSB combined) for the 80 best-cleaved DSBs (bottom). **d**, Quantification of qDRIP-seq signal at DSBs (*n* = 80), represented in **c** for the 5′ (brown) or 3′-terminated strands (blue). Centre line shows median; box limits show first and third quartiles; whiskers show maximum and minimum without outliers; points indicate outliers. *P* values, paired two-sided nonparametric Wilcoxon tests. **e**, RNA:DNA hybrid levels measured by DRIP–qPCR upon CtIP depletion or MRE11 inhibition (endonuclease activity, PFM01 or exonuclease activity, Mirin), before

and after DSB induction at a negative control (Ctrl Neg) region and at a TC-DSB (DSB 526). Mean and s.e.m. of independent biological experiments (*n* ≥ 4) are shown (normalized to −DSB condition). *P* values, paired two-sided *t*-test. **f**, Average profiles of qDRIP-seq, RPA ChIP-seq and END-seq after 4 h or 24 h of DSB induction. The dotted grey boxes highlight the extend of resection at 4 h and 24 h post-DSB induction. **g**, Genomic tracks (hg19) of the strand-specific qDRIP-seq, RPA ChIP-seq and END-seq at a TC-DSB (DSB 526). Arrows show the local increase of qDRIP-seq and RPA ChIP-seq signals adjacent to the DSB. **h**, Zoom of the Crick strand of **g** for the qDRIP-seq and RPA ChIP-seq tracks (hg19). **i**, Average profile of qDRIP-seq and RPA ChIP-seq oriented by RNA:DNA hybrid directionality on a ±2-kb window around the 20 best hybrid-accumulating DSBs. The arrow points at the local increase in RPA and RNA:DNA hybrids. **j**, Average END-seq profile in siRNA Ctrl- (siCtrl) or siRNA SETX- (siSETX) transfected cells on ±10 kb around the 80 best-cleaved DSBs. Arrows indicate where DNA end resection reaches its furthest point. Source numerical data are available in Source data.

RPA signals; Fig. 3f,g and Extended Data Fig. 5a,b), RNA:DNA hybrid accrual is asymmetrical (Fig. 3b,c,g–i). We hypothesized that RNA:DNA hybrid asymmetry could be caused by pre-existing RNA hybridizing back to their template strand at TC-DSBs. In agreement with this, DSB-induced RNA:DNA hybrids were mostly detected at TC-DSBs

(Fig. 4a and Extended Data Fig. 6a), irrespective of the DSB position (in promoters, introns or exons) (Fig. 4a and Extended Data Fig. 6b). Moreover, we showed that the accumulation of hybrids is linked to the directionality of the damaged gene (Fig. 4a–c). Altogether, this suggests that the RNA moiety engaged in the RNA:DNA hybrid that

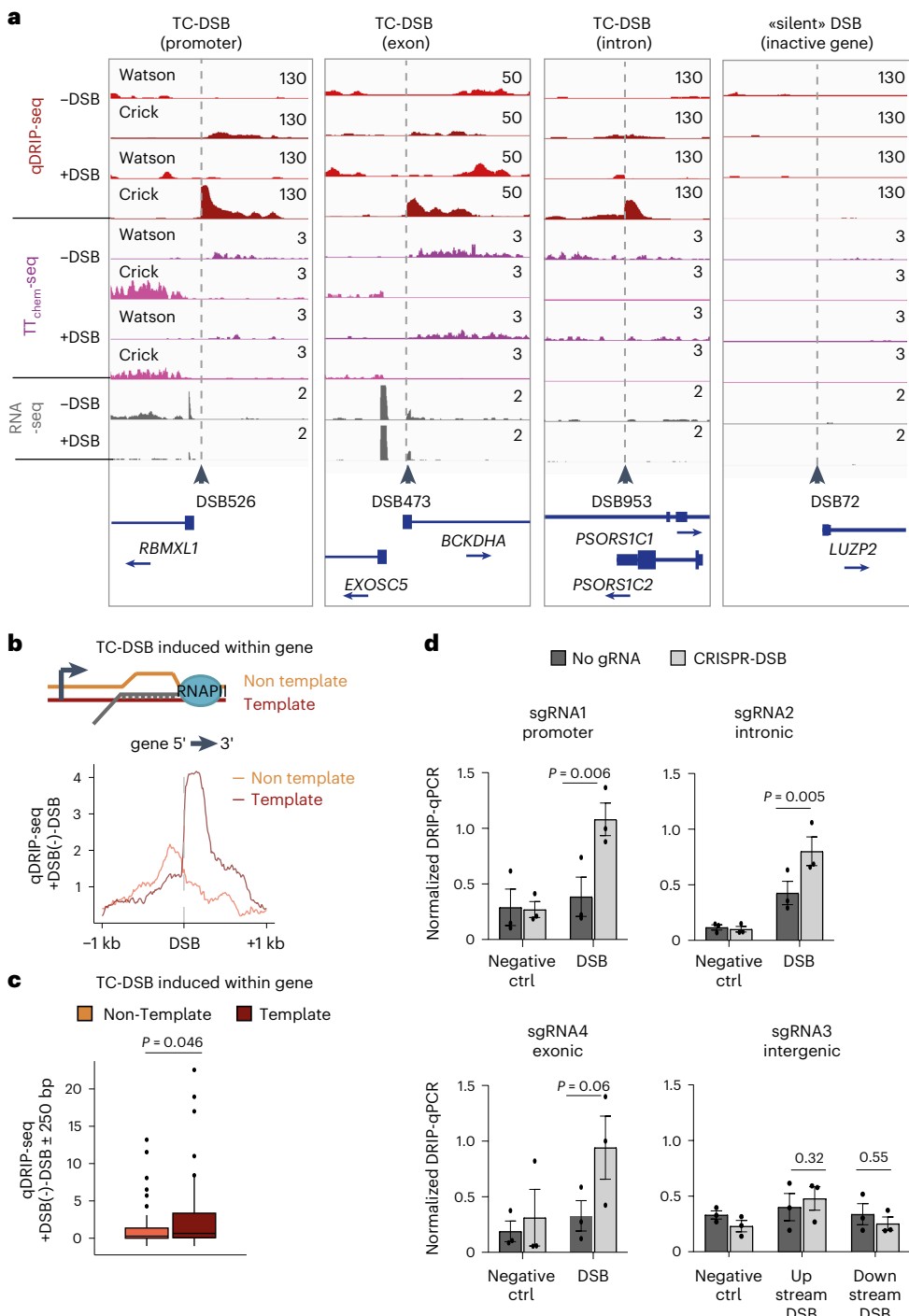

**Fig. 4 | RNA:DNA hybrids result from the hybridization of pre-existing RNA on the template strand. a,** Genomic tracks (hg19) showing strand-specific qDRIP-seq and TT_chem-seq, on the Watson and Crick strands and RNA-seq in −DSB and +DSB conditions at TC-DSBs located in a promoter, exon or intron and in a silent DSB located in a transcriptionally inactive gene. **b,** Schematic representation and average profile of qDRIP-seq signal on the template (red) or nontemplate (orange) strands of TC-DSBs located within genes. The signal was oriented according to gene directionality. **c,** Quantification of the qDRIP-seq signal on a ±250-bp window around TC-DSBs located within genes (n = 45). Centre line shows the median; box limits show first and third quartiles; whiskers show maximum and minimum without outliers; points show outliers. P values, paired two-sided nonparametric Wilcoxon tests. **d,** DRIP–qPCR detection of RNA:DNA hybrids around CRISPR-induced DSBs induced at different positions of the *RBMXL1* gene (sgRNA1, promoter, sgRNA2, intron, sgRNA4, exon) or in an intergenic region (sgRNA3) normalized by the *FOS* positive control. Mean and s.e.m. of n = 3 biological replicate are shown. P values, paired two-sided t-test. Source numerical data are available in Source data.

forms in *cis* to DSB mostly corresponds to the pre-mRNA exiting from the RNAPII and hybridizing with its template strand. Of interest, we observed several cases of TC-DSBs induced in promoters for which the RNA:DNA hybrids correspond to the PROMoter uPstream Transcript (PROMPT) (Fig. 4a) suggesting that antisense transcripts at

promoters can also form hybrids following DSB. To validate these findings, we used Cas9-induced DSBs using single guide RNA (sgRNA) targeting the promoter, an intron and an exon of *RBMXL1* (respectively sgRNA1, sgRNA2 and sgRNA4), as well as an intergenic locus (sgRNA3) (Extended Data Fig. 1d and Extended Data Fig. 6c). All Cas9-DSBs were

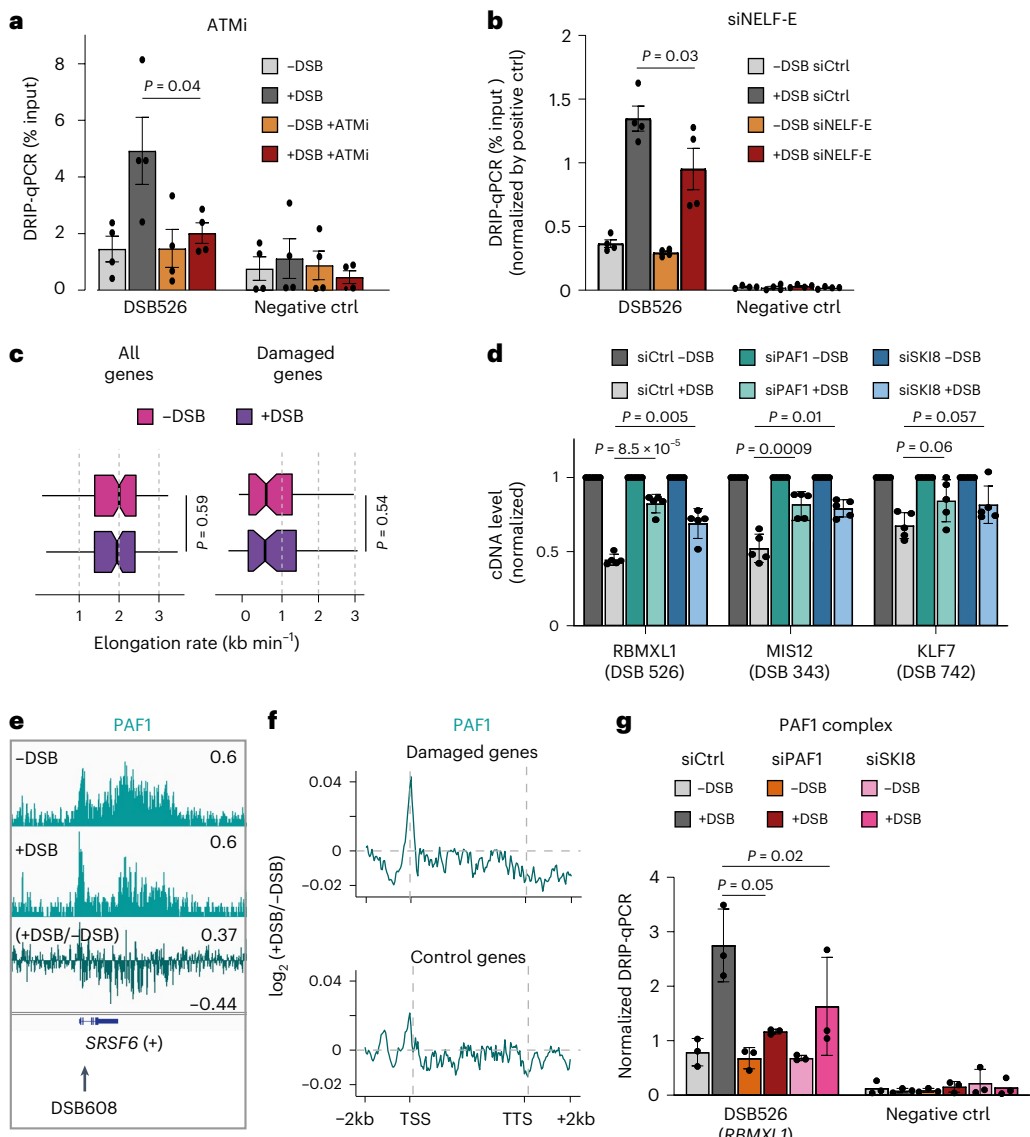

**Fig. 5 | PAF1-dependent transcriptional repression results in RNA:DNA hybrid accumulation at TC-DSBs. a**, DRIP–qPCR before and after DSB induction upon ATM inhibition (ATMi) at a Ctrl Neg region and at a TC-DSB in the *RBMXL1* gene (DSB 526). Mean and s.e.m. for *n* = 4 biological replicates are shown. *P* values, paired two-sided *t*-test. **b**, same as **a** for cells transfected with Ctrl or NELF-E siRNA with DRIP–qPCR normalized by the *RPL13A* positive control. **c**, Measurement of elongation rates by DRB/TT_chem-seq across genes >60 kb for all genes (left, *n* = 5,644) or damaged genes (right, *n* = 49) before (−DSB) and after DSB induction (+DSB). Centre line shows the median; box limits represent first and third quartiles; whiskers show maximum and minimum without outliers; points show outliers. *P* values nonparametric paired two-samples Wilcoxon tests. **d**, RT–qPCR quantifying cDNA levels normalized by TBP before (−DSB)

and after DSB induction (+DSB) for three damaged genes (*RBMXL1*, *MIS12* and *KLF7*) carrying TC-DSBs in cells transfected with control (Ctrl), PAF1 or SKI8 siRNA. Mean and s.e.m. for *n* = 5 biological replicates are shown. *P* values, paired two-sided *t*-tests. **e**, Genomic tracks (hg19) of PAF1 ChIP-seq before and after DSB induction as well as the log₂ fold change ratio (+DSB/−DSB) at a TC-DSB (DSB 608). **f**, Average profiles of PAF1 ChIP-seq enrichment following DSB induction as log₂ fold change ratio (+DSB/−DSB) for damaged and control genes. **g**, DRIP–qPCR normalized by the *LYRM1* positive control before and after DSB induction at a Ctrl Neg region and at DSB 526 in cells transfected with Ctrl, PAF1 or SKI8 siRNA. Mean and s.e.m. for *n* = 3 biological replicates are shown. *P* values, paired two-sided *t*-tests. Source numerical data are available in Source data.

efficiently induced (Extended Data Fig. 1e and Extended Data Fig. 6d) and those induced at *RBMXL1* triggered transcriptional downregulation as expected (Extended Data Fig. 6e). As observed at AsiSI-induced DSBs, RNA:DNA hybrids did not accumulate at the Cas9-DSB in the intergenic locus (sgRNA3; Fig. 4d), but did accumulate at DSBs in *RBMXL1*, irrespective of the DSB position (promoter, intron or exon) (Fig. 4d).

We also took advantage of Cas9-induced DSBs to determine whether RNA:DNA hybrids at DSB could form as a result of the hybridization of a *trans*-produced RNA (arising from the undamaged allele in diploid cells) rather than the *cis*-produced RNAs. For this, we induced

the sgRNA1/Cas9-DSB in the HAP1 haploid cell line. RNA:DNA hybrids also formed at the DSB induced in *RBMXL1* in this condition (Extended Data Fig. 6f). This suggests that DSB-induced RNA:DNA hybrids result from *cis*-produced RNAs rather than *trans* RNAs.

Altogether, our data suggest that the RNA moiety engaged in RNA:DNA hybrids forming in *cis* to DSB mostly corresponds to pre-existing RNA species hybridizing back to their respective template strands, leading to asymmetric and strand-specific RNA:DNA hybrids at TC-DSBs. Notably, this process can involve pre-mRNAs as well as unstable transcripts such PROMPTs or intron-containing RNAs.

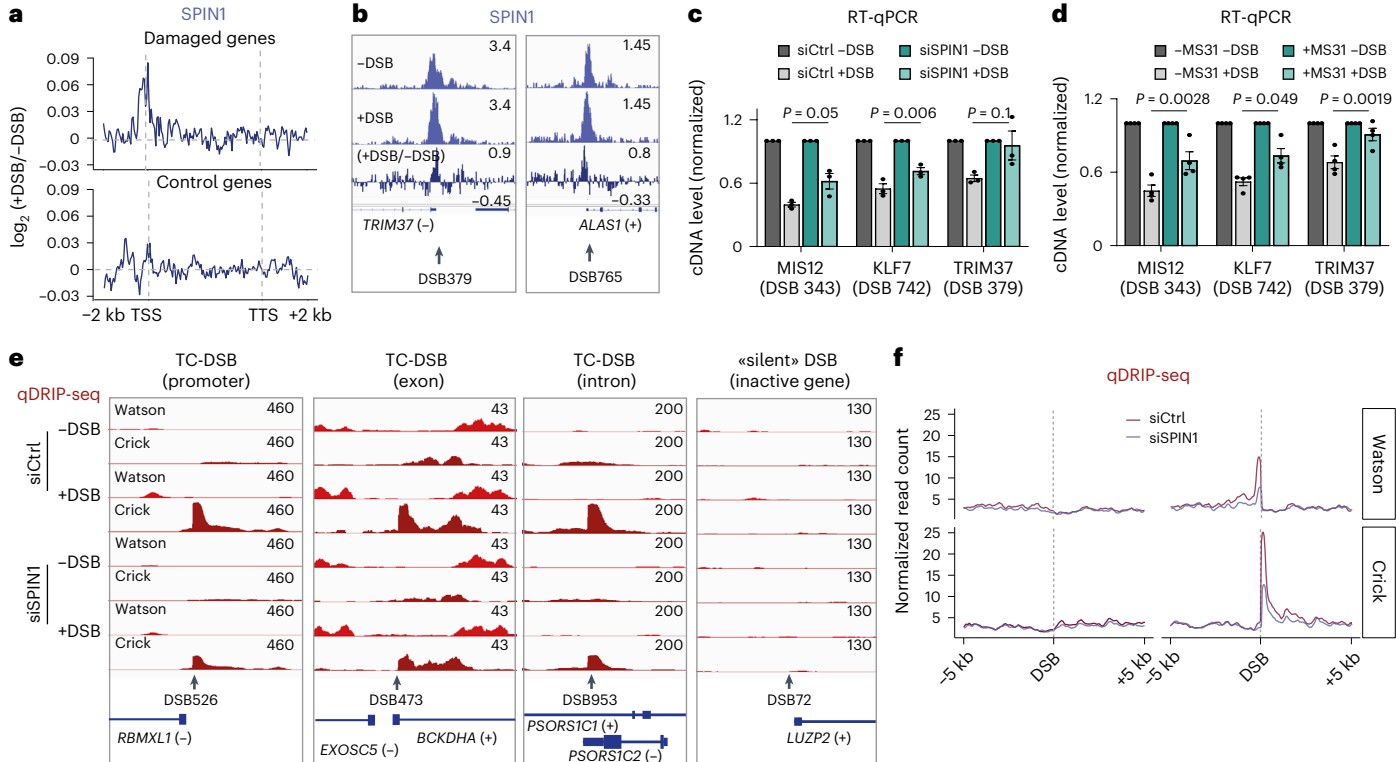

**Fig. 6 | SPIN1-dependent transcriptional repression results in RNA:DNA hybrid accumulation at TC-DSBs. a**, Average profiles of SPIN1 ChIP-seq enrichment following DSB induction as $log_2$ fold change ratio (+DSB/−DSB) for damaged and control genes. **b**, Genomic tracks (hg19) of SPIN1 ChIP-seq before and after DSB induction as well as the $log_2$ fold change ratio (+DSB/−DSB) at two TC-DSBs (DSB 379 and DSB 765). **c**, RT–qPCR quantifying cDNA levels normalized by RPLP0 before (−DSB) and after DSB induction (+DSB) for three genes (*MIS12*, *KLF7* and *TRIM37*) carrying TC-DSBs in cells transfected with control (Ctrl) or SPIN1 siRNA.

Mean and s.e.m. for *n* = 3 biological replicates are shown. *P* values, paired two-sided *t*-tests. **d**, Same as **c** for cells treated with MS31. *n* = 4 biological replicates. **e**, Genomic tracks (hg19) showing qDRIP-seq signal in siCtrl and siSPIN1-transfected cells on Watson and Crick strands in −DSB and +DSB conditions at TC-DSBs located in a promoter, exon or intron and in a silent DSB. **f**, Average profile of the qDRIP-seq signal in siCtrl and siSPIN1 cells on the Watson and Crick strands on ± 5 kb around the 80 best-cleaved DSBs. Source numerical data are available in Source data.

## PAF1- and SPIN1-dependent transcriptional repression accounts for RNA:DNA hybrid accumulation at TC-DSBs

Given the above results, we hypothesized that the transcriptional repression observed in *cis* to DSBs may trigger increased retention of pre-mRNA or PROMPT, providing more opportunities for hybridization with the ssDNA overhang left after resection and therefore accounting for RNA:DNA hybrid accumulation. To test this, we first inhibited the ATM kinase, which was previously identified as being required for DSB-induced transcriptional repression[44] (Extended Data Fig. 7a). As predicted, ATM inhibition impaired the accumulation of RNA:DNA hybrids detected by DRIP–qPCR (Fig. 5a). Depletion of NELF-E, a pausing factor for RNAPII also reported to participate in DSB-induced *cis* transcriptional repression[45] (Extended Data Fig. 7b,c), similarly decreased RNA:DNA hybrid levels at TC-DSBs (Fig. 5b). These results suggest that RNA:DNA hybrids arise downstream of transcriptional repression that triggers a local retention of the pre-existing RNA (pre-mRNA or PROMPT) and thus favours its hybridization to the template strand converted into a ssDNA overhang by DNA end processing.

As R-loop accumulation has been previously associated with reduced elongation speed at the 5′ and 3′ ends of genes outside the context of DNA damage[31] and given the elongation defect detected at damaged genes (Fig. 2d–h), we aimed to evaluate whether DSBs could be responsible for a reduction in RNAPII elongation velocity by performing DRB/TT$_{chem}$-seq[37]. This method allows to quantify the progression of RNAPII by measuring nascent transcripts at various time points (5, 10, 20, 30 and 40 min) following the release of a transient inhibition of RNAPII elongation by 5,6-dichloro-1-β-D-ribofurano sylbenzimidazole (DRB) (Extended Data Fig. 7d,e). DRB/TT$_{chem}$-seq

allowed to recapitulate the average elongation rate of 2 kb min⁻¹ in the absence of damage[37] but failed to identify significant changes in RNAPII velocity after DSB induction, either at the level of all genes or when focusing specifically on damaged genes (Fig. 5c).

To gain further insights into the mechanisms triggering such a decrease in RNAPII productive elongation following DSBs, we performed a mini-siRNA screen that assessed DSB-induced transcription inhibition upon depletion of various proteins involved in transcriptional pausing or termination (DGCR8, the integrator subunits INTS11 and INTS12, EXOSC10, XRN2, SCAF4, PAF1 and SKI8/WDR61 (ref. 46)). Out of the eight proteins tested, the depletion of two proteins from the PAF1 complex, PAF1 and SKI8, led to impaired DSB-induced transcriptional repression (Fig. 5d and Extended Data Fig. 7f,g). In human cells, the PAF1 complex counteracts the transition of RNAPII to a productive elongation mode, and its acute depletion triggers unscheduled RNAPII escape from the promoter-proximal pausing site[47]. We hence performed ChIP-seq against PAF1 in damaged and undamaged cells and obtained the expected PAF1 profile across all genes, validating our dataset (Extended Data Fig. 7h). Of interest, at damaged genes, and in contrast to a control gene set, DSB induction triggered an increase of PAF1 at the 5′ end (Fig. 5e,f and Extended Data Fig. 7i), indicating that PAF1 is recruited at damaged genes, where it contributes to transcriptional repression. Notably, PAF1 depletion indeed led to a reduction in DSB-induced RNA:DNA hybrid levels (Fig. 5g), in agreement with our model that transcriptional repression accounts for hybrid accumulation.

Finally, to identify additional players in DSB-induced transcriptional repression, we mined data from a proteomic screen set to

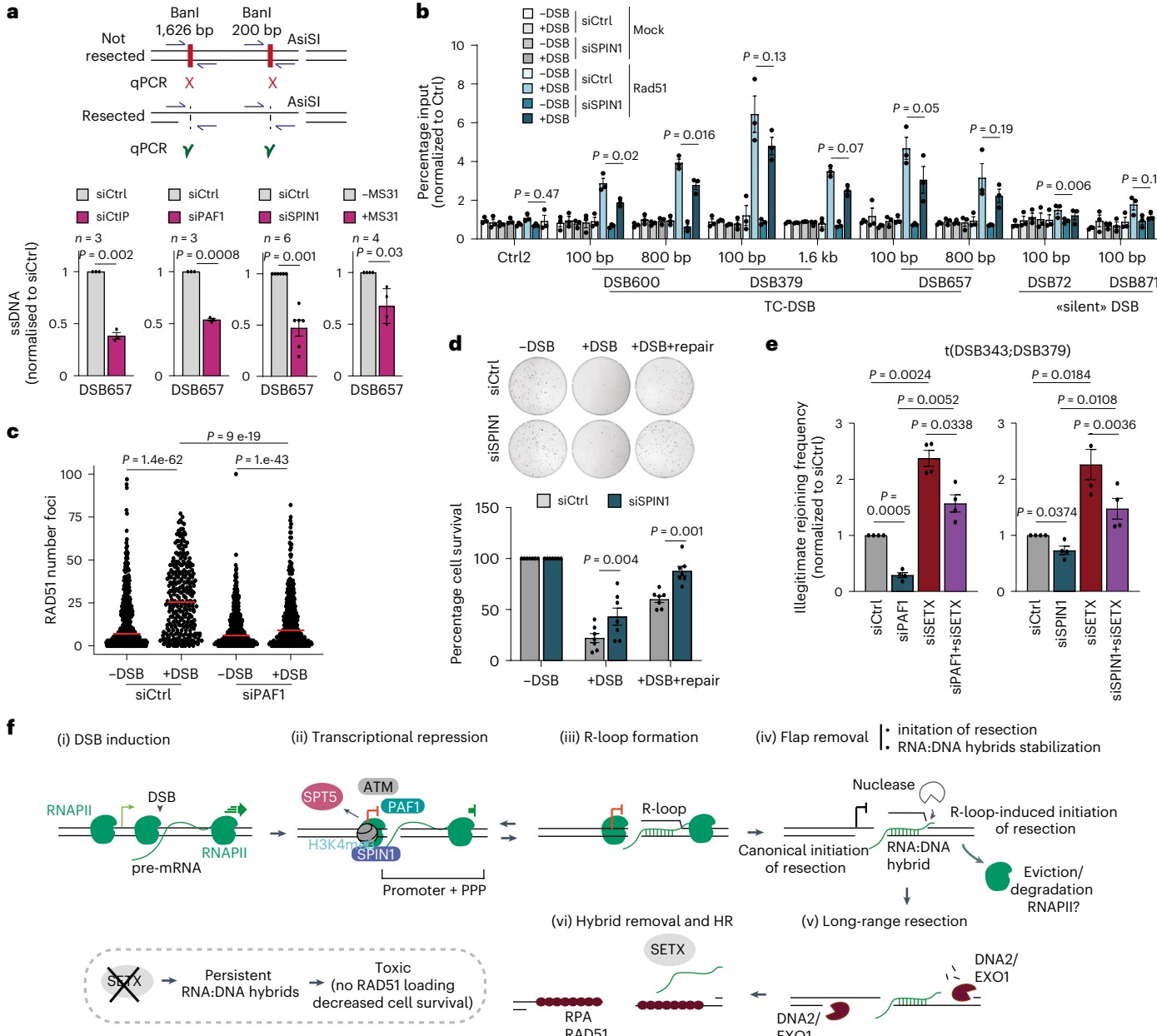

**Fig. 7 | RNA:DNA hybrids induced by transcriptional repression at DSBs promote resection and toxicity. a**, Schematic representation of the resection assay (top) and quantification of ssDNA after DSB in siCtrl, siCtIP, siPAF1, siSPIN1 or upon MS31 treatment at 200 bp away from a TC-DSB (DSB 657). Mean and s.e.m. for $n \geq 3$ biological replicates are shown. $P$ values, paired two-sided $t$-test. **b**, RAD51 ChIP–qPCR efficiency (normalized to undamaged Ctrl1) before (−DSB) or after DSB (+DSB) in control or SPIN1 siRNA-treated cells at 100 bp and 800 bp from TC-DSBs (DSB 600, 379 and 657) and silent DSBs (DSB 72 and 871). Mean and s.e.m. for $n = 3$ biological replicates are shown. $P$ values, paired two-sided $t$-tests. **c**, Representative example showing the number of RAD51 foci in siCtrl- or siPAF1-transfected cells in −DSB and +DSB conditions. Red shows the median. $P$ values, paired two-sided nonparametric Wilcoxon tests. **d**, Clonogenic assay in control (Ctrl) or SPIN1 siRNA-treated DIvA-AID cells. A representative experiment is shown on the top panel. Mean and s.e.m. for $n = 7$ biological replicates are shown (bottom). $P$ values, paired two-sided $t$-tests. **e**, Illegitimate rejoining frequency between two TC-DSBs (DSB 343 and DSB 379) in DIvA-AID cells

(+DSB +repair) and normalized to siCtrl upon siPAF1, siSETX or siPAF1+ siSETX (left) or upon siCtrl, siSPIN1, siSETX or siSPIN1+siSETX (right). Mean and s.e.m. for $n = 4$ biological replicates are shown. $P$ values, paired two-sided $t$-tests. **f**, Model for RNA:DNA hybrid formation at TC-DSBs. Upon DSB induction within a transcribing gene, an ATM-dependent pathway, which entails the recruitment of PAF1 and of the H3K4me3 reader SPIN1, as well as the eviction of SPT5, triggers transcriptional repression via a decrease in PPP release. As a consequence, R-loops accumulate on the template strand of the damaged gene. The displaced ssDNA further becomes a substrate for endonucleolytic cleavage (flap removal), thus simultaneously initiating resection and converting the R-loop into a more stable RNA:DNA hybrid. This R-loop-mediated, alternative resection initiation pathway would commit the DSB to HR repair. RNA:DNA hybrids shall further be removed by SETX and potentially other RNA:DNA helicases, to allow HR repair and to avoid RNA:DNA hybrid-dependent toxicity. Source numerical data are available in Source data.

identify interactors of the SMC1 cohesin subunit before and after DSB induction[48], as the cohesin complex was shown to be a key player in DSB-induced transcriptional shut down[49]. This proteomic analysis identified 114 proteins specifically enriched upon damage, including

spindlin 1 (SPIN1), which was reported both in vitro and in vivo to spe-cifically recognize H3K4me3 (ref. 50), a histone modification, involved in PPP release[51] and previously shown to be involved in transcriptional repression in *cis* to DSBs[52]. To assess whether, similarly to PAF1, SPIN1

accumulates at damaged genes, we performed ChIP-seq experiments. As expected for an H3K4me3 reader, SPIN1 distribution on the genome analysed by ChIP-seq was highly similar to H3K4me3, with an accumulation at TSS and an enrichment at promoters correlating with transcriptional activity (Extended Data Fig. 8a–c). Following DSB induction, SPIN1 specifically increased at promoters of damaged genes (Fig. 6a,b). Furthermore, we found that SPIN1 depletion significantly impaired transcriptional repression at TC-DSBs (Fig. 6c and Extended Data Fig. 8d) suggesting that this factor plays a key role in this process. To establish whether SPIN1 displays this function through H3K4me3 binding, we used MS31, an inhibitor of SPIN1–H3K4me3 interaction. MS31 treatment did not affect overall SPIN1 and H3K4me3 levels (Extended Data Fig. 8e) and MS31 inhibited the recruitment of SPIN1 to chromatin without altering H3K4me3 occupancy (Extended Data Fig. 8f). Similar to SPIN1 depletion, MS31 treatment impaired in *cis* transcriptional repression (Fig. 6d) further involving the SPIN1–H3K4me3 interaction in DSB-induced transcriptional shut down. Altogether, this suggests that SPIN1 plays a role in the repair of TC-DSBs by regulating transcriptional repression through its ability to bind H3K4me3. Of note, qDRIP-seq revealed a reduction in RNA:DNA hybrid levels at TC-DSBs upon SPIN1 depletion (Fig. 6e,f and Extended Data Fig. 8g), which is similar to the results obtained by interfering with other factors involved in transcriptional repression (such as ATM, NELF-E, PAF1 and SKI8).

Collectively, our data suggest that following DSBs, transcriptional repression takes place at damaged genes in part thanks to the eviction of SPT5 (Fig. 2) and the recruitment of SPIN1 and the PAF1 complex. This transcriptional repression pathway further contributes to RNA:DNA hybrid accumulation at the site of damage.

### Transcriptional repression-induced RNA:DNA hybrids at DSBs foster resection and are toxic

Given our above findings that RNA:DNA hybrid accumulation may foster resection initiation (Fig. 3), we postulated that interfering with transcriptional repression could impair DSB end processing by decreasing RNA:DNA hybrids. To measure resection, we used a previously established assay allowing the quantification of ssDNA by qPCR following enzymatic digestion of genomic DNA (Fig. 7a). As expected, CtIP (used as a positive control), PAF1 and SPIN1 depletions, as well as MS31 treatment triggered decreased resection (Fig. 7a). Moreover, SPIN1 depletion reduced RAD51 levels at TC-DSBs (Fig. 7b) and PAF1 depletion impaired RAD51 foci formation (Fig. 7c). In agreement, MS31 treatment also impaired the formation of RAD51 and BRCA1 foci following DSB induction with etoposide (Extended Data Fig. 9a). Altogether, these data point to a role of transcriptional repression in initiating resection and promoting HR. This is consistent with previous studies that closely linked transcriptional repression to resection[4] and that identified SPIN1 (in human) and PAF1 (in plants) as contributing factors to HR[53,54].

We previously reported that excessive RNA:DNA hybrid accumulation at TC-DSB is toxic[8,43]. We thus investigated whether impairing DSB-induced transcriptional repression and therefore RNA:DNA hybrid accumulation could be beneficial for cell survival following TC-DSB induction. We found that SPIN1 depletion or MS31 treatment increased clonogenic potential after DSB induction (Fig. 7d and Extended Data Fig. 9b), in agreement with their function in RNA:DNA hybrid formation. Notably, MS31 treatment also potentiated cell survival following DSB induction with etoposide (Extended Data Fig. 9c). Similarly, depletion of NELF-E, which is involved in transcriptional repression, also increased cell survival post-DSB induction, in agreement with a role of transcriptional repression in RNA:DNA hybrid formation at DSBs (Extended Data Fig. 9d). We previously reported that impairing RNA:DNA hybrid removal via SETX depletion increased cell death and illegitimate rejoining events between distant DSBs following TC-DSB induction[8]. Here, we found that PAF1 and SPIN1 depletion reduced the frequency of illegitimate rejoining events observed upon SETX

knockdown (Fig. 7e) and that SPIN1 depletion partially alleviated the toxicity of SETX depletion (Extended Data Fig. 9e), as predicted if these factors act upstream of RNA:DNA hybrid formation. Overall, these data suggest that the transcriptional repression induced in *cis* to DSB impairs cell survival in a manner that relies on the induction of toxic RNA:DNA hybrids at DSBs.

## Discussion

In this manuscript, we aimed at identifying the mechanisms responsible for RNA:DNA hybrid accumulation at DSBs. Altogether, our data showed that DSB-induced RNA:DNA hybrids (1) form on the ssDNA overhang generated by end resection, (2) do not form at all DSBs but mostly at TC-DSBs (DSB generated in RNAPII-enriched loci, undergoing transcription), and (3) form unidirectionally and on the transcribed strand, suggesting that pre-existing RNA located at the damaged locus largely accounts for the RNA moiety of RNA:DNA hybrids. RNA:DNA hybrid accumulation is fostered by transcriptional repression that depends on the H3K4me3 reader SPIN1 and the transcriptional regulator PAF1. While RNA:DNA hybrids may promote the initiation of resection, their removal is further necessary to ensure cell survival and faithful TC-DSB repair (Fig. 7f).

### DSB-induced RNA:DNA hybrids mostly originate from the hybridization of pre-existing RNA

De novo recruitment of RNAPII/III at DNA ends has been proposed to account for RNA:DNA hybrid formation at sites of DSBs[16,55] although the identity of the RNA polymerase involved is debated[13,15,16]. Here, by analysing RNAPII and RNAPIII recruitment using both ChIP–qPCR and high-resolution genome-wide ChIP-seq at (1) a large number of enzymatically induced and annotated DSBs in euchromatin, (2) CRISPR/Cas9-induced DSBs, and (3) etoposide-induced DSBs, we failed to provide evidence of the recruitment of either RNAPII or RNAPIII at sites of damage although RNA:DNA hybrid accumulation was readily detectable (refs. 8,43 and this study). Consistently, TT$_{chem}$-seq did not detect de novo nascent transcription initiating from DSBs. Thus, our findings are in agreement with previous studies[30,36,44] but contrast with other reports[7,12–16,56], describing RNAPII or RNAPIII accrual at sites of DNA damage induced by micro-irradiation using imaging[14–16] or at nuclease-induced DSBs using ChIP[7,12,15,16,56]. Although the reasons for such discrepancies are yet unclear, we can envision several potential explanations. First, live-imaging at sites of micro-irradiation, while providing valuable kinetic parameters, will be strongly influenced by the extent and variety of laser-induced DNA damage as well as overexpression of the fluorescently tagged protein. Furthermore, local changes in protein dynamics (such as the decreased RNAPII elongation or alterations in PPP release) might be difficult to distinguish from de novo recruitment in such assays. Second, RNA polymerase recruitment might be too transient for detection by ChIP-seq. Indeed, because AsiSI-induced DSB can be quickly re-generated upon faithful repair, ChIP-seq experiments on cell populations represent a mixture of events ranging from early cleavage to late repair. While this allows sampling of the various steps in the repair process, some transient states may remain below the detection threshold, especially if a single molecule of RNAPII is recruited at the DSB in a very transient manner. However, we also failed to detect RNAPII recruitment using more sensitive, quantitative PCR on ChIP samples, at AsiSI-induced DSBs, but also at CRISPR/Cas9 and etoposide-induced breaks (Fig. 1e,g and Extended Data Fig. 1i) and under similar experimental conditions, the transient binding of NHEJ factors such as XRCC4 and LIG4 is easily detectable by ChIP-seq at all AsiSI-induced DSBs[32,33]. Further investigations using advanced microscopy should help to determine whether RNA polymerases can be transiently loaded to DNA ends in vivo. Third, it remains possible that RNAPII/III recruitment and de novo RNA synthesis is specific for DSB induced in heterochromatin, which we did not investigate in our study.

Irrespective of the reason underlying such discrepancies, while RNAPII/III recruitment at DSBs was not observed, we readily detected RNA:DNA hybrid accumulation at DSBs as previously reported[7–9,16,18,22,27,43,55–57]. Of importance, these RNA:DNA hybrids not only form at DSB mostly located in RNAPII-bound loci (transcribing loci, enriched in RNAPII before DSB induction; in agreement with refs. [8,21,22]), but their RNA moiety mainly corresponds to pre-existing RNAs transcribed from the damaged locus (either pre-mRNA or PROMPT). Altogether, while we cannot formally exclude that a very transient recruitment of RNA polymerase at DNA ends could contribute to the formation of the RNA:DNA hybrids via de novo RNA synthesis, our data rather indicate that most of the RNA:DNA hybrids forming at DSBs arise from pre-existing RNA hybridizing to its template strand.

### PAF1 and SPIN1 contribute to in cis DSB-induced transcriptional repression required for RNA:DNA hybrid formation

Following DSB induction, an ATM- and DNA-PK-dependent signalling pathway elicits rapid in *cis* inhibition of RNAPII transcription[36,44]. Mechanistically, this DSB-induced gene repression pathway entails the recruitment of several factors such as NELF-E as well as alterations in the histone modification landscape at the promoter of damaged genes, including KDM5A-mediated H3K4me3 demethylation (reviewed previously,[28,29]). We found that this transcriptional shut down is accompanied by reduced RNAPII elongation, as measured with $TT_{chem}$-seq, and the eviction of SPT5, an elongation factor enabling RNAPII escape from promoter-proximal pausing. Interestingly, outside the DNA damaging context, acute depletion of SPT5 triggers the degradation of RNAPII arrested at PPP[38,39]. As eviction/degradation of RNAPII was previously reported in *cis* to DSBs[30], in agreement with the observed decreased RNAPII occupancy reported here (Figs. 1 and 2), we can speculate that a similar mechanism may allow proteasomal degradation of promoter-proximally arrested and elongation incompetent RNAPII complexes accumulating at damaged genes.

We also identified two additional players, namely, SPIN1 and PAF1, as being recruited at damaged promoters and required for DSB-induced transcriptional repression. SPIN1 was reported to specifically interact with H3K4me3 (ref. [50]), a histone mark required for the efficient release of RNAPII from promoter-proximal pausing[51]. As for the PAF1 complex, it has also been involved in the regulation of PPP release in a manner that depends on the recruitment of the integrator–PP2A complex[47,58]. Of interest, PAF1 mediates RNAPII removal from transcription blocking lesions[59] and the INTS6 subunit of the integrator complex, which mediates promoter-proximal termination[60], was recently reported to be recruited at DSBs[61]. Altogether, this points towards a model whereby DSB-induced transcriptional repression occurs, at least in part, through a H3K4me3/SPIN1 and PAF1-dependent pathway which may regulate early productive elongation and escape from promoter-proximal pausing site at damaged genes. Notably, the depletion or inactivation of several of the molecular players in transcriptional repression (ATM, NELF-E, SPIN1 and two subunits of the PAF1 complex) triggered a decrease in DSB-induced RNA:DNA hybrids, suggesting that this pathway is involved in the formation of RNA:DNA hybrids at TC-DSBs. Given that most TC-DSBs induced by AsiSI (Extended Data Fig. 1a) or etoposide[62] are induced in the 5′ end of genes, where RNAPII would arrest upon damage, one can speculate that the retention of paused RNAPII combined with the release of topological constraints proximal to DSB sites[22] could enhance the frequency of annealing of the pre-existing RNA to its template DNA strand, thereby forming R-loops at sites of damage. These R-loops would be rapidly converted into more stable RNA:DNA hybrids, as soon as resection initiation takes place (Fig. 7f) (see below).

### RNA:DNA hybrids promote short-range resection

We found that the DNA moiety of RNA:DNA hybrids mostly corresponds to the ssDNA overhang left by end resection, in agreement with a recent report showing colocalization of both RNA and ssDNA tracts[18]. Notably, we also found that RNA:DNA hybrid accumulation stimulates RPA ssDNA binding on a short scale (<1 kb) around the DSB, in agreement with previous studies suggesting that R-loops potentiate resection (for instance, refs. [5,27,57]). We propose that transcriptional repression-induced R-loops, by displacing the 5′-terminated strand, fosters the initiation of resection through flap removal by endonucleolytic cleavage (Fig. 7f). RNA:DNA hybrids may also contribute to KU70/80 recruitment at TC-DSBs (as at stressed replication forks in yeast[63]) further favouring MRN recruitment and nicking activity[64,65]. Endonuclease-mediated nicking would simultaneously stabilize the RNA:DNA hybrid and produce a favoured substrate for long-range resection nucleases (Fig. 7f). Although it was recently shown in yeast that transcription rather impedes MRX-dependent 5′ strand nicking[65], it was previously proposed that XPG, the MRN complex and CtIP, carrying endonucleolytic activities[66,67] can mediate non-canonical resection at sites of RNA:DNA hybrid accumulation[6,27,57]. Such an alternative pathway for priming resection would agree with the involvement of PAF1 and SPIN1 in HR (this study and refs. [53,54]) and could account for the long-observed link between transcriptional repression and resection/HR.

Yet, while contributing to resection, we show here that these transcriptional repression-induced RNA:DNA hybrids are toxic if not removed (Fig. 7). In agreement, excessive RNA:DNA hybrid accumulation at sites of damage has been associated with defective HR repair[8,22,23,25,27], deletions that arise on the side of the hybrid[6,7,24,27] and an increase in translocation frequency[8], likely accounting for the observed toxicity.

In summary, we propose that RNA:DNA hybrids form at sites of damage mainly due to the annealing of pre-existing RNA. While they can prime DSB for resection, they are globally detrimental for cell survival and increase the frequency of translocations suggesting that they represent by-products and impediments to the repair process.

## Online content

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

## Methods

### Cell culture and treatments

DIvA (AsiSI-ER-U2OS)[32] AID-DIvA[68], Lenti-X HEK293T (Clontech, 632180) (for producing Nanoblade viral-derived particles) and HAP1 cells (C859, Horizon) were cultured in Dulbecco's modified Eagle's medium (DMEM) supplemented with antibiotics, 10% FCS (Invitrogen) and either 1 µg ml$^{-1}$ puromycin (DIvA cells) or 800 µg ml$^{-1}$ G418 (AID-DIvA cells) at 37 °C under a humidified atmosphere with 5% $CO_2$. AsiSI-dependent DSBs were induced by the addition of 300 nM 4-OHT (Sigma, H7904) for 4 h or 24 h. AsiSI degradation in AID-DIvA (DSB repair) was performed via the addition of 500 µg ml$^{-1}$ auxin (IAA, Sigma; I5148) for 4 h. Etoposide (Sigma, E1383) was used at 50 µM for 30 min for END-seq, 50 µM for 1 h for ChIP, 1 µM for 4 h for immunofluorescence and 0.5, 1, 2 and 5 µM for 4 h for clonogenic assays. MS31 (MedChemExpress, HY-125837A) was used at 50 µM and added 30 min before 4-OHT treatment. ATMi (KU-55933, Sigma, SML1109) was used at 20 µM and added to 1 h before 4-OHT treatment. MRE11 inhibition was performed using PFM01 or Mirin (MedChemExpress, respectively HY-116770 and HY-117693) at 100 µM and added 1 h before 4-OHT treatment. siRNA (Supplementary Table 2) transfections were performed using 4D-Nucleofector from Lonza according to the manufacturer's instruction. Cells were collected 48 h after siRNA transfection.

### DSB induction with CRISPR/Cas9

For ChIP, Cas9/sgRNA ribonucleoprotein complexes were delivered using Nanoblades viral-derived particles as in refs. 48,69 and cells were collected 7 h after transduction. sgRNA sequences (Supplementary Table 3) were cloned in the BLADE plasmid (a gift from P. Mangeot, T. Ohlmann and E. Ricci, (Addgene plasmid #134912)). For quantitative PCR with reverse transcription (RT–qPCR) and DRIP–qPCR, DSBs were induced by delivering sgRNA as ribonucleoprotein complexes using the Alt-R CRISPR-Cas9 System from Integrated DNA Technologies according to the manufacturer's instructions. RNP complexes were delivered into DIvA cells using Lipofectamine CRISPRMAX (Invitrogen). After 7 h, cells were collected for RNA extraction or DRIP.

### Chromatin immunoprecipitation

ChIP experiments were performed as in refs. 32,68. Specifically, 10–200 µg chromatin was incubated with antibodies overnight (Supplementary Table 4). ChIP samples were analysed by qPCR (primers listed in Supplementary Table 5) in a CFX384 real-time system (Bio-Rad) using Bio-Rad CFX Manager v.3.1 software. Immunoprecipitation efficiencies were calculated as the per cent of input DNA immunoprecipitated and data were normalized to a control region far from any DSB induced by AsiSI, CRISPR/Cas9 or Etoposide without DSB induction.

For ChIP-seq, multiple ChIP experiments were pooled, sonicated for 10–15 cycles (30 s on, 30 s off, high setting) with a Bioruptor (Diagenode), then concentrated with a vacuum concentrator (Eppendorf). For SPIN1, H3K4me3, total RNAPII, RNAPII S2P, RNAPII S5P, RNAPII S7P, RNAPII Y1P, POLR3A, POLR3E, SPT5 and PAF1 ChIP-seq, about 0.5–10 ng purified DNA was subjected to library preparation (NEBNext Ultra II DNA Library kit, NEB). For RPA ChIP-seq, immunoprecipitated DNA was subjected to a strand-specific library preparation (Accel-NGS 1S Plus DNA library kit, Swift Biosciences) according to the manufacturer's instructions. Sequencing depth and read length are included in the reporting summary.

### DRIP–qPCR and qDRIP-seq

DRIP assay was carried out as in ref. 8 and assayed by qPCR (primers listed in Supplementary Table 6).

qDRIP-seq was adapted from ref. 40. In brief, $2.5 × 10^6$ of OHT-treated or untreated DIvA cells were mixed $1.67 × 10^6$ *Drosophila* S2 cells and lysed overnight. Immunoprecipitation was performed in triplicate by incubating 4 µg sonicated DNA with 10 µg S9.6 antibody (Antibodies Incorporated). Sequencing libraries were prepared using

the Accel-NGS 1S Plus DNA library kit (Swift Biosciences) according to the manufacturer's instructions using 12 PCR cycles. Libraries were pooled at equimolar concentrations and sequenced via an Illumina NextSeq500 system using 75 paired-end reads at European Molecular Biology Laboratory (EMBL) GeneCore.

### END-seq

END-seq was performed according to ref. 70 with the following minor modifications: $5–7 × 10^6$ DIvA cells per condition were embedded into two agarose plugs and digestion with Exonuclease VII and T was performed. All washing steps were performed in 15-ml tubes instead of 50-ml tubes. Then, 20 cycles of PCR reaction were achieved with NEB-Next multiplex oligos for Illumina (E7335S, New England Biolabs) and PCR products were purified with SPRIselect or Ampure XP beads (Beckman Coulter) using first a size selection with a 0.6× beads-to-sample ratio followed by two consecutive purifications with a 0.9× ratio. Typically, 6–9 END-seq conditions were pooled in equimolar concentration for a single sequencing run of 50 bp paired-end reads on a NextSeq 2000 at EMBL GeneCore.

### TT$_{chem}$-seq and DRB/TT$_{chem}$-seq

TT-seq and DRB/TT-seq were performed according to ref. 37 with minor modifications. Total RNA spike-ins were prepared from mid-log phase *Saccharomyces cerevisiae* (BY4741 strain) cells grown in YPD + 2% glucose and labelled with 5 mM 4-thiouracil (Sigma-Aldrich, 440736) for 5 min. Total RNA was extracted using the PureLink RNA Mini kit (Thermo Fisher Scientific, 12183020, enzymatic protocol, including a 20 min on-column treatment with RQ1 DNase (Promega, M6101)).

For TT$_{chem}$-seq, nascent RNA was labelled with 1 mM 4-thiouridine (4SU, Glentham Life Sciences, GN6085) for 15 min. After medium removal and a wash with 1× PBS, TRIzol (Thermo Fisher) was added and cells were scraped. Control samples without 4SU or with an overnight incubation with 0.2 mM 4SU were also generated.

For DRB/TT$_{chem}$-seq, DSBs were induced by treating DIvA cells with 300 nM OHT for 3 h, followed by a 1 h treatment with 100 µM DRB (Sigma-Aldrich, D1916). DRB inhibition was released by two washes with pre-warmed PBS and addition of fresh medium. 4SU labelling was performed directly after DRB release (5- and 10-min time points) or as 10-min pulses starting 10, 20 or 30 min after the DRB release (20-, 30- and 40-min time points). Cells were collected with TRIzol as for TT$_{chem}$-seq.

Total RNA was isolated using standard chloroform extraction and NaCl/isopropanol precipitation, treated with RQ1 DNase (Promega, M6101) and purified by phenol–chloroform extraction and NaCl/isopropanol precipitation. Then, 100 µg of human 4SU-labelled RNA was spiked-in with 1 µg of 4-thiouracil-labelled *S. cerevisiae* RNA. Chemical RNA fragmentation, biotin conjugation and streptavidin purification were performed according to ref. 37. Sequencing libraries were prepared using the NEBNext Ultra II Directional RNA library prep kit (NEB E7760) omitting the initial fragmentation step and sequenced on a Nextseq500 in high mode for 75-bp single-end reads at EMBL GeneCore.

### Annotation of AsiSI-induced DSBs

The 80 robustly induced DSBs[33] were manually annotated into specific categories based on transcriptional activity and genomic location using Integrated Genome Viewer (v.2.14.1) with hg19 NCBI RefSeq annotation (Supplementary Table 1). DSBs lying 1 kb upstream of the TSS or within a gene body with visible coverage of TT$_{chem}$-seq and RNA-seq in non-treated cells were classified as TC-DSBs ($n$ = 65). DSBs with no visible TT$_{chem}$-seq and RNA-seq coverage and intergenic DSBs were identified as silent DSBs ($n$ = 15).

TC-DSBs were further categorized into three groups based on location relative to the gene body: 31 DSBs in promoters (within 1 kb upstream of the TSS), 36 DSBs in the 5′ region of genes (within 1 kb downstream from the TSS) and 9 gene body DSBs (intragenic DSBs

occurring more than 1 kb downstream of the TSS). TC-DSBs were also classified by visual inspection as either exonic (19 DSBs) or intronic (26 DSBs). Of note, 11 TC-DSBs lied within overlapping genes or lying within one gene but also located less than 1 kb away from the TSS of another gene. These TC-DSBs were associated with both genes (provided that both exhibited visible TT$_{chem}$-seq and RNA-seq signal) and can therefore display two different annotations (Supplementary Table 1).

To classify DSBs based on transcriptional activity (Extended Data Fig. 1f), total RNAPII ChIP-seq from DIvA cells before DSB induction were computed on a window of ±5 kb around DSBs and categorized into low, medium, and high RNAPII groups of 30 DSBs each.

Control genes were selected by binning damaged genes (DSB within a gene or DSB within 1 kb upstream from the TSS) into five groups based on RNA-seq counts in non-treated cells. Genes were then sampled to have the same number of genes per expression category. Genes within ±1 Mb from all annotated AsiSI sites were excluded as well as significantly upregulated or downregulated genes previously identified from RNA-seq differential expression analysis after 4 h OHT treatment[71].

### High-throughput sequencing data analyses

Raw sequencing files (fastq) were quality-checked using FastQC. Data were aligned to hg19 with bwa-mem, sorted and duplicates were removed using SAMtools. BigWig coverage tracks were generated using bamCoverage from deeptools and, unless specified, normalized by the total read count for each sample. Strand-specific library samples were processed using SAMtools view and merge with flags filters 80 and 160 for reverse fragments and 96 and 144 for forward fragments to generate strand-specific bam and subsequent bigWig coverage files. Differential coverage was calculated using BigwigCompare from deeptools with default bin size. Overall data handling and representation was performed in R 4.2 using different Bioconductor and tidyverse packages (GenomicRanges, plyranges, rtracklayer, dplyr, reshape2, stringr and ggplot2). Data were visualized using Integrated Genome Browser (https://www.bioviz.org/) and Integrated Genome Viewer (https://igv.org). Further data analysis specifications for each sequencing protocol are outlined below.

### ChIP-seq

For total RNAPII, RNAPII S5P, RNAPII S7P, RNAPII S2P, RNAPII Y1P and SPT5, we calculated scale factors to compensate for the potential global transcriptional repression upon damage as described in ref. 34. For each dataset, a coverage matrix was generated for all Ensembl protein coding genes including 2 kb upstream of the TSS and 2 kb downstream from the TTS. Gene bodies were divided into 100 bins (of various sizes to compensate for gene length) while the 2-kb regions upstream of the TSS and downstream TTS were both divided into 40 bins of 50 bp. The average signal per bin was then calculated to retrieve a metagene profile for −DSB and +DSB conditions. Scale factors were then determined by extracting the average signal around the TSS (bins 30 to 50) in the +DSB condition, and dividing that by the average signal around the TSS in the −DSB condition. The calculated scaling factors were then applied to the existing coverage tracks in R and exported as bigWig files.

### qDRIP-seq data processing

Reads were trimmed 15 bp using Trimmomatic v.0.39 to remove remaining primers according to the recommendations (Accel-NGS 1S Plus DNA library kit, Swift Biosciences). Trimmed reads were aligned to a custom reference genome merging hg19 and dm6 (*Drosophila* spike-in) chromosomes, generating two separate bam files. Strand-specific bam files and bigWig coverage tracks were generated as described above and normalized by a scaling factor of 1,000,000/number of reads mapped to dm6.

### END-seq data processing

END-seq strand-specific bigWig coverage tracks were generated using bamCoverage from deeptools using the '−filterRNAstrand' parameter.

### TT$_{chem}$-seq and DRB/TT$_{chem}$-seq data processing

TT$_{chem}$-seq and DRB/TT$_{chem}$-seq reads were processed as previously described[37] (https://github.com/crickbabs/DRB_TT-seq). In brief, reads were separately aligned to both human (hg19) and *S. cerevisiae* (R64-1-1) genome assemblies using STAR109 with the flags --quantMode GeneCounts and --twopassMode Basic. Reads were sorted with SAMtools sort, duplicates marked with PICARD (https://broadinstitute.github.io/picard/) and indexed using SAMtools. BigWig coverage tracks were then normalized by the scaling factor 1,000,000/number of reads mapped to *S. cerevisiae* R64-1-1 using bamCoverage with the '−scaleFactor' flag.

DRB/TTchem-seq metagene profiles, wave peak calling and elongation rate calculations were performed similarly to refs. 37,72. Specifically, BAM files for each DRB time point were used to calculate the coverage (bin size 100 bp) of Ensembl protein coding genes of length >60 kb using the function bamCoverage from the R package bamsignals applying scale factors. The coverage of each gene was calculated 2 kb upstream and 120 kb downstream of the TSS. Metagene profiles were calculated by taking a trimmed mean (0.01) per bin and applying a smoothing spline using the smooth.spline function (spar = 0.3). Elongation rates per gene were determined by calculating the coverage at each time point, fitting a smoothed spline per gene and identifying the 'wave peak' as the spline's maxima. Genes were then filtered out if they contained missing values, if the wave peak did not change through time, and did not proceed the previous time point. Finally, a linear regression model was fitted to the wave peaks to calculate the elongation rate per kb min$^{-1}$.

### Data visualization

Coverage matrices used for averaged profiles, heatmaps, and boxplots were generated using deeptools (https://deeptools.readthedocs.io/en/latest/) and further visualized using custom R scripts or deeptools. Matrices used for averaged metagene profiles were created using deeptools and further visualized using custom R scripts. In brief, the mean coverage of each gene was computed in 200-bp intervals including 3 kb upstream of the TSS and downstream of the transcription end site. Second, gene bodies were divided into 100 equally sized bins, so that average profiles could be computed as a per cent of the entire gene length. For boxplots, the centre line represents the median, box ends represent the first and third quartiles, and whiskers represent the minimum and maximum values without outliers. Outliers were defined as below the first quartile (− 1.5 × interquartile range) and above the third quartile (+ 1.5 × interquartile range).

### Clonogenic assay

After siRNA transfection, 4,000 DIvA-AID cells were seeded in 10-cm dishes. Cells were then treated with 4-OHT followed by two washes with 1× PBS and, when indicated, incubated with IAA 4 h followed by two washes with 1× PBS. DMEM was then added back to the cells, which were incubated for 10 days. In the absence of siRNA transfection, 2,500 DIvA cells were seeded in a 10-cm dish and treated the next day with MS31 before DSB induction with either 4-OHT for 4 h or with different concentrations of etoposide for 4 h. After washing, cells were incubated for 10 days followed by staining with crystal violet (V5265, Sigma). Plates were imaged with the ChemiDoc Touch Imaging System (Bio-Rad) and colonies were counted using the Spot Detector plugin in Icy.

### RNA extraction and RT−qPCR

RNA was extracted with QIAGEN RNeasy kit (QIAGEN) following the manufacturer's instruction. Then, 500 ng of RNA was reverse transcribed using AMV reverse transcriptase (Promega). cDNAs obtained were quantified by qPCR and normalized by RPLP0 cDNA levels (primers listed in Supplementary Table 7). RNA-seq was performed as described in refs. 8,71.

## Resection assay

Resection assay was performed as in refs. 43,73. In brief, cells were collected by scraping. Genomic DNA was extracted using the DNeasy kit (QIAGEN) according to the manufacturer's instructions. DNA was treated with RNase H (NEB) and 200 ng of RNase H-treated DNA was either digested with Ban I (NEB) or left undigested overnight at 37 °C followed by heat inactivation of the enzyme. qPCR was performed using the following primers located at 200 bp from the DSB 657 (in *KDERL3*) Fw: ACCATGAACGTGTTCCGAAT and Rev: GAGCTCCGCAAAGTTTCAAG.

## DSB-induced translocation assay

DNA was extracted using the DNeasy kit (QIAGEN). Illegitimate rejoining frequencies between different AsiSI sites, t(DSB 343;DSB 379), was analysed by qPCR (in 3–4 replicates) using specific primers as in ref. 8. Results were normalized using two control regions (Norm1 and Norm17) both far from any AsiSI sites and γH2AX domains. Normalized translocation frequencies were calculated using the Bio-Rad CFX Manager v.3.1 software (primers listed in Supplementary Table 8).

## ddPCR assay for cleavage efficiency

The cleavage efficiency of DSB 526 was measured using ddPCR using primers and probes designed to amplify a region spanning the AsiSI site of DSB 526, for which cleaved alleles do not produce any signal, and an internal reference corresponding to a transcribed region far from any AsiSI site (Ctrl) (primers and probe listed in Supplementary Table 9). In brief, genomic DNA was extracted using the DNeasy kit (QIAGEN) according to the manufacturer's instructions and diluted to 10 ng $\mu$l$^{-1}$. Then, 20 $\mu$l ddPCR reactions were composed of 50 ng of genomic DNA, 900 nM primers (DSB 526 Fw + Rev and Ctrl Fw + Rev), 250 nM probes (FAM-labelled for DSB 526, HEX-labelled for control region), 0.25 U $\mu$l$^{-1}$ HindIII (NEB) in 1× Supermix for Probes (no dUTP) (Bio-Rad). Droplets were generated in 70 $\mu$l Droplet generation Oil for Probes (Bio-Rad) with the QX200 Droplet Generator (Bio-Rad). Droplets were then transferred to a 96-well ddPCR plate (Bio-Rad), heat sealed with Pierceable Foil (180 °C for 5 s) and run on a T100 Thermal Cycler (Bio-Rad) with the following programme: 95 °C for 10 min; 40 cycles of 94 °C 30 s and 60 °C for 1 min (2 °C change per second); 98 °C for 10 min; 25 °C for 15 min. The PCR results for ~10,000–20,000 droplets per reaction were analysed using the QX200 Droplet Reader System and QuantaSoft software v.1.7.4 (Bio-Rad). Absolute quantifications in number of copies per $\mu$l for the −DSB and +DSB samples were provided by QuantaSoft Poisson function application and used to determine the cleavage efficiency for DSB 526 in the DSB-induced DIvA population.

## Western blot

Whole cell extracts were prepared and loaded on a NuPAGE 4–12% Bis-Tris Gel or 3–8% Tris-Acetate (Invitrogen) before proteins were transferred onto PVDF membranes (Invitrogen). Overnight incubation primary antibodies was performed before washing and secondary HRP-coupled secondary antibodies addition (Supplementary Table 4). Signals were revealed by chemiluminescence (Super Signal West Dura Extended Duration Substrate, Thermo Scientific) and acquired using a ChemiDoc Touch Imaging System (Bio-Rad).

## Immunofluorescence

Cells were plated into glass coverslips the day before the experiment. Following treatments, cells were fixed with 4% paraformaldehyde for 15 min and permeabilized with 0.5% Triton X-100 for 10 min at room temperature. Coverslips were blocked with 1× PBS–BSA 3% (Sigma) for 30 min and incubated overnight at 4 °C with the appropriate primary antibody (Supplementary Table 4). Cells were washed in 1× PBS–BSA 3% and incubated with secondary antibodies for 1 h at room temperature. Cells were washed in 1× PBS–BSA 3% and incubated for 15 min

with Hoechst 33342 (Sigma). Cells were washed again two times in 1× PBS–BSA 3% (or with 1× PBS–Tween 0.1% for BRCA1 and RAD51 immunofluorescence) before being mounted onto coverslips for image acquisition using MetaMorph on a DM6000 wide-field microscope equipped with a cooled charge-coupled device camera (CoolSNAP HQ2), using a ×40 objective. The number of foci was determined using CellProfiler (https://cellprofiler.org/).

## Statistics and reproducibility

Quantifications and statistical analyses (paired two-sided *t*-tests or Wilcoxon tests) were performed using GraphPad Prism v.10.3.1, the rstatix package in R v.4.2 or Microsoft Excel. Parametric (*t*-test) and nonparametric (Wilcoxon test) tests were applied as a function of normal and non-normal data distribution assumptions that were not formally tested. Equal variance was not assumed. The experiments were not randomized. No statistical method was used to predetermine sample size but our sample sizes are similar to those reported in previous publications[22,26,30]. The number of biological replicates, the sample size and the type of statistical tests used are mentioned in the Figure legends or numerical source data. The experiments were not randomized. The investigators were not blinded to allocation during the experiments and outcome assessment. No data were excluded from the analyses.

## Reporting summary

Further information on research design is available in the Nature Portfolio Reporting Summary linked to this article.

## Data availability

All high-throughput sequencing data (ChIP-seq, qDRIP-seq, END-seq, TT$_{chem}$-seq, DRB/TT$_{chem}$-seq and RNA-seq) have been deposited to Array Express (https://www.ebi.ac.uk/arrayexpress/) under accession number E-MTAB-13197. The hg19 reference genome was obtained from the University of California, Santa Cruz (https://hgdownload.soe.ucsc.edu/goldenPath/hg19/bigZips/). The mass spectrometry proteomics data are accessible on the ProteomeXchange Consortium via the PRIDE[74] partner repository with the dataset identifier PXD062315. Numerical source data are available as an Excel file published alongside with the paper. Source data are provided with this paper.

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

## Acknowledgements

We thank the genomics core facility of EMBL for high-throughput sequencing. We thank O. Schiltz and A. Stella from the proteomic facility at Institut de Pharmacologie et de Biologie Structurale (IPBS) in Toulouse. We thank N. Firmin for technical help. Funding

in the laboratory of G.L. is provided by grants from the European Research Council (ERC-AdG-101019963), the Agence Nationale pour la Recherche (ANR-18-CE12-0015), the Association Contre le Cancer, the Institut Thématique Multi-Organismes (ITMO) Cancer (no. 20CN067-00), the Ligue Nationale Contre le Cancer comité de l'Aude and the Fondation Bettencourt-Schueller. A.M. was a recipient of the Association Contre le Cancer PJA1. E.A. is a recipient of a Fondation pour la Recherche Médicale (FRM) fellowship (SPF202309017488). S.C. was a recipient of a PhD fellowship from the Joint Training and Research Programme on Chromatin Dynamics & the DNA Damage Response (H2020 ITN aDDRess, grant no. 812829). N.P. and T.C. are INSERM researchers.

## Author contributions

E.L., F.S., L.P., A.-L.F., E.A., B.L.B., N.P., M.C., M.P., T.C. and A.M. performed and analysed experiments. V.R., S.C. and A.M. performed bioinformatic analyses of all high-throughput sequencing datasets. G.L. wrote the paper with the help of A.M. and T.C. All authors commented and edited the paper. A.-L.F., S.C. and L.P. contributed equally to this work.

## Competing interests

The authors declare no competing interests.

## Additional information

**Extended data** is available for this paper at https://doi.org/10.1038/s41556-025-01669-y.

**Correspondence and requests for materials** should be addressed to Thomas Clouaire, Aline Marnef or Gaëlle Legube.

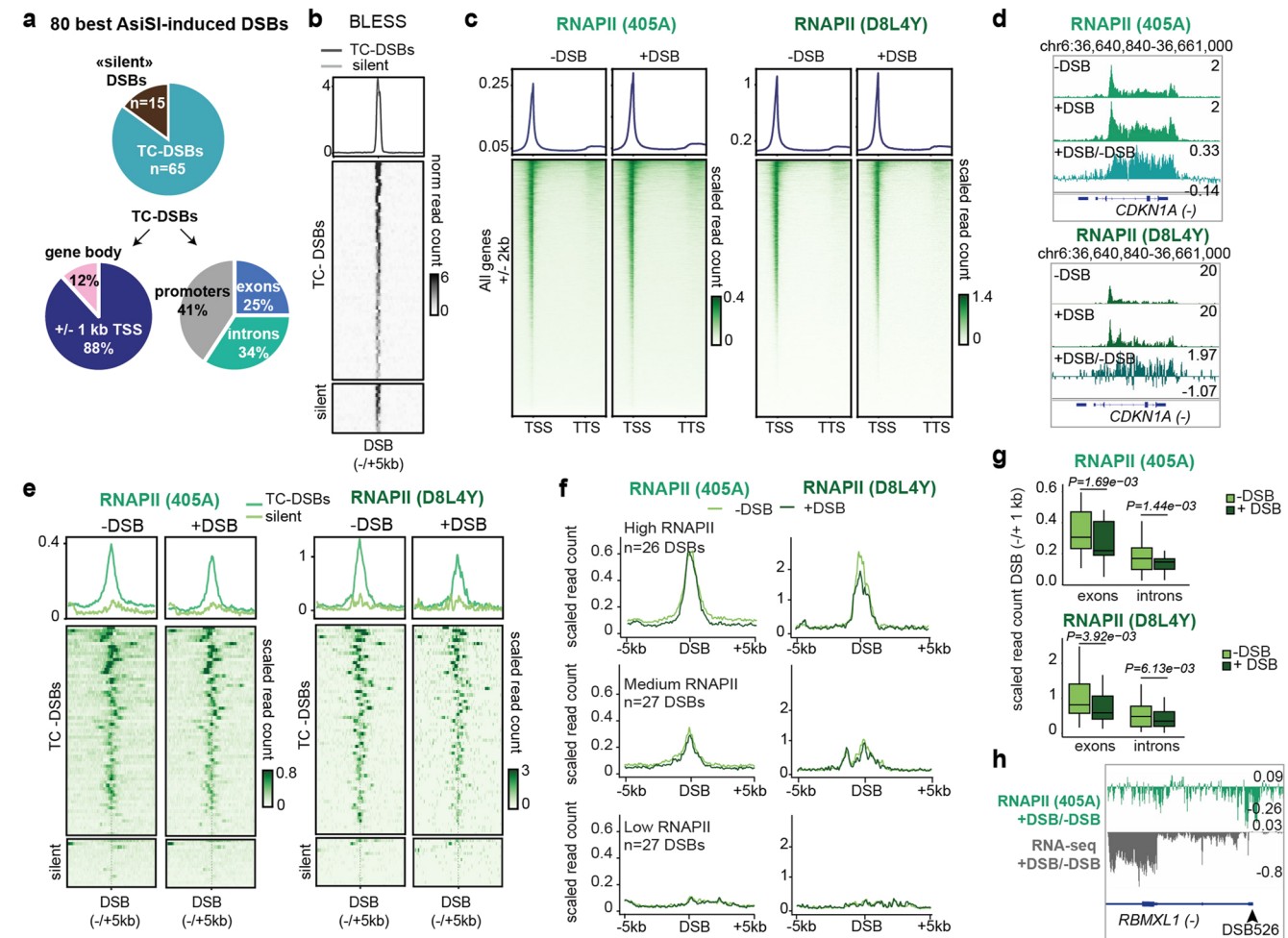

**Extended Data Fig. 1 | RNAPII levels at DSBs and control loci. (a)** Distribution of 80 best AsiSI-induced DSBs. There are 15 'silent' DSBs (which comprise intergenic DSBs and DSBs falling within transcriptionally silent genes) and 65 TC-DSBs (top panel). Most TC-DSBs fall within 1 kb of the TSS (88%, bottom left panel) and are distributed across promoters, exons and introns (respectively 41, 25 and 34%, bottom right panel). **(b)** Average profile and heatmap of BLESS signal after break induction on −/+ 5 kb around silent and TC-DSBs. **(c)** Average profile and heatmap of RNAPII ChIP-seq signal (using 405 A and D8L4Y antibodies) on all genes −/+ 2 kb (scaled read count). **(d)** Browser tracks (hg19) of RNAPII ChIP-seq (using 405 A and D8L4Y antibodies) before (-DSB) and after DSB induction (+DSB) and Log2 fold change ratio (+DSB/-DSB) at the *CDKN1A* (encoding p21) DDR gene.

**(e)** Average profile and heatmap of RNAPII ChIP-seq signals (using 405 A and D8L4Y antibodies) on −/+ 5 kb around silent and TC-DSBs (scaled read count). **(f)** Average profile and heatmap of RNAPII ChIP-seq signals (using 405 A and D8L4Y antibodies) on −/+ 5 kb around the 80 best DSBs classified according to RNAPII level before breakage (high, medium, low). **(g)** Quantification of RNAPII ChIP-seq signals on a −/+ 1 kb window around DSBs in exons ($n = 19$) or introns ($n = 26$). Centre line: median; Box limits: 1st and 3rd quartiles; Whiskers: Maximum and minimum without outliers; Points: outliers. P values, paired two-sided nonparametric Wilcoxon tests. **(h)** Genomic tracks (hg19) of RNAPII ChIP-seq and RNA-seq Log2 fold change ratio +DSB/-DSB at a TC-DSB in the *RBMXL1* gene (DSB 526). Source numerical data are available in source data.

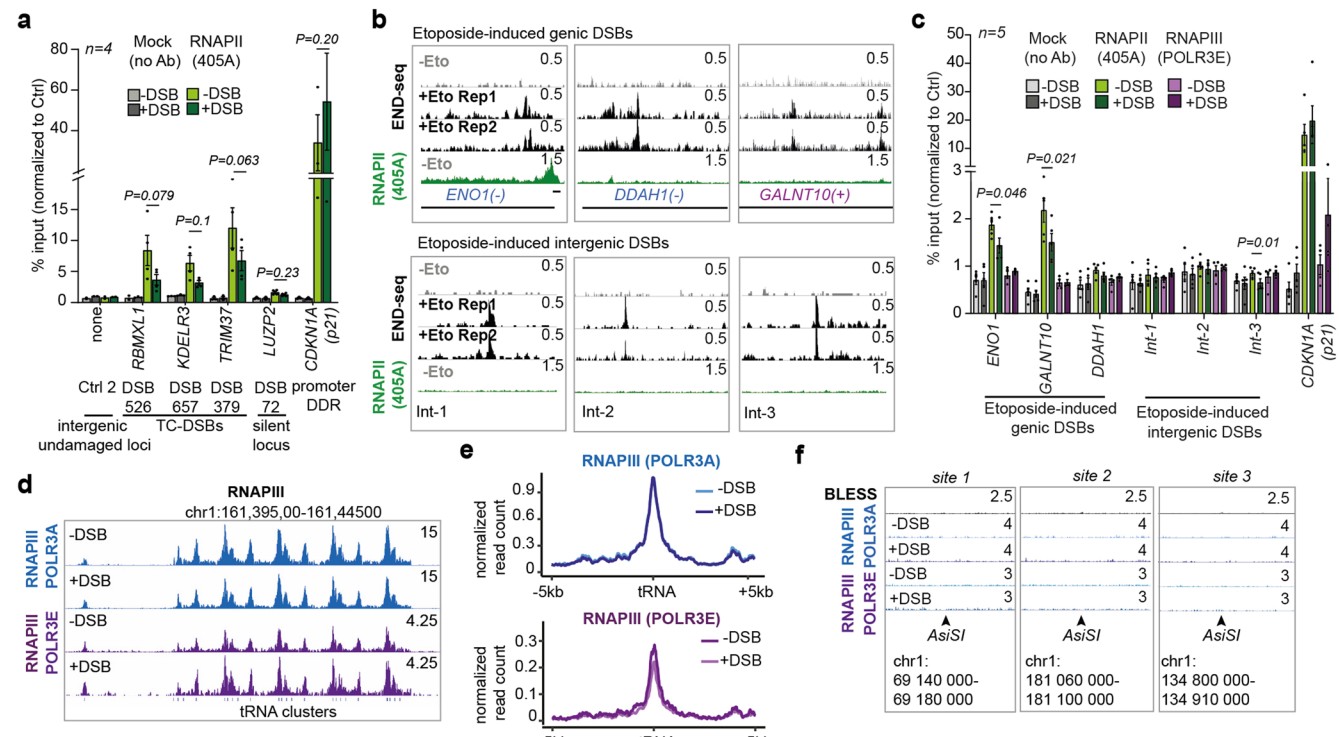

**Extended Data Fig. 2 | RNAPII and RNAPIII levels at DSBs and control loci.**
(**a**) ChIP–qPCR efficiency (% input normalized to an undamaged control locus) of Mock (no antibody) or RNAPII before and after DSB induction at an intergenic undamaged locus (Ctrl), three TC-DSBs (DSB 526, 657 and 379), a DSB in a silent locus (DSB 72) and the promoter of the DDR gene *CDKN1A* gene (encoding p21) as a control. Mean and s.e.m. for *n* = 4 biological replicates are shown. P values, paired two-sided t-tests. (**b**) Genomic tracks (hg19) of END-seq before (-Eto) and after Etoposide (+Eto, two replicates Rep1 and Rep2) and RNAPII ChIP-seq (405 A) in the absence of Etoposide (-Eto). Three examples of etoposide-induced genic DSBs (top panels) and etoposide-induced intergenic DSBs (bottom panel) are shown. (**c**) ChIP–qPCR efficiency (% input normalized to the undamaged control locus Ctrl2) of Mock (no antibody), RNAPII and RNAPIII ChIP–qPCR

before (-Eto) or after Etoposide treatment (+Eto) at the promoter of *CDKN1A* (p21) or at six etoposide-induced DSBs (3 genic and 3 intergenic are shown). Mean and s.e.m. for *n* = 5 (RNAPII) or *n* = 4 (RNAPIII) biological replicates are shown. P values, paired two-sided t-tests. Only significant P values between − and + DSB are indicated. (**d**) Genomic tracks (hg19) of POLR3A and POLR3E ChIP-seq before and after DSB induction at a tRNA cluster on chromosome 1. (**e**) Average profiles of POLR3A and POLR3E ChIP-seq centred on tRNA genes before (-DSB) and after DSB induction (+DSB). (**f**) Genomic tracks (hg19) of BLESS, POLR3A and POLR3E ChIP-seq in -DSB and +DSB conditions at sites shown in ref. 16 (NB: these three AsiSI sites did not display cleavage in DIvA cells, see BLESS track). Source numerical data are available in source data.

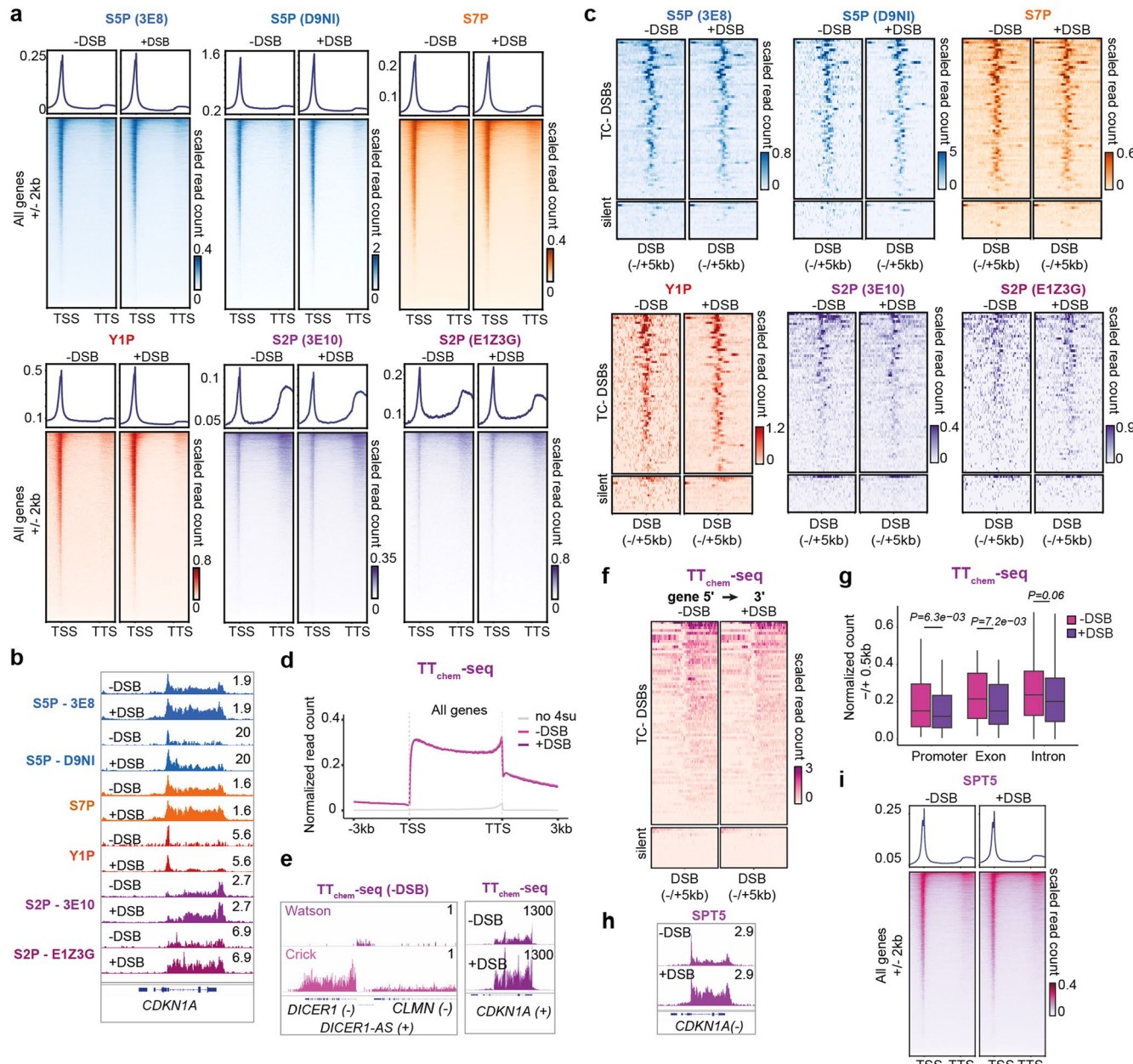

**Extended Data Fig. 3 | Levels of RNAPII CTD phosphorylation, nascent transcription and SPT5 at DSBs and control loci.** (**a**) Metagene profile and heatmaps of Y1P, S7P, S5P and S2P RNAPII CTD phosphorylation before DSB induction on all genes (−/+ 2 kb) using scaled read counts. (**b**) Genomic tracks (hg19) of Y1P, S7P, S5P and S2P RNAPII ChIP-seq before and after DSB induction on the DDR gene *CDKN1A* (encoding p21) using scaled read counts. (**c**) Heatmap of RNAPII CTD phosphorylation before DSB induction at silent and TC-DSBs. (**d**) Metagene profile of the TT$_{chem}$-seq signal without 4SU incorporation and with 4SU incorporation in - and + DSB conditions. (**e**) Left panel: Genomic tracks (hg19) of stranded TT$_{chem}$-seq in the absence of damage showing nascent RNA

detection at the lowly expressed *DICER-AS* gene. Right panel: total TT$_{chem}$-seq on *CDKN1A* in -DSB and +DSB conditions. (**f**) Heatmaps of TT$_{chem}$-seq at silent and TC-DSBs orientated according to gene directionality. (**g**) Quantification of the TT$_{chem}$-seq signal on a −/+ 500 bp window around TC-DSBs falling in a promoter (*n* = 31), exon (*n* = 19) or intron (*n* = 26) (Centre line: median; Box limits: 1st and 3rd quartiles; Whiskers: Maximum and minimum without outliers; Points: outliers. P values, paired two-sided nonparametric Wilcoxon tests). (**h**) Same as in (**e**) for SPT5 ChIP-seq. (**i**) Same as in (**a**) but for SPT5 ChIP-seq. Source numerical data are available in source data.

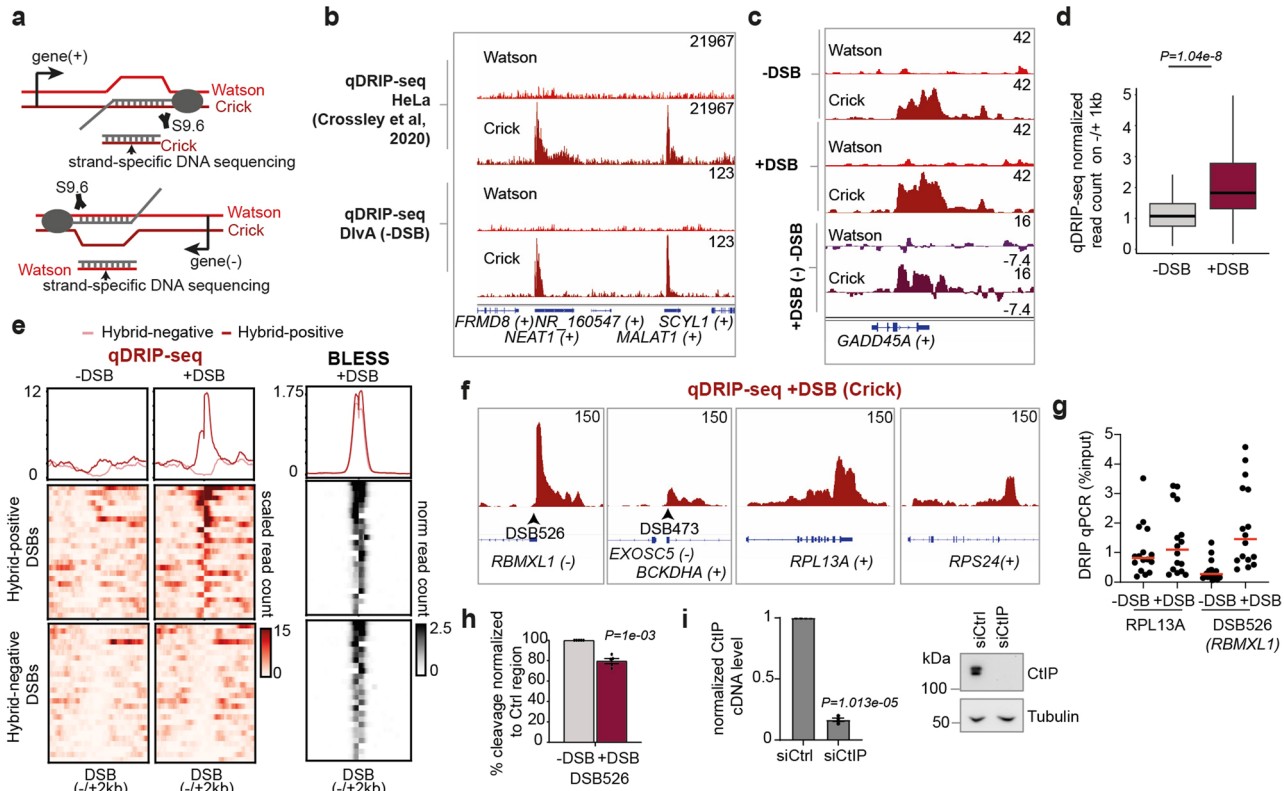

**Extended Data Fig. 4 | DSB-induced RNA:DNA hybrid mapping by qDRIP-seq.**
(**a**) Schematic representation of strand-specific qDRIP-seq. For (+) orientated genes, the resulting R-loops are detected on the Crick strand and for (−) orientated genes, R-loops are detected on the Watson strand. (**b**) Genomic track (hg19) of qDRIP-seq signal on the Watson and Crick strands for the (+) orientated *NEAT1* and *MALAT1* genes in a previously published qDRIP-seq obtained in HeLa cells (ref. 40) and in DIvA cells in the absence of DSB. (**c**) Genomic track (hg19) of strand-specific qDRIP-seq signals (Watson and Crick strands) at *GADD45A*. The differential enrichment of the qDRIP-seq signal on the Crick strand following DSB induction (+DSB (−) -DSB) is also shown. (**d**) Box plot of the qDRIP-seq signal on −/+ 1 kb around DSBs (*n* = 80) before (-DSB) and after DSB induction (+DSB) (Centre line: median; Box limits: 1st and 3rd quartiles; Whiskers: Maximum and minimum without outliers; Points: outliers; P values paired two-sided nonparametric Wilcoxon tests). (**e**) Average profiles and heatmaps of qDRIP-seq classified according to the presence (hybrid-positive) or absence of hybrids

(hybrid-negative) at DSBs. BLESS signal is also shown for these two categories. (**f**) Genomic track (hg19) of qDRIP-seq at two hybrid-positive TC-DSBs and at two genes (*RPL13A* and *RPS24*) known to accumulate R-loops and whose exact levels have been previously measured by SMRF-seq (ref. 41). (**g**) DRIP−qPCR at the R-loop positive *RPL13A* gene and at the TC-DSB induced in *RBMXL1* (DSB 526). The median value of *n* = 16 biologically independent experiments is shown (red). (**h**) Percentage of cleavage (normalized to an undamaged control locus) of DSB 526 as measured using ddPCR. (Mean and s.e.m. of *n* = 4 biological replicates are shown. P-value, paired two-sided t-test). (**i**) Left panel: RT−qPCR quantifying CtIP cDNA levels in control (Ctrl) or CtIP siRNA-transfected cells. Mean and s.e.m. for *n* = 4 biological replicates are shown. P-value, paired two-sided t-test. Right panel: Western blot against CtIP and Tubulin in siCtrl- and siCtIP-transfected cells. A representative experiment is shown (out of *n* = 2). Source numerical data and unprocessed blots are available in source data.

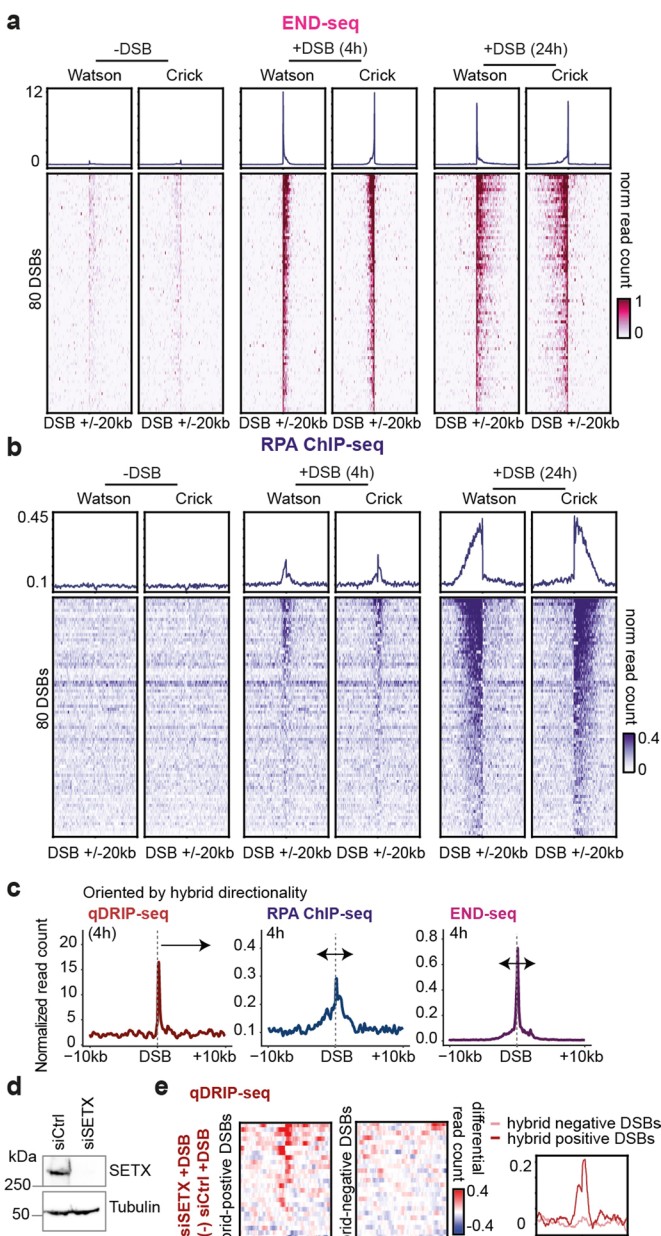

**Extended Data Fig. 5 | RNA:DNA hybrids and resection. (a, b)** Heatmaps showing strand-specific signals for END-seq (**a**) and RPA ChIP-seq (**b**) on the 80 best-cleaved DSBs in the absence of DSB and after 4 h or 24 h of DSB induction. Heatmaps are sorted by decreasing END-seq or RPA ChIP-seq. (**c**) Average profiles of END-seq, RPA ChIP-seq and qDRIP-seq (both strands combined) oriented by RNA:DNA hybrid directionality on a −/+ 10 kb window around the 80 best-cleaved DSBs. The arrows show the extent of resection as measured by END-seq and RPA ChIP-seq. (**d**) Level of SETX and Tubulin measured by Western blot in siCtrl and siSETX transfected cells. A representative experiment is shown (out of $n = 3$). (**e**) Average profiles and heatmaps of the differential enrichment of the qDRIP-seq signal after depletion of SETX (siSETX +DSB (-) siCtrl +DSB) on hybrid-positive and -negative DSBs. Unprocessed blots are available in source data.

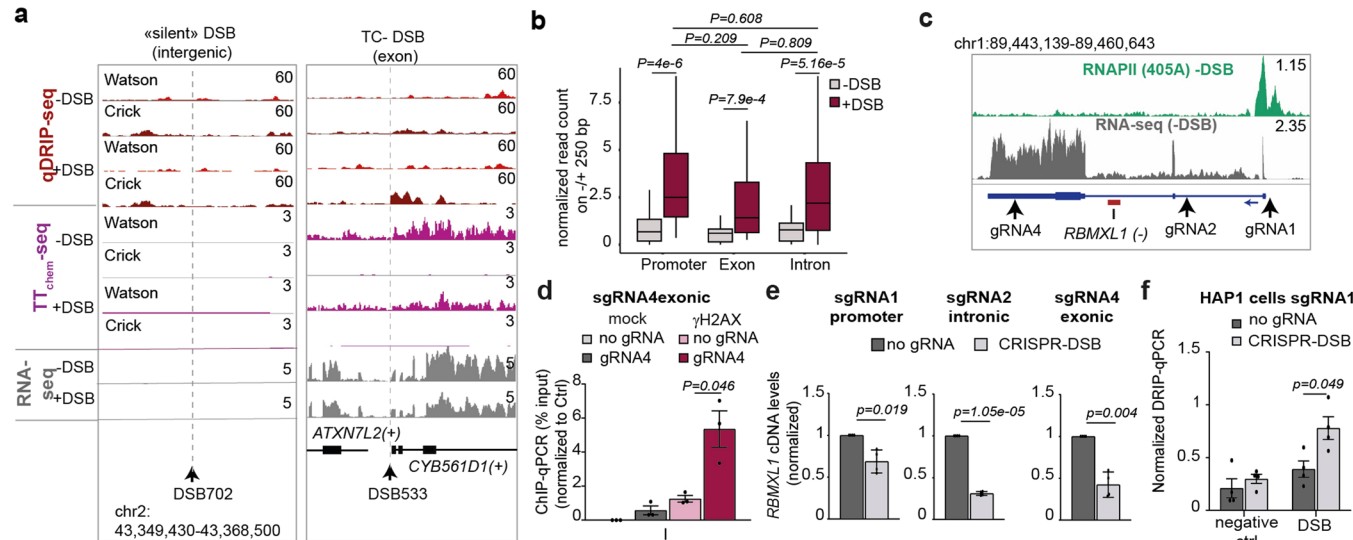

**Extended Data Fig. 6 | DSB-induced RNA:DNA hybrids accumulate on the template strand upon transcriptional repression.** (**a**) Genomic tracks (hg19) of qDRIP-seq on the Watson and Crick strands -DSB and +DSB at a TC-DSB located in an exon and in a silent DSB located in an intergenic region. The strand-specific TT$_{chem}$-seq signal is also shown as well as the RNA-seq signal in -DSB and +DSB conditions. (**b**) Quantification of the qDRIP-seq signal on −/+ 250 bp around TC-DSBs located in promoter ($n = 31$), exon ($n = 19$) or intron ($n = 26$) (Centre line: median; Box limits: 1st and 3rd quartiles; Whiskers: Maximum and minimum without outliers; Points: outliers. P values, paired two-sided nonparametric Wilcoxon tests). (**c**) Genomic tracks (hg19) of RNAPII ChIP-seq (405 A) and RNA-seq on *RBMXL1* showing the position of sgRNA1, 2 and 4 and of the primers

(I) used in (**d**). (**d**) ChIP−qPCR (% input normalized to the undamaged control locus Ctrl2) of γH2AX before (no gRNA) or after CRISPR-generated DSBs with sgRNA4 (mean and s.e.m. for $n = 3$ biological replicates are shown. P-value, paired two-sided t-test). (**e**) RT−qPCR quantifying *RBMXL1* cDNA levels normalized to *TBP* upon no gRNA or CRISPR-generated DSBs with sgRNA1, 2 or 4. Mean and s.e.m. for $n = 4$ biological replicates are shown. P values, paired two-sided t-tests. (**f**) DRIP−qPCR detection of RNA:DNA hybrids around DSB induced with sgRNA1 in HAP1 cells normalized to the *FOS* positive control (mean and s.e.m. for $n = 4$ biological replicates are shown. P-value, paired two-sided t-test). Source numerical data are available in source data.

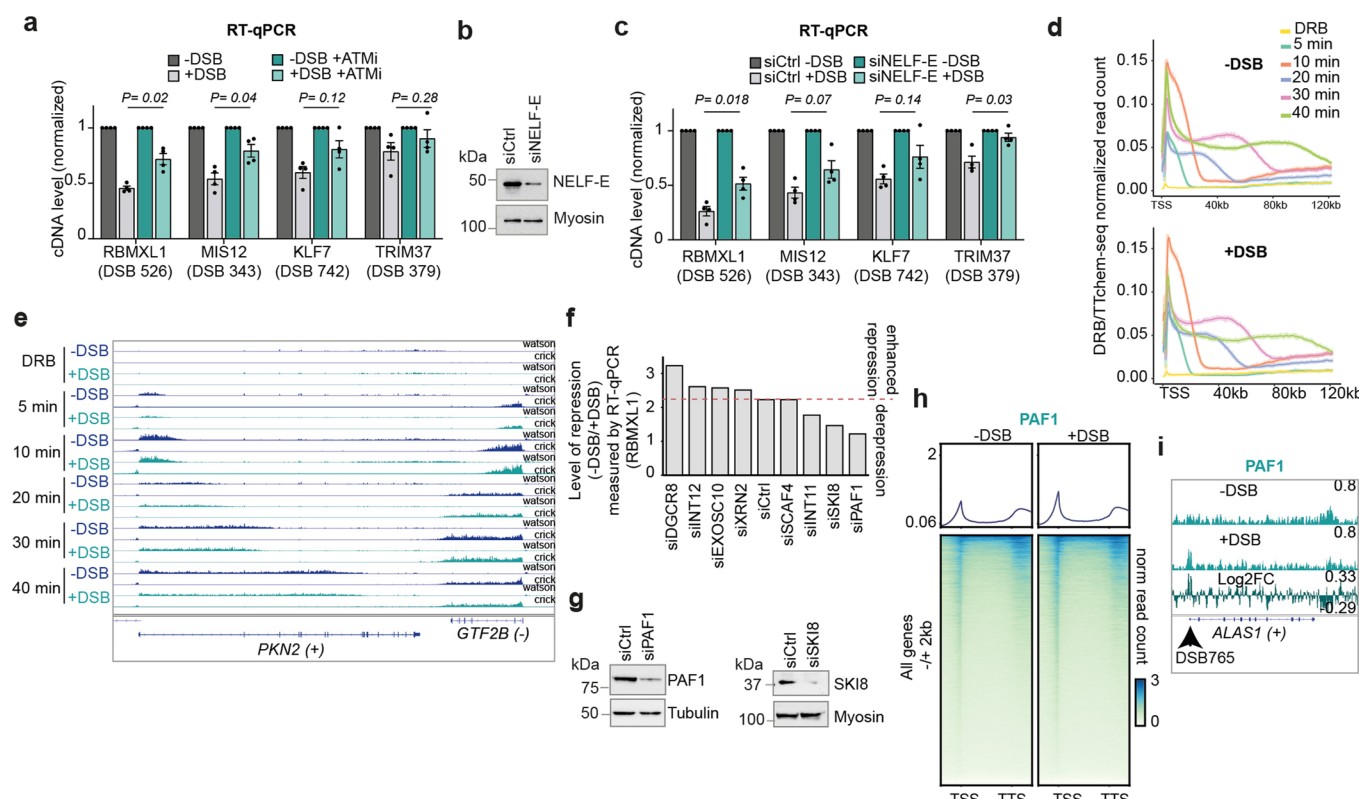

**Extended Data Fig. 7 | ATM, NELF-E, the PAF complex participate in DSB-induced transcriptional repression in *cis*.** (**a**) RT–qPCR quantifying cDNA levels normalized to RPLP0 at 4 damaged genes (RBMXL1, MIS12, KLF7 and TRIM37) before and after DSB upon ATMi treatment. Mean and s.e.m. for *n* = 4 biological replicates are shown. P values are calculated with paired two-sided t-tests. (**b**) Levels of NELF-E and Myosin measured by Western blot after depletion of NELF-E. A representative experiment is shown (out of *n* = 2) (**c**) same as (**a**) for siRNA NELF-E-transfected cells. (**d**) Waves of elongation on all genes measured with DRB/TTchem-seq in -DSB and +DSB conditions. (**e**) Genomic track (hg19) example of a (+) orientated long gene (*PKN2*) and a (-) orientated short gene (*GTF2B*) showing waves of elongation measured by DRB/TTchem-seq on both

Watson and Crick strands in -DSB and +DSB conditions. (**f**) Level of repression measured by RT–qPCR for *RBMXL1* following DSB induction upon transfection of cells with different siRNAs (indicated on the y axis). The red dotted line represents the baseline of repression detected with siCtrl. (**g**) Levels of PAF1 and Tubulin (left panel) and SKI8 and Myosin (right panel) measured by Western blot after depletion of either PAF1 or SKI8 (*n* = 1). (**h**) Average profile and heatmap of PAF1 ChIP-seq signal on all genes −/+ 2 kb (normalized read count). (**i**) Genomic tracks (hg19) of PAF1 ChIP-seq before and after DSB induction as well as the Log2 fold change ratio (+DSB/-DSB) at a TC-DSB (DSB 765). Source numerical data and unprocessed blots are available in source data.

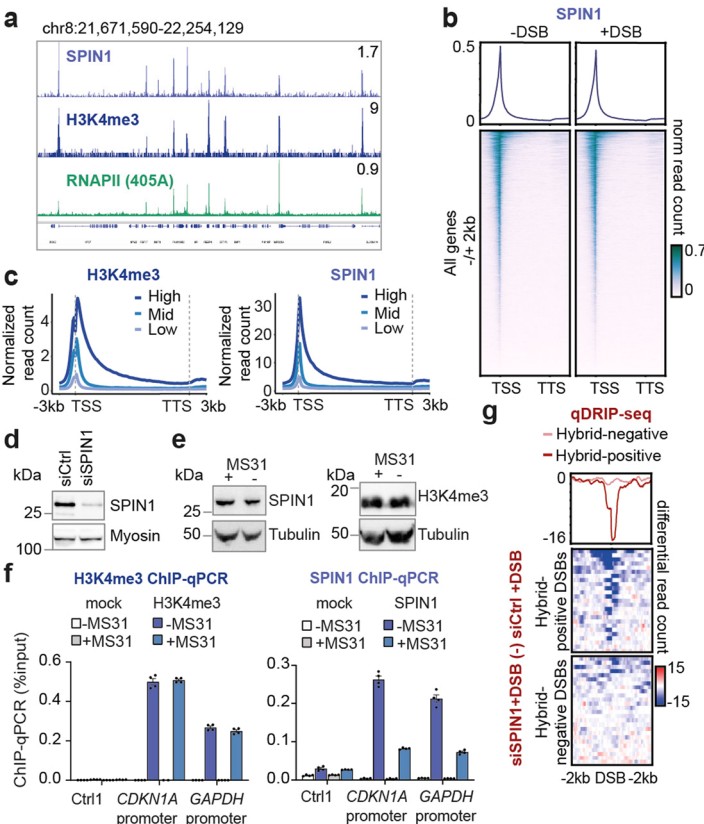

**Extended Data Fig. 8 | SPIN1 participates in DSB-induced transcriptional repression in cis. (a)** Genomic tracks (hg19) of SPIN1, H3K4me3 and RNAPII (405 A) ChIP-seq as well as RNA-seq before DSB induction showing the similarities between the signals detected. **(b)** Average profile and heatmap of SPIN1 ChIP-seq signal on all genes −/+ 2 kb (normalized read count). **(c)** Metagene profiles of ChIP-seq signals for H3K4me3 (left panel) and SPIN1 (right panel) on genes classified into three classes according to their RNAPII levels (high, mid or low) before DSB induction. **(d)** Levels of SPIN1 and Myosin detected by Western blot after siRNA-mediated depletion of SPIN1. A representative experiment is shown

(out of $n = 4$). **(e)** Levels of SPIN1, H3K4me3 and Tubulin detected by Western blot after treatment with MS31 ($n = 1$). **(f)** ChIP–qPCR efficiency (%input) of either H3K4me3 (left panel) or SPIN1 (right panel) at a control intergenic locus (Ctrl1) and at the promoter of *CDKN1A* and *GAPDH* in the presence or absence of MS31 treatment (mean and s.e.m. of technical replicates ($n = 4$)). **(g)** Average profiles and heatmaps of the differential enrichment of qDRIP-seq signal after depletion of SPIN1 (siSPIN1 +DSB (-) siCtrl +DSB) on hybrid-positive and -negative DSBs. Source numerical data and unprocessed blots are available in source data.

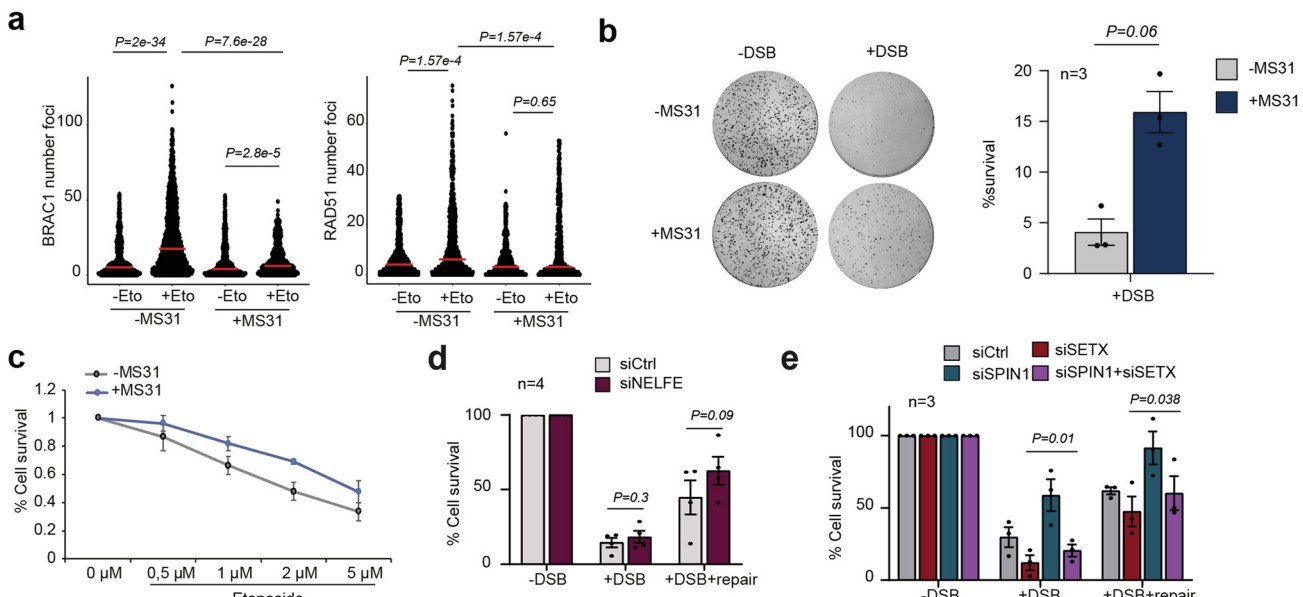

**Extended Data Fig. 9 | Factors participating in DSB-induced transcriptional repression participate in HR and cell survival.** (**a**) Number of BRCA1 or RAD51 foci in the presence or absence of Etoposide (Eto) and/or MS31. Red line: median. P values, paired two-sided nonparametric Wilcoxon tests. (**b**) Clonogenic assay in MS31-treated DIvA-AID cells after DSB induction (+DSB). A representative experiment is shown on the left panel. Mean and s.e.m. for *n* = 3 biological replicates are shown on the right panel. P values, paired two-sided t-test. (**c**) Percentage of cell survival upon treatment with different doses of Etoposide treatment (indicated on the x axis) and in the presence or absence of MS31. (**d**) Clonogenic assay in control (Ctrl) or NELF-E siRNA-treated DIvA-AID cells in -DSB and +DSB conditions and after IAA treatment allowing DSB repair (+DSB +repair). Mean and s.e.m. for *n* = 4 biological replicates are shown. P values, paired two-sided t-tests. (**e**) same as in (**d**) for control (Ctrl), SPIN1 or SPIN1 + SETX siRNA-treated DIvA-AID cells, *n* = 3 biological replicates. P values, paired two-sided t-test. Source numerical data are available in source data.

# Reporting Summary

## Statistics

For all statistical analyses, confirm that the following items are present in the figure legend, table legend, main text, or Methods section.

| n/a | Confirmed | |
|---|---|---|
| ☐ | ☒ | The exact sample size (*n*) for each experimental group/condition, given as a discrete number and unit of measurement |
| ☐ | ☒ | A statement on whether measurements were taken from distinct samples or whether the same sample was measured repeatedly |
| ☐ | ☒ | The statistical test(s) used AND whether they are one- or two-sided<br>*Only common tests should be described solely by name; describe more complex techniques in the Methods section.* |
| ☒ | ☐ | A description of all covariates tested |
| ☒ | ☐ | A description of any assumptions or corrections, such as tests of normality and adjustment for multiple comparisons |
| ☐ | ☒ | A full description of the statistical parameters including central tendency (e.g. means) or other basic estimates (e.g. regression coefficient) AND variation (e.g. standard deviation) or associated estimates of uncertainty (e.g. confidence intervals) |
| ☒ | ☐ | For null hypothesis testing, the test statistic (e.g. *F*, *t*, *r*) with confidence intervals, effect sizes, degrees of freedom and *P* value noted<br>*Give P values as exact values whenever suitable.* |
| ☒ | ☐ | For Bayesian analysis, information on the choice of priors and Markov chain Monte Carlo settings |
| ☒ | ☐ | For hierarchical and complex designs, identification of the appropriate level for tests and full reporting of outcomes |
| ☒ | ☐ | Estimates of effect sizes (e.g. Cohen's *d*, Pearson's *r*), indicating how they were calculated |

*Our web collection on statistics for biologists contains articles on many of the points above.*

## Software and code

Policy information about availability of computer code

| Data collection | Bio-Rad CFX Manager version 3.1<br>ChemiDoc™ Touch Imaging System.and Image Lab Touch version 5.2.1<br>MetaMorph version 7.1.0.0 |
|---|---|
| Data analysis | Cell Profiler 4.2.8<br>Icy software version 2.0.2.0 (https://icy.bioimageanalysis.org/ software )<br>Bio-Rad CFX Manager version 3.1<br>QuantaSoft version 1.74.4 (Bio-Rad)<br>bwa (https://bio-bwa.sourceforge.net/) version 0.7.12-r1039<br>samtools (http://www.htslib.org/) version 1.9<br>bamCoverage from deepTools (https://deeptools.readthedocs.io/en/develop/)<br>R 4.2<br>GenomicRanges 1.38<br>plyranges 1.6.10<br>Integrated Genome Browser version 9.1.6 (https://www.bioviz.org)<br>Integrated Genome Viewer version 2.19.1 (https://igv.org)<br>deeptools 3.4.3<br>rtracklayer 1.58.0<br>stringr 1.5.0<br>dplyr 1.0.10 |

ggplot2 3.4.0
rstatix 0.7.1

For manuscripts utilizing custom algorithms or software that are central to the research but not yet described in published literature, software must be made available to editors and reviewers. We strongly encourage code deposition in a community repository (e.g. GitHub). See the Nature Portfolio guidelines for submitting code & software for further information.

## Data

Policy information about availability of data

All manuscripts must include a data availability statement. This statement should provide the following information, where applicable:
- Accession codes, unique identifiers, or web links for publicly available datasets
- A description of any restrictions on data availability
- For clinical datasets or third party data, please ensure that the statement adheres to our policy

All high-throughput sequencing data (ChIP-seq, qDRIP-seq, END-seq, TTchem-seq, DRB/TTchem-seq and RNA-seq) have been deposited to Array Express (https://www.ebi.ac.uk/arrayexpress/) under accession number E-MTAB-13197.
The hg19 reference genome was obtained from UCSC (https://hgdownload.soe.ucsc.edu/goldenPath/hg19/bigZips/).

## Human research participants

Policy information about studies involving human research participants and Sex and Gender in Research.

| | |
|---|---|
| Reporting on sex and gender | Not relevant to our study. |
| Population characteristics | NA |
| Recruitment | NA |
| Ethics oversight | NA |

Note that full information on the approval of the study protocol must also be provided in the manuscript.

# Field-specific reporting

Please select the one below that is the best fit for your research. If you are not sure, read the appropriate sections before making your selection.

☒ Life sciences ☐ Behavioural & social sciences ☐ Ecological, evolutionary & environmental sciences

For a reference copy of the document with all sections, see nature.com/documents/nr-reporting-summary-flat.pdf

# Life sciences study design

All studies must disclose on these points even when the disclosure is negative.

| | |
|---|---|
| Sample size | No sample size calculation was performed. The number of samples in each experiment was determined based on standards practice in the field (see reference 35, 39 and 43 for exemples) in most cases in triplicates or above. The number of independent experiments are indicated in the legend of each Figure. |
| Data exclusions | No data were excluded from analysis. |
| Replication | - ChIP-seq before and after DSB induction: n=1. However, sequenced DNA comes from a pool of independent ChIP: n=10 for SPIN1, n=12 for PAF1, n=1 for SPT5, n=4 for RNAPII total (405), n=3 for RNAPII total (D8L4Y); n=6 for Y1P, n=4 for S2P (3E10), n=3 for S2P (E1Z3G), n=4 for S5P (3E8), n=3 for S5P (D9NI), n=4 for S7P, n=5 for POLR3A, n=5 for POLR3E. ChIP-seq before and after 4h and 24h DSB induction: n=1 (but sequenced DNA from a pool of independent ChIP: n=9 RPA)
- qDRIP-seq before and after 4h and 24h DSB induction: n=1 (but sequenced DNA from a pool of 3 independent qDRIP). qDRIP-seq before and after DSB induction in siCtrl and siSPIN1 transfected cells: n=1 (but sequenced DNA from a pool of 3 independent qDRIP).
-TTchem-seq or DRB/TTchem-seq: n=1
- DRIP-qPCR: n≥3
- END-seq before and 4h and 24h after DSB induction: n=1. END-seq before and after DSB induction in siRNA transfected cells: n=1
- RT-qPCR in siRNA transfected cells and MS31-treated cells: n≥3
- Resection assay: n≥3
- All ChIP-qPCR n≥3 apart from that after DSB induction with MS31-treated cells: n=1.
- Clonogenic assay before and after DSB induction with siRNA transfected cells: n≥3. Clonogenic assay before and after DSB induction with MS31-treated and MS31-/Etoposide-treated cells and Mirin-treated cells: n≥3.
- Western blots: representative experiments are shown for siRNA-mediated depletion efficiency out of n=2 for CtIP, n=3 for SETX, n=2 for NELF-E, n=4 for SPIN1. For siRNA-mediated depletion efficiency of PAF1 and SKI8 and for SPIN1 and H3K4me3 levels upon MS31 treatment n=1. |

- Fluorescence microscopy: n=1 analysis on more than 100 cells per conditions
- Illegitimate re-joining: frequency n=4.

**Randomization**   Randomization is not relevant because we did not use different experimental groups in our study.

**Blinding**   Blinding was not relevant to our study since we did not have experimental group to compare.

# Reporting for specific materials, systems and methods

We require information from authors about some types of materials, experimental systems and methods used in many studies. Here, indicate whether each material, system or method listed is relevant to your study. If you are not sure if a list item applies to your research, read the appropriate section before selecting a response.

## Materials & experimental systems

| n/a | Involved in the study |
|---|---|
| ☐ | ☒ Antibodies |
| ☐ | ☒ Eukaryotic cell lines |
| ☒ | ☐ Palaeontology and archaeology |
| ☒ | ☐ Animals and other organisms |
| ☒ | ☐ Clinical data |
| ☒ | ☐ Dual use research of concern |

## Methods

| n/a | Involved in the study |
|---|---|
| ☐ | ☒ ChIP-seq |
| ☒ | ☐ Flow cytometry |
| ☒ | ☐ MRI-based neuroimaging |

## Antibodies

**Antibodies used**

RNAPII (405A) Bethyl A304-405A 2 µL/200 µg of DNA ChIP
RNAPII (D8L4Y)  Cell Signaling 14958  7 µL/ 100 µg of DNA ChIP
POLR3A Abcam ab96328 6 µL/150 µg of DNA ChIP
POLR3E Bethyl A303-707A 5 µL/150 µg of DNA ChIP
Y1P Active Motif 61383 3 µL/200 µg of DNA ChIP
S2P (3E10) Chromotek 3E10 75 µL/150 µg of DNA ChIP
S2P (E1Z3G) Cell Signaling 13499  7 µL/ 100 µg of DNA ChIP
S5P (3E8) Chromotek 3E8 75 µL/150 µg of DNA ChIP
S5P (D9N5I) Cell Signaling 13523  7 µL/ 100 µg of DNA ChIP
S7P Chromotek 4E12 75 µL/150 µg of DNA ChIP
S9.6 Antibodies Incorporated 6 µg/4 µg of DNA DRIP
SPT5 Bethyl A3030-707A 2.5 µL/100 µg of DNA ChIP
PAF1 Abcam ab137519 2 µL/150 µg of DNA ChIP
  1:1000 WB
SKI8 (WDR61) Invitrogen PA540079 1:500 WB
RPA Abcam ab10359 4 µg/200 µg ChIP
gH2AX Abcam ab176458  1 µL/200 µg of DNA ChIP
RAD51 Abcam ab176458 2 µL/200 µg of DNA ChIP
RAD51 sc8349 1:100 IF
H3K4me3 Abcam ab8580 1 µL/25 µg of DNA ChIP
  1:1000 WB
SPIN1 Proteintech, 12105-1-AP 7 µL/200 µg of DNA ChIP
  1:1000 WB
NELF-E Abcam Ab170104 1:1000 WB
Myosin Sigma, M3567 1:2000 WB
Tubulin Sigma T6199 1:10000 WB
Anti-rabbit-HRP  Sigma A0545 1:10000 WB
Anti-mouse-HRP Sigma, A2554 1:10000 WB
gH2AX Millipore JBW301 1:1000 IF
BRCA1 Calbiochem OP92-100UG 1:500 IF

**Validation**

SPT5 (Bethyl A300-869A) validate in ChIP (https://pubmed.ncbi.nlm.nih.gov/35325203/)
PAF1 (Abcam, ab137519) validated in ChIP (https://pubmed.ncbi.nlm.nih.gov/33852864/) and western blot (https://pubmed.ncbi.nlm.nih.gov/30367041/)
SKI8 (Invitrogen PA540079) validated in western blot (https://www.thermofisher.com/antibody/product/WDR61-Antibody-Polyclonal/PA5-40079)
SPIN1 (Proteintech, 12105-1-AP) validated in ChIP (https://pubmed.ncbi.nlm.nih.gov/36736887/) and in western blot (https://www.ptglab.com/fr/products/SPIN1-Antibody-12105-1-AP.htm)
RNAPII (Bethyl A304-405A) validated in ChIP (https://www.fortislife.com/products/primary-antibodies/rabbit-anti-rna-polymerase-ii-antibody/BETHYL-A304-405)
RNAPII (Cell signalling D8L4Y) validated in Chip (https://pubmed.ncbi.nlm.nih.gov/30472187/)
POLR3A (Abcam ab96328) validated in ChIP (https://pubmed.ncbi.nlm.nih.gov/35637192/)
POLR3E (Bethyl A3030-707A) validated in ChIP with our data
Y1P (Active Motif 61383) validated in ChIP (https://doi.org/10.1016/j.celrep.2016.05.010)

S2P (Chromotek 3E10) validated in ChIP (https://pubmed.ncbi.nlm.nih.gov/24478330/)
S2P (E1Z3G) validated in ChIP (https://pubmed.ncbi.nlm.nih.gov/28782042/)
S5P (Chromotek 3E8) validated in ChIP (https://pubmed.ncbi.nlm.nih.gov/24478330/)
S5P (D9NI) validated in ChIP (https://pubmed.ncbi.nlm.nih.gov/24478330/)
S7P (Chromotek 4E12) validated in ChIP (https://pubmed.ncbi.nlm.nih.gov/24478330/)
S9.6 (Antibodies Incorporated) validated in DRIP (https://academic.oup.com/nar/article/48/14/e84/5858111)
RPA32/2 (Abcam ab10359) validated in ChIP (https://genesdev.cshlp.org/content/35/19-20/1356.long)
NELF-E (Abcam ab170104) validated in Western Blot (https://www.abcam.com/products/primary-antibodies/nelfe-antibody-epr11600-ab170104.html#lb)
Myosin (Sigma, M3567) validated in Western Blot (https://www.sigmaaldrich.com/FR/fr/search/m3567?focus=products&page=1&perpage=30&sort=relevance&term=m3567&type=product)
H3K4me3 (Abcam ab8580) validated in ChIP and Western Blot (abcam.com/products/primary-antibodies/histone-h3-tri-methyl-k4-antibody-chip-grade-ab8580.html)
α-Tubulin (Sigma T6199) validated in Western Blot (https://www.sigmaaldrich.com/FR/fr/product/sigma/t6199)
Anti-rabbit-HRP (Sigma A0545) validated in Western Blot (https://www.sigmaaldrich.com/FR/fr/product/sigma/a0545)
Anti-mouse-HRP (Sigma, A2554) validated in Western Blot (https://www.sigmaaldrich.com/FR/fr/product/sigma/a2554)
gH2AX (Abcam ab81299) validated in ChIP (https://doi.org/10.1016/j.molcel.2018.08.020)
gH2AX (Millipore JBW301) validated in IF (https://www.sigmaaldrich.com/FR/fr/product/mm/05636)
RAD51 (Abcam ab176458) validated in ChIP (https://www.ncbi.nlm.nih.gov/pmc/articles/PMC6993210/)
RAD51 (Santacruz, sc8349) validated in IF (https://pubmed.ncbi.nlm.nih.gov/29416069/)

# Eukaryotic cell lines

Policy information about cell lines and Sex and Gender in Research

| | |
|---|---|
| Cell line source(s) | Cell lines developped from U2OS cells (ATCC® HTB-96™) in Gaelle Legube's laboratory (DIvA cell line and AID-DIvA cell line). Lenti-X HEK293T were purchased from Clontech (632180). HAP1 cells were purchased from Horizon (C859). |
| Authentication | Authentication of the U2OS cell line was performed by the provider ATCC which uses morphology, karyotyping and PCR based approaches to confirm the identity of human cell lines. DIvA and AID-DIvA , both derived from U2OS cells, Lenti-X HEK293T and HAP1 cells  were not further authenticated. |
| Mycoplasma contamination | All cell lines (DIvA, AID-DIvA, Lenti-X HEK293T and HAP1) were regularly tested for absence of mycoplasma contamination by using the TransDetect® PCR Mycoplasma Detection Kit (TransGen Biotech). All cell lines used in this study were tested negative for Mycoplasma. |
| Commonly misidentified lines (See ICLAC register) | No commonly misidentified cell lines were used in the study. Cell lines used in the study are not registered in ICLAC. |

# Methodology

**Replicates**

- ChIP-seq before and after DSB induction: n=1. However, sequenced DNA comes from a pool of independent ChIP: n=10 for SPIN1, n=12 for PAF1, n=1 for SPT5, n=4 for RNAPII total (405), n=3 for RNAPII total (D8L4Y); n=6 for Y1P, n=4 for S2P (3E10), n=3 for S2P (E1Z3G), n=4 for S5P (3E8), n=3 for S5P (D9NI), n=4 for S7P, n=5 for POLR3A, n=5 for POLR3E). ChIP-seq before and after 4h and 24h DSB induction: n=1 (but sequenced DNA from a pool of independent ChIP: n=9 RPA)
- qDRIP-seq before and after 4h and 24h DSB induction: n=1 (but sequenced DNA from a pool of 3 independent qDRIP). qDRIP-seq before and after DSB induction in siCtrl and siSPIN1 transfected cells: n=1 (but sequenced DNA from a pool of 3 independent qDRIP).
-TTchem-seq or DRB/TTchem-seq: n=1

**Sequencing depth**

RNAPOLII_Total_405_mOHT - 138 million single end reads (85nt)
RNAPOLII_Total_405_pOHT - 152 million single end reads (85nt)
RNAPOLII_Total_D8L4Y_mOHT - 48 million single end reads (122nt)
RNAPOLII_Total_D8L4Y_pOHT - 27 million single end reads (122nt)
POLR3A_mOHT - 17 million single end reads (85nt)
POLR3A_pOHT - 12.8 million single end reads (85nt)
POLR3E_mOHT - 22.3 million single end reads (85nt)
POLR3E_pOHT - 11.6 million single end reads (85nt)
RNAPOLII_Y1P_mOHT - 96.6 million single end reads (85nt)
RNAPOLII_Y1P_pOHT - 86.5 million single end reads (85nt)
RNAPOLII_S2_3E10_mOHT - 107 million single end reads (85nt)
RNAPOLII_S2_3E10_pOHT - 115 million single end reads (85nt)
RNAPOLII_S2_E1Z3G_mOHT - 84 million single end reads (122nt)
RNAPOLII_S2_E1Z3G_pOHT – 69 million single end reads (122nt)
RNAPOLII_S5_3E8_mOHT - 148 million single end reads (85nt)
RNAPOLII_S5_3E8_pOHT - 133 million single end reads (85nt)
RNAPOLII_S5_D9NI_mOHT - 65 million single end reads (122nt)
RNAPOLII_S5_D9NI_pOHT - 86 million single end reads (122nt)
RNAPOLII_S7_mOHT - 126 million single end reads (85nt)
RNAPOLII_S7_pOHT - 152 million single end reads (85nt)
Spin1_mOHT - 23.2 million paired end reads (80nt)
Spin1_pOHT - 31.3 million paired end reads (80nt)
PAF1_DIVA - 49 million single end reads (122 nt)
PAF1_OHT - 38 million single end reads (122 nt)
SPT5_DIVA - 129 million single end reads (132 nt)
SPT5_OHT - 143 million single end reads (132 nt)
RPA_mOHT - 86.2 million paired end reads (75nt)
RPA_pOHT_4H - 77.2 million paired end reads (75nt)
RPA_pOHT_24H - 70.2 million paired end reads (75nt)
qDRIP_DIVA - 83 million paired end reads (75nt)
qDRIP_OHT - 82 million paired end reads (75nt)
qDRIP_OHT_24h - 92 million paired end reads (75nt)
ENDseq_NT – 29 million reads paired end reads (61nt)
ENDseq_Etoposide_rep1 - 39 million reads paired end reads (61nt)
ENDseq_Etoposide_rep2 -56 million reads paired end reads (61nt)
ENDseq_siCTRL_DIVA - 65 million reads paired end reads (61nt)
ENDseq_siCTRL_OHT - 65 million reads paired end reads (61nt)
ENDseq_siSETX_DIVA -71 million reads paired end reads (61nt)
ENDseq_siSETX_OHT -71 million reads paired end reads (61nt)
TTchemseq_no4SU – 53 million reads single end reads (75nt)
TTchemseq_DIVA– 64 million reads single end reads (75nt)
TTchemseq_OHT – 66 million reads single end reads (75nt)
DRB_TTchemseq_DIVA_DRB – 62 million reads single end reads (75nt)
DRB_TTchemseq_DIVA _DRB_5min - 70 million reads single end reads (75nt)
DRB_TTchemseq_DIVA _DRB_10min – 75 million reads single end reads (75nt)
DRB_TTchemseq_DIVA _DRB_20min – 97 million reads single end reads (75nt)
DRB_TTchemseq_DIVA _DRB_30min – 73 million reads single end reads (75nt)
DRB_TTchemseq_DIVA _DRB_40min – 107 million reads single end reads (75nt)
DRB_TTchemseq_OHT_DRB —— 65 million reads single end reads (75nt)
DRB_TTchemseq_OHT_DRB_5min - 52 million reads single end reads (75nt)
DRB_TTchemseq_OHT _DRB_10min – 61 million reads single end reads (75nt)
DRB_TTchemseq_OHT _DRB_20min – 75 million reads single end reads (75nt)
DRB_TTchemseq_OHT _DRB_30min – 72 million reads single end reads (75nt)
DRB_TTchemseq_DRB_OHT _40min – 59 million reads single end reads (75nt)

**Antibodies**

RNAPII (405A) Bethyl A304-405A 2 μL/200 μg of DNA ChIP
RNAPII (D8L4Y)  Cell Signaling 14958  7 μL/ 100 μg of DNA ChIP
POLR3A Abcam ab96328 6 μL/150 μg of DNA ChIP
POLR3E Bethyl A303-707A 5 μL/150 μg of DNA ChIP
Y1P Active Motif 61383 3 μL/200 μg of DNA ChIP
S2P (3E10) Chromotek 3E10 75 μL/150 μg of DNA ChIP
S2P (E1Z3G) Cell Signaling 13499  7 μL/ 100 μg of DNA ChIP
S5P (3E8) Chromotek 3E8 75 μL/150 μg of DNA ChIP
S5P (D9N5I) Cell Signaling 13523  7 μL/ 100 μg of DNA ChIP

# ChIP-seq

## Data deposition

☒ Confirm that both raw and final processed data have been deposited in a public database such as GEO.

☐ Confirm that you have deposited or provided access to graph files (e.g. BED files) for the called peaks.

| | |
|---|---|
| **Data access links**<br>*May remain private before publication.* | E-MTAB-13197 |
| **Files in database submission** | RNAPOLII_Total_mOHT<br>RNAPOLII_Total_pOHT<br>POLR3A_mOHT<br>POLR3A_pOHT<br>POLR3E_mOHT<br>POLR3E_pOHT<br>RNAPOLII_Y1P_mOHT<br>RNAPOLII_Y1P_pOHT<br>RNAPOLII_S2_mOHT<br>RNAPOLII_S2_pOHT<br>RNAPOLII_S5_mOHT<br>RNAPOLII_S5_pOHT<br>RNAPOLII_S7_mOHT<br>RNAPOLII_S7_pOHT<br>Spin1_mOHT<br>Spin1_pOHT<br>RPA_mOHT<br>RPA_pOHT_4H<br>RPA_pOHT_24H |
| **Genome browser session**<br>(e.g. UCSC) | No longer applicable |
| | S7P Chromotek 4E12 75 µL/150 µg of DNA ChIP<br>S9.6 Antibodies Incorporated 6 µg/4 µg of DNA DRIP<br>SPT5 Bethyl A3030-707A 2.5 µL/100 µg of DNA ChIP<br>PAF1 Abcam ab137519 2 µL/150 µg of DNA ChIP<br>RPA Abcam ab10359 4 µg/200 µg ChIP<br>gH2AX Abcam ab176458  1 µL/200 µg of DNA ChIP<br>RAD51 Abcam ab176458 2 µL/200 µg of DNA ChIP<br>H3K4me3 Abcam ab8580 1 µL/25 µg of DNA ChIP<br>SPIN1 Proteintech, 12105-1-AP 7 µL/200 µg of DNA ChIP |
| **Peak calling parameters** | Peak calling was not performed in this study |
| **Data quality** | All sequencing data has been submitted to fastqc analysis. During the sam to bam step, sequencing reads has been filtered based on their Phred score (-q 25). |
| **Software** | bedtools v2.26.0<br>bwa 0.7.12-r1039<br>samtools 1.9<br>deeptools 3.4.3<br>R 4.2 |

