## [Peer Review File · Nature Cell Biology]

Transcriptional repression facilitates for RNA:DNA hybrid accumulation at DNA Double Strand breaks

Corresponding Author: Dr Gaelle Legube

Version 0:

Decision Letter:

*Please delete the link to your author homepage if you wish to forward this email to co-authors.

Dear Dr Legube,

Thank you again for submitting your manuscript, "Transcriptional repression accounts for RNA:DNA hybrid accumulation at DNA Double Strand breaks", to Nature Cell Biology. It has now been seen by 3 referees, who are experts in RNA:DNA hybrids (Referee #1); DSBs (Referee #2); and transcription and DNA damage (Referee #3). As you will see from their comments (attached below), they found this work of potential interest but have raised substantial concerns, which in our view would need to be addressed with considerable revisions before we can consider publication in Nature Cell Biology.

Nature Cell Biology editors discuss the referee reports in detail within the editorial team, including the chief editor, to identify key referee points that should be addressed with priority, as opposed to requests that are overruled as being beyond the scope of the current study. To guide the scope of the revisions, I have listed these points below. Our standard revision period is six months, and we are committed to providing a fair and constructive peer-review process, so please feel free to contact me if you would like to discuss any of the referee comments further or if you anticipate any issues or delays addressing the reviews.

The referees appreciated the importance of understanding the interplay between RNA:DNA hybrids at breaks and transcriptional activity. However, they were not yet convinced by the proposed mechanisms for RNA:DNA hybrid accumulation after damage and their relationship with RNAPII/transcription. Given past conflicting reports in this area, it is important for the study to offer convincing and definitive evidence in support of its conclusions. Therefore, concerns regarding the strength of the model would need to be addressed with experiments and data, and reconsideration of the study for this journal and re-engagement of referees would depend on strength of these revisions. In particular, in our view, it would be essential to dedicate efforts in revision to address the following overarching points:

A- Both Revs#2 and #3 were not yet convinced by the core conclusions and asked for validation using another method to induce breaks as well as controls for the DlvA studies:

Rev#2 points #1-2-3, #6

Rev#3 "Figure 1B shows that in this cellular system used to elegantly induce DSBs..." full paragraph

B- the reviewers also requested further analyses of RNAPII's behavior to better understand what happens to RNAPII and transcription after break induction and to better understand what happens at the pool of TC-DSBs within all DSB sites:

Rev#1 points #1-2

Rev#3:

"The authors suggest that DSB's result in a decreased elongation speed, which results in the accumulation of RNA:DNA hybrids at break sites, however this is mainly based on correlation (e.g. ATM inhibition). The authors should show that indeed the decreased elongation speed is the cause for RNA:DNA hybrids, for example using slowly elongating RNA Pol II mutants.

- Additionally, if Pol II elongation rate are severely downregulated, as suggested by the authors, further downstream of the break site a decrease in Ser2 modified Pol II is expected, the genes will not elongate that fast through the gene body. However this decreased Ser2 ChIP signal at the end of genes upon DSB induction is not detected. The slower transcription rate, or increased termination rate should be more convincingly shown.

- The authors propose a model in which the nascent RNA of a Pol II molecule that has passed the AsiSI site before DSB induction will hybridize back to the resected DNA. This is an interesting model. However I wonder how frequent this event would be. When we assume that splicing happens co-transcriptional in an efficient manner, intron will be removed rapidly. Therefore, only nascent RNA that has just been synthesized, and thus with Pol II very closely located to the break site, could be the source of the RNA:DNA hybrid. Since not every gene is actively transcribed at a given moment in a single cell, and that genes are relatively long with maybe only 1 or a few Pol II molecules engaged, the change that the nascent RNA hybridized with the resected DNA seems very small.

- o Would this suggest that Pol II is loaded on the DSB, explaining the RNA:DNA hybrids formed ?

- o Are more hybrids detected when the DSB is generated in an exon vs an intron, exons will not be co-transcriptionally removed and thus a higher chance of hybridizing.

- o Is there an increased DRIP-seq signal if transcription initiation is inhibited shortly after DSB induction, in this scenario Pol II will be near the end of genes and

- Along this line, how frequent is the presence of the RNA:DNA hybrids at break sites? Does this happen at every break, or only in a very minor fraction. This is an important question to address as this will provide info on how relevant this phenomenon is. The RPA ChIP data

also suggests that only a minor fraction of DSBs has hybrid, the RPA ChIP signal seems very at the exact same genomic regions as where the hybrids are detected, while it is expected that RPA would bind the resected DNA, but not the RNA:DNA hybrids.

• The authors state the RNA:DNA hybrids are formed in cis, however what is the proof that the RNA:DNA hybrids are not formed by RNA transcribed by the other allele? To rule out such a scenario experiments in haploid conditions should be executed."

C- The reviewers questioned the involvement of MRE11, ATM, Spindlin in RNA:DNA hybrid accumulation at DSBs, and the mechanism should be examined further:

Rev#1 points #3, 5, 6

Rev#2 points #5, 7, 8, 9

D- All other referee concerns pertaining to strengthening existing data, providing controls, methodological details, clarifications and textual changes, should also be addressed.

E- Finally, please pay close attention to our guidelines on statistical and methodological reporting (listed below) as failure to do so may delay the reconsideration of the revised manuscript. In particular, please provide:

We would be happy to consider a revised manuscript that would satisfactorily address these points, unless a similar paper is published elsewhere, or is accepted for publication in Nature Cell Biology in the meantime.

- ensure that it conforms to our format instructions and publication policies (see below and www.nature.com/nature/authors/).

- provide a point-by-point rebuttal to the full referee reports verbatim, as provided at the end of this letter.

- provide the completed Editorial Policy Checklist (found here <https://www.nature.com/authors/policies/Policy.pdf>), and Reporting Summary (found here <https://www.nature.com/authors/policies/ReportingSummary.pdf>). This is essential for reconsideration of the manuscript and these documents will be available to editors and referees in the event of peer review. For more information see <http://www.nature.com/authors/policies/availability.html> or contact me.

Nature Cell Biology is committed to improving transparency in authorship. As part of our efforts in this direction, we are now requesting that all authors identified as 'corresponding author' on published papers create and link their Open Researcher and Contributor Identifier (ORCID) with their account on the Manuscript Tracking System (MTS), prior to acceptance. ORCID helps the scientific community achieve unambiguous attribution of all scholarly contributions. You can create and link your ORCID from the home page of the MTS by clicking on 'Modify my Springer Nature account'. For more information please visit <http://www.springernature.com/orcid>.

Link Redacted

We hope that you will find our referees' comments and editorial guidance helpful. Please do not hesitate to contact me if there is anything you would like to discuss. Thank you again for considering NCB for your work.

Best wishes,

Melina

Melina Casadio, PhD
Senior Editor, Nature Cell Biology
ORCID ID: <https://orcid.org/0000-0003-2389-2243>

Reviewers' Comments:

Reviewer #1:

Remarks to the Author:

In the manuscript entitled "Transcriptional repression accounts for RNA:DNA hybrid accumulation at DNA Double Strand breaks", the authors carefully examined the accumulation of RNAPII and RNAPIII recruitment, as well as the distribution and strand-specificity of RNA:DNA hybrids on a genome-wide scale following the induction of multiple DSBs in both active and intergenic loci. By doing so, the authors show that RNA:DNA hybrids arise at transcribed loci undergoing DSB formation (TC-DSBs) from pre-existing RNA that hybridize with single-stranded DNA after resection at the DSB. In addition, the data indicate a role of the DNA damage kinase ATM as well as Spindlin 1, a reader of H3K4 trimethylation, in the transcriptional repression of the locus and RNA:DNA hybrid accumulation. Together, the authors propose a model where DSB-induced RNA:DNA hybrids accumulate at DSBs within actively transcribed RNAPII-enriched loci (TC-DSBs) as a result of ATM/Spindlin1 dependent alteration of RNAPII processivity. The hybrids increase short-range resection at the DSB and therefore promote RPA and RAD51 loading to initiate the HR-mediated DSB repair, however it is essential to remove the hybrids in a long-term to allow completion of the repair process and preservation of genome stability.

Overall, this is a solid piece of work and the presented data strongly support the authors' conclusions. The findings also provide an important contribution to the field given the partially contradictory studies that have been published on the role of RNA:DNA hybrids in DSB repair. The presented model can synergize with previous results and explain how the dynamics of R-loop formation during short-range versus long-range resection can at the same time differentially improve or impair efficient repair of DSBs located in actively transcribed loci. I'm in favor for publication of this impressive piece of work if the authors can address a few remaining issues stated below:

Major comments:

- 1) Along the whole manuscript, the authors discriminate between DSBs that occur on silent loci without detectable RNAPII and TC-DSBs on active loci with detectable RNAPII. What is the fraction of TC-DSBs among the 80 robustly-induced DSB sites. Based on previous publications, about half of these 80 sites can be considered in proximity of active genes and the other half in intergenic locations. Is that also true here based on the RNAPII ChIP-Seq results presented in this study? Also this would be an important information in Extended Data Figure 1c how many of the sites are classified as low/medium and high RNAPII occupancy sites.
- 2) Figure 2: The authors analyze RNAPII CTD phosphorylation states by ChIP-Seq at TC-DSBs and intergenic DSB loci and find a small accumulation of S2P/S5P and Y1P at the TSS consistent with a decreased elongation rate and/or premature termination. If transcription elongation from the is affected, why do the authors not observe a decreased RNAPII occupancy in the gene body of these affected genes? Can the authors normalize with a ChIP-Seq dataset using an antibody detecting total RNAPII? This will allow the authors to calculate proper RNAPII stalling/pausing indexes on the affected genes.
- 3) Figure 3i. The authors use depletion of SETX as one of the major R-loop helicases to accumulate RNA:DNA hybrid levels and observe increased resection by END-Seq. The authors should confirm that SETX was indeed depleted and RNA:DNA hybrids accumulated under the used conditions to support this conclusion.
- 4) Figure 4b. The authors only perform statistical tests on the difference between -DSB and +DSBs on R-loops formed on the template strand. However, a small (but maybe also significant) increase is observed on R-loops formed on the non template strand. The authors should at least discuss this in the manuscript.
- 5) Figure 5d/e. The authors analyze cDNA levels at different DSB loci after SPIN1 knockdown or MS31 treatment, a specific inhibitor of SPIN1/H3K4me3 interaction. To what extent does SPIN1 knockdown or MS31 treatment on its own (under -DSB condition) affect cDNA levels of the investigated loci? This information is lost due to the double-normalization. Additionally, the authors should provide evidence that MS31 treatment is indeed blocking the interaction with H3K4me3 as expected, for example by ChIP-qPCR of SPIN1 with and without MS31 treatment at TSS sites.
- 6) Figure 6h. The model postulates that SETX is the responsible R-loop helicase to remove the hybrids that would otherwise become toxic. However, no strong evidence is provided that SETX would be the major player here, considering the fact that a plethora of other DEAD-box RNA helicases have now been described to show R-loop helicase activities and in vivo functions. The authors should adjust the model and discuss this in the manuscript.

Minor comments:

- 1) Extended Data Figure 5l. The presented ChIP is done from n=1 biological replicates. It would be good to provide additional biological replicates to strengthen their conclusions.

Reviewer #2:

Remarks to the Author:

The authors investigated the mechanisms by how DNA:RNA hybrids are generated in the vicinity of DNA double-strand breaks (DSBs). Multiple previous studies have reported that DSBs induce the formation of an R-loop composed of ssDNA and DNA:RNA hybrids. To date, there are two distinct models regarding the origin of RNA in the R-loop, i.e., de novo RNA synthesis by RNA pol2 or pol3 which is recruited in response to DSBs, or the use of pre-existing RNA synthesized before DSB induction. Although the existence of DNA:RNA hybrids near DSBs is evident, the origin of the RNA in these hybrids has been debated. In this study, the authors demonstrated no recruitment of RNA polymerases in response to DSB induction using the AsiSI-based assay. In addition, the authors found that RNA:DNA hybrids are generated in correspondence with the direction of transcription, proposing a model in which the RNA in the DNA:RNA hybrid has already been synthesized by ongoing transcription prior to DSB induction. Furthermore, the authors showed that the DNA:RNA hybrid formed on the ssDNA overhang generated by resection and the hybrid potentiated CtIP/MRE11-dependent resection, followed by RPA recruitment. Finally, the authors showed that Spindlin 1 was recruited to DSBs and promoted transcriptional repression, resulting in the formation of DNA:RNA hybrids. Overall, the study is well-designed, and the proposed model is refined and plausible. In addition, although sequence-based assays require sophisticated skills and knowledge, the obtained data must be highly reliable because the authors' group has expertise in the analysis of DSB repair using NGS technology in D1VA cells. Nevertheless, because the model proposed by the authors conflicts with the other model from other groups supporting the model of de novo RNA synthesis, the release of this paper will have a significant impact on the field of DNA damage response. Therefore, the word choice for data interpretation must be made cautiously. Overall, I am supportive of the model in this study; however, the authors should address the concerns listed below.

1. The major concern is that the AsiSI-based assay is likely unable to detect a single molecule at a DSB site and a repair event that occurs over a period of seconds to minutes. The authors stated that the D1VA assay could identify the transient binding of NHEJ factors. However, because XRCC4 forms filaments at DSB sites, its detection may be easier than the other NHEJ factors. Can the authors detect Ku70/80 and DNA-PKcs recruitment using ChIP-seq?

Ionizing radiation (IR) can generate DSBs in less than 1 s. If the recruitment of RNAP per break is within a minute or less and there is only

a single molecule per break, such a transient reaction may not be detected by ChIP-seq. In addition, the disadvantage of the AsiSI-based assay is that it requires ~4 h to introduce DSBs, and the timing of DSB induction must be heterogeneous at each locus and in each cell. Thus, the sensitivity of the assay in terms of the number of molecules and transient events was still lower than that of the live imaging-based assays. I am positive about the model proposed in this study; however, I strongly suggest that the authors tone down this statement throughout the manuscript. If the authors insist on the model with minimal modifications, further control experiments may be required.

2. In this study, all the data were obtained using the DivA assay. I do not think that the repair process of a restriction enzyme can be generalized because, as the authors described in the Discussion, AsiSI continuously digests DNA until mutations in the recognition sequence for AsiSI are generated. Does CRISPR-Cas9-based digestion show similar results? DSBs generated by CRISPR-Cas9 may fail to form RNA:DNA hybrids because of the presence of gRNA. If possible, the authors should confirm the critical findings at other TC-sites using a CRISPR-Cas9-based assay.

3. In Figure 6, the authors used etoposide, which induces DSBs by inhibiting topoisomerase II activity. Because topo II activity is associated with transcription, some DSBs must be TC-DSBs. However, indeed, most DSBs are repaired by NHEJ but not HR (PMID:24316220). Although the authors refer to a paper showing the involvement of SETX in etoposide-induced DSBs, it is unclear how many etoposides induced DSBs are associated with transcription. As mentioned above, a major concern is that the conclusion of this study is based on the results of the DivA assay. If etoposide is available as a TC-DSB inducer, the authors should confirm the key findings, such as RPA, RAD51, and gH2AX foci, by etoposides. In addition, identification of the BRCA1 foci will be informative in this study.

4. In Figure 3, the authors concluded that DSB-induced RNA:DNA hybrids form on the ssDNA overhang generated by end resection (as stated in the Discussion section). However, this process is not included in the model in Figure 6h. This should be clarified. As an alternative possibility, does failure to initiate resection promote hybrid degradation? The authors may not need to perform additional experiments to answer this question; however, the proposed mechanisms should be carefully spelled out in a revised manuscript.

5. In Figure 3, the authors used mirin, which is an inhibitor of MRE11 3'-5' exonuclease activity. Since the direction of the exonuclease is opposite (resection undergoes 5'-3' direction), it is unclear how MRE11 3'-5' exonuclease is involved in this process. The authors should clarify the requirement of MRE11 endonuclease activity using an inhibitor, PFM01, or MRE11 mutant.

6. The authors' group previously uncovered transcription-coupled DSB repair pathway choices using the DivA assay. In this study, the authors probably used TC-DSB sites that do not recruit NHEJ factors. In contrast, several recent studies have shown that NHEJ factors bind to the DSB ends before resection (PMID:31934630; PMID:36917982). The authors almost ignored the fact that NHEJ factors are present in DSBs undergoing HR. Or NHEJ is not recruited at TC-DSB sites at any time points? The authors should mention the interaction between NHEJ factors and the TC-DSB/hybrid. Importantly, if the DivA assay fails to detect NHEJ factors at TC-DSB sites, RNAP may be also undetectable?

7. As shown in Figure 6h, SPIN1 is located on the left side but not on the right side of the DSB. However, as shown in Figure 5 and Extended data Figure 5d, SPIN1 is recruited to both sides of the DSBs. The authors should clarify this point further.

8. In Extended data Figure 5l-m, the authors show a decreased gH2AX signal in MS31-treated cells. Is this result simply explained by the reduction in the number of DSBs induced by AsiSI? The change of chromatin structure may affect the efficiency of DNA cleavage by restriction enzymes. If the number of DSB induction is not affected, ATM activity is reduced in SPIN1 depleted or MS31-treated cells. Therefore, ATM activity should be confirmed, for example by monitoring ATM autophosphorylation after IR.

9. As shown in Figure 3, RPA recruitment (i.e., resection) was observed on both sides. In contrast, hybrids were observed on the click strand on the right side. How was resection initiated on the left side? Additional experiments are not required; however, the left-side DNA in the model figure of Figure 6 should be amended for clarity because the DNA on both sides must be resected. The current version is confusing.

10. In the manuscript, the authors use the term "short-range resection." This term should be defined in earlier of the manuscript. If MRE11 endonuclease activity is required for this process, "initiation of resection" may be a better choice of words.

A. Summary of the key results

this is written in the letter

B. Originality and significance: if not novel, please include reference

As written in the letter, this is original and important finding in the field of DDR.

C. Data & methodology: validity of approach, quality of data, quality of presentation

No problem

D. Appropriate use of statistics and treatment of uncertainties

No problem

E. Conclusions: robustness, validity, reliability

The data is reliable, but some statement should be amended as mentioned in the letter.

F. Suggested improvements: experiments, data for possible revision

this is written in the letter

G. References: appropriate credit to previous work?

Yes, no problem

F. Clarity and context: lucidity of abstract/summary, appropriateness of abstract, introduction and conclusions

No problem

Reviewer #3:

Remarks to the Author:

In the manuscript entitled: "transcriptional repression account for RNA:DNA hybrid accumulation at DNA double strand breaks" Lesage and co-authors have performed a omics-based tour de force to answer one of the big debates in the TC-DSB field, whether Pol II is recruited at DSBs and what the cellular function to the DNA damage response could be. In this study the authors were unable to identify recruitment of Pol II at DSB sites, both in transcribed or non-transcribed loci. Interestingly, the authors revealed the presence of RNA:DNA hybrids at DSBs only in transcribed genes. These RNA:DNA hybrids arise in a non-symmetrical manner, suggesting that these are caused by the hybridization of nascent RNA with the 3' overhang upon resection. Finally the authors identify that Spindlin 1 promotes RNA:DNA hybrids at DSBs, as a consequence of ATM mediated transcriptional shut-down at DSBs in transcribed loci.

This study addresses the issue whether Pol II is actively recruited at DSB-sites in a convincingly manner, which is an important issue for the field, together with the additional insights provided this manuscript is a strong candidate for publication in NCB, given that the concerns indicated below are addressed in a convincingly manner.

Major comments:

- Figure 1B shows that in this cellular system used to elegantly induce DSBs, on the majority of DSB sites an accumulation of Pol II is observed. This is in itself a striking observation, this could for example be caused by the fact that these DSB are predominately localized near transcriptional pause sites, or that these DSB site have a unique chromatin structure resulting in more Pol II. This high Pol II abundance could explain why no Pol II accumulation could be observed upon DSB induction. As one of the strong points of this manuscript is to show that no Pol II is accumulated at DSBs, the authors should show that their model is also relevant for all types of DSBs and use alternative methods to induce DSBs and confirm that main results.
- How efficient is the AsiSI mediated DSB induction, in what percentage of cells is a break induced, is at a high percentage of AsiSI sites in every cell a break induced? This is crucial info, as if this is only in a low percentage of cells, this might explain why not a Pol II recruitment by ChIP could be detected.
- The authors suggest that DSB's result in a decreased elongation speed, which results in the accumulation of RNA:DNA hybrids at break sites, however this is mainly based on correlation (e.g. ATM inhibition). The authors should show that indeed the decreased elongation speed is the cause for RNA:DNA hybrids, for example using slowly elongating RNA Pol II mutants.
- Additionally, if Pol II elongation rate are severely downregulated, as suggested by the authors, further downstream of the break site a decrease in Ser2 modified Pol II is expected, the genes will not elongate that fast through the gene body. However this decreased Ser2 ChIP signal at the end of genes upon DSB induction is not detected. The slower transcription rate, or increased termination rate should be more convincingly shown.
- The authors propose a model in which the nascent RNA of a Pol II molecule that has passed the AsiSI site before DSB induction will hybridize back to the resected DNA. This is an interesting model. However I wonder how frequent this event would be. When we assume that splicing happens co-transcriptional in an efficient manner, intron will be removed rapidly. Therefore, only nascent RNA that has just been synthesized, and thus with Pol II very closely located to the break site, could be the source of the RNA:DNA hybrid. Since not every gene is actively transcribed at a given moment in a single cell, and that genes are relatively long with maybe only 1 or a few Pol II molecules engaged, the change that the nascent RNA hybridized with the resected DNA seems very small.
 - o Would this suggest that Pol II is loaded on the DSB, explaining the RNA:DNA hybrids formed ?
 - o Are more hybrids detected when the DSB is generated in a exon vs an intron, exons will not be co-transcriptionally removed and thus a higher chance of hybridizing.
 - o Is there a increased DRIP-seq signal if transcription initiation is inhibited shortly after DSB induction, in this scenario Pol II will be near the end of genes and
- Along this line, how frequent is the presence of the RNA:DNA hybrids at break sites? Does this happen at every break, or only in a very minor fraction. This is an important question to address as this will provide info on how relevant this phenomenon is. The RPA ChIP data also suggests that only a minor fraction of DSBs has hybrid, the RPA ChIP signal seems very at the exact same genomic regions as where the hybrids are detected, while it is expected that RPA would bind the resected DNA, but not the RNA:DNA hybrids.
- The authors state the RNA:DNA hybrids are formed in cis, however what is the proof that the RNA:DNA hybrids are not formed by RNA transcribed by the other allele? To rule out such a scenario experiments in haploid conditions should be executed.

Minor comments:

- Fig. 4C, D and 6B should be shown for other genes as well, or DRIP-seq should be performed.
- Fig. Ext. 4C, DRIP-PCR should be shown for these genes as well
- Is the DSB induction affected by the treatments or knockdowns used, e.g. ATM inhibition, MS31 or NELFE depletion?

MANUSCRIPT FORMAT – please follow the guidelines listed in our Guide to Authors regarding manuscript formats at Nature Cell

Biology.

Methods should be written concisely, but should contain all elements necessary to allow interpretation and replication of the results. As a guideline, Methods sections typically do not exceed 3,000 words. The Methods should be divided into subsections listing reagents and techniques. When citing previous methods, accurate references should be provided and any alterations should be noted. Information must be provided about: antibody dilutions, company names, catalogue numbers and clone numbers for monoclonal antibodies; sequences of RNAi and cDNA probes/primers or company names and catalogue numbers if reagents are commercial; cell line names, sources and information on cell line identity and authentication. Animal studies and experiments involving human subjects must be reported in detail, identifying the committees approving the protocols. For studies involving human subjects/samples, a statement must be included confirming that informed consent was obtained. Statistical analyses and information on the reproducibility of experimental results should be provided in a section titled "Statistics and Reproducibility".

All Nature Cell Biology manuscripts submitted on or after March 21 2016 must include a Data availability statement at the end of the Methods section. For Springer Nature policies on data availability see <http://www.nature.com/authors/policies/availability.html>; for more information on this particular policy see <http://www.nature.com/authors/policies/data/data-availability-statements-data-citations.pdf>. The Data availability statement should include:

- Accession codes for primary datasets (generated during the study under consideration and designated as "primary accessions") and secondary datasets (published datasets reanalysed during the study under consideration, designated as "referenced accessions"). For primary accessions data should be made public to coincide with publication of the manuscript. A list of data types for which submission to community-endorsed public repositories is mandated (including sequence, structure, microarray, deep sequencing data) can be found here <http://www.nature.com/authors/policies/availability.html#data>.
- Unique identifiers (accession codes, DOIs or other unique persistent identifier) and hyperlinks for datasets deposited in an approved repository, but for which data deposition is not mandated (see here for details <http://www.nature.com/sdata/data-policies/repositories>).
- At a minimum, please include a statement confirming that all relevant data are available from the authors, and/or are included with the manuscript (e.g. as source data or supplementary information), listing which data are included (e.g. by figure panels and data types) and mentioning any restrictions on availability.
- If a dataset has a Digital Object Identifier (DOI) as its unique identifier, we strongly encourage including this in the Reference list and citing the dataset in the Methods.

We recommend that you upload the step-by-step protocols used in this manuscript to the Protocol Exchange. More details can found at www.nature.com/protocolexchange/about.

DISPLAY ITEMS – main display items are limited to 6-8 main figures and/or main tables for Articles, Resources, Technical Reports; and

5 main figures and/or main tables for Letters. For Supplementary Information see below.

All imaging data should be accompanied by scale bars, which should be defined in the legend.

Cropped images of gels/blots are acceptable, but need to be accompanied by size markers, and to retain visible background signal within the linear range (i.e. should not be saturated). The boundaries of panels with low background have to be demarked with black lines. Splicing of panels should only be considered if unavoidable, and must be clearly marked on the figure, and noted in the legend with a statement on whether the samples were obtained and processed simultaneously. Quantitative comparisons between samples on different gels/blots are discouraged; if this is unavoidable, it should only be performed for samples derived from the same experiment with gels/blots were processed in parallel, which needs to be stated in the legend.

Unprocessed scans of all key data generated through electrophoretic separation techniques need to be presented in a supplementary figure that should be labelled and numbered as the final supplementary figure, and should be mentioned in every relevant figure legend. This figure does not count towards the total number of figures and is the only figure that can be displayed over multiple pages, but should be provided as a single file, in PDF or TIFF format. Data in this figure can be displayed in a relatively informal style, but size markers and the figure panels corresponding to the presented data must be indicated.

The total number of Supplementary Figures (not including the "unprocessed scans" Supplementary Figure) should not exceed the

number of main display items (figures and/or tables (see our Guide to Authors and March 2012 editorial <http://www.nature.com/ncb/authors/submit/index.html#suppinfo>; <http://www.nature.com/ncb/journal/v14/n3/index.html#ed>). No restrictions apply to Supplementary Tables or Videos, but we advise authors to be selective in including supplemental data.

GUIDELINES FOR EXPERIMENTAL AND STATISTICAL REPORTING

REPORTING REQUIREMENTS – To improve the quality of methods and statistics reporting in our papers we have recently revised the reporting checklist we introduced in 2013. We are now asking all life sciences authors to complete two items: an Editorial Policy Checklist (found here <https://www.nature.com/authors/policies/Policy.pdf>) that verifies compliance with all required editorial policies and a reporting summary (found here <https://www.nature.com/authors/policies/ReportingSummary.pdf>) that collects information on experimental design and reagents. These documents are available to referees to aid the evaluation of the manuscript. Please note that these forms are dynamic 'smart pdfs' and must therefore be downloaded and completed in Adobe Reader. We will then flatten them for ease of use by the reviewers. If you would like to reference the guidance text as you complete the template, please access these flattened versions at <http://www.nature.com/authors/policies/availability.html>.

Version 1:

Decision Letter:

Dear Dr. Legube,

Thank you for your patience and for submitting your revised manuscript "Transcriptional repression accounts for RNA:DNA hybrid accumulation at DNA Double Strand breaks" (NCB-A51827A). It has now been seen by the original referees and their comments are below. The reviewers find that the paper has improved in revision, and therefore we'll be happy in principle to publish it in Nature Cell Biology, pending minor revisions to satisfy the referees' final requests and to comply with our editorial and formatting guidelines.

Thank you again for your interest in Nature Cell Biology Please do not hesitate to contact me if you have any questions.

Best regards,

George Inglis

George Inglis, PhD
Senior Editor

<https://www.nature.com/ncb/research-cross-journal-editorial-team> Research Cross-Journal Editorial Team
Nature Cell Biology

Reviewer #1 (Remarks to the Author):

The authors have adequately addressed my previous concerns with additional experiments and analyses that significantly improved and further substantiated and extended their previous conclusions. Importantly, the authors now provide convincing evidence that de novo recruitment of RNAPII/RNAPIII at DSB sites is likely no major mechanism at play but rather the accumulation of RNA:DNA hybrids is caused by pre-existing RNA that is hybridizing at the DSB site and shows a role in the initiation of resection. The authors could further generalize their findings by showing similar effects at non-AscSI break sites induced by CRISPR/Cas9 or Etoposide. Finally, the authors provide a new model that the transcriptional shut down observed at DSB sites is regulated at the level of RNAPII promoter proximal pausing, as evidenced by the involvement of SPT5 eviction and PAF1 complex recruitment. Together, these data suggest that DSBs elicit a control of transcriptional activity at the promoter proximal pausing step and provides a fresh perspective to this debated field that can synergize many of the previous (in part conflicting) studies. I'm fully supportive publication in Nature Cell Biology and have no remaining concerns.

Reviewer #2 (Remarks to the Author):

The authors adequately addressed all of my concerns. I have no further issues with the present study.

Reviewer #3 (Remarks to the Author):

I would like to thank the authors for their thorough revisions. The numerous additional experiments significantly strengthen the manuscript, providing important new insights that directly address my initial concerns. The new data convincingly support the authors' conclusions and enhance the overall impact of the study. I have no further major concerns, and I find the article acceptable for publication in its current form that will be of great interest to the field.

Version 2:

Decision Letter:

Dear Dr. Legube,

I am pleased to inform you that your manuscript, "Transcriptional repression facilitates for RNA:DNA hybrid accumulation at DNA Double Strand breaks", has now been accepted for publication in Nature Cell Biology.

Please note that *Nature Cell Biology* is a Transformative Journal (TJ). Authors may publish their research with us through the traditional subscription access route or make their paper immediately open access through payment of an article-processing charge (APC). Authors will not be required to make a final decision about access to their article until it has been accepted. [Find out more about Transformative Journals](https://www.springernature.com/gp/open-research/transformative-journals)

Authors may need to take specific actions to achieve

research/funding/policy-compliance-faqs"> compliance with funder and institutional open access mandates. If your research is supported by a funder that requires immediate open access (e.g. according to

If you have not already done so, we strongly recommend that you upload the step-by-step protocols used in this manuscript to protocols.io (<https://protocols.io>), an open online resource that allows researchers to share their detailed experimental know-how. All uploaded protocols are made freely available and are assigned DOIs for ease of citation. Protocols and Nature Portfolio journal papers in which they are used can be linked to one another, and this link is clearly and prominently visible in the online versions of both. Authors who performed the specific experiments can act as primary authors for the Protocol as they will be best placed to share the methodology details, but the Corresponding Author of the present research paper should be included as one of the authors. By uploading your Protocols onto protocols.io, you are enabling researchers to more readily reproduce or adapt the methodology you use, as well as increasing the visibility of your protocols and papers. You can also establish a dedicated workspace to collect your lab Protocols. Further information can be found at <https://www.protocols.io/help/publish-articles>.

Nature Cell Biology encourages authors presenting evidence for cell, biological, molecular, and genetic interactions to consider communicating these findings using Biofactoid (<https://biofactoid.org/>). This tool helps users share a searchable representation of interactions (e.g. binding, gene expression, post-translational modification) between genes, gene products, or chemicals. Information added to Biofactoid, with author attribution, is shared on social media and public databases, such as Pathway Commons, where it can be discovered and analyzed in the context of a large and growing corpus of knowledge.

Best regards,

George

George Inglis, PhD
Senior Editor

Nature Cell Biology

** Visit the Springer Nature Editorial and Publishing website at

Point by point response to reviewer comments

Overview of the revised manuscript and general comments

First of all, we would like to thank all three referees for their thorough work and understanding of our manuscript. We also want to thank them for their enthusiasm regarding our findings. Referees have raised valid and constructive criticism regarding our approaches and how certain limits may have impacted our initial conclusions.

In general, most of their points were aimed at strengthening our proposed model for RNA:DNA hybrids formation at DSBs. Thanks to several insightful suggestions made by the 3 referees, we have considerably improved our manuscript and added a large amount of novel data, which allowed us not only to **generalize our findings to other DSB induction methods** but to also provide **additional insights regarding the mechanisms of DSB-induced transcriptional repression**.

We are confident that the RNA:DNA hybrids detected in our study are not produced *via de novo* RNAPII/III loading and transcription at DSBs and we strongly believe that this does not constitute a prevalent mechanism accounting for their formation in most contexts.

Indeed:

- 1- We could not detect any RNA polymerase recruitment at DSBs **induced by AsiSI, CRISPR/Cas9 or etoposide** despite many attempts by ChIP-qPCR and ChIP-seq.
- 2- Consistently, we **did not detect *de novo* transcription at DSBs by measuring nascent transcripts with TT_{chem}-seq.**
- 3- qDRIP-seq revealed that **the hybridized RNA strand mainly corresponds to a pre-existing RNA** (either pre-mRNA or PROMPT), strongly suggesting the requirement of pre-existing RNA molecules for their formation.
- 4- If such a *de novo* recruitment occurs, it should, according to the previously proposed model, yield **bidirectional and symmetric R-loops/RNA:DNA hybrids at all DSBs, which is clearly not observed here.**
- 5- our work is also in agreement with previous reports using different models and methods (for example imaging at FokI-induced DSBs in PMID: 20550933) as well as with a recently published work demonstrating unambiguously that DSB induced by CRISPR/Cas9-induced in active genes leads to a fast *in-cis* transcriptional repression of the damaged gene and a strong decrease of RNAPII (measured by ChIP-seq and ChIP-qPCR, PMID: 38954517).

Thus, while we do not exclude that RNA Polymerase recruitment may occur in a very transient manner or/and in some specific contexts, we **consider it unlikely to account for R-loop formation at DSBs.**

Moreover, we also provide **more insights into the mechanisms of transcriptional repression that leads to RNA:DNA hybrids accumulation.** We now report that DSB-induced transcriptional repression involves SPT5 eviction and PAF1 complex recruitment, suggesting that DSB elicit a control of transcriptional activity at the promoter proximal pausing step (see below and in the manuscript for references). This step has been recently suggested to function as a key quality control checkpoint for transcriptional integrity by allowing the assembly of a fully competent elongation complex (PMID: 39504960). Our novel data thus suggest that this transcriptional quality checkpoint is under the control of the DNA Damage response. Of

importance, depletion of PAF1 not only impairs transcriptional repression upon damage but also interferes with RNA:DNA hybrids accumulation at DSBs. This phenotype is strikingly similar to the depletion of SPIN1 (which was described in the original manuscript), further validating a role in transcriptional repression in DSB-induced hybrid formation.

Major additions to the revised manuscript include:

- Assessment of nascent transcription by TT_{chem}-seq and RNAPII velocity by DRB/TT_{chem}-seq
- ChIP-qPCR, DRIP-qPCR and RT-qPCR experiments performed at several Cas9-induced DSBs.
- END-seq and ChIP-qPCR following DSB induction by etoposide
- DRIP-qPCR in HAP1 haploid cells or using an inhibitor of MRE11 endonuclease activity (PFM01)
- PAF1 and SPT5 ChIP-seq
- Several panels to establish the role of PAF1/SPIN1 in transcriptional repression, RNA:DNA hybrids formation, resection, and HR
- The consequences of this pathway on genome instability by measuring translocation frequencies.

Given that in the revised version we now focus a lot more on transcriptional repression, many figures have changed. To improve clarity, some panels were modified in terms of presentation (for instance all genome browsers snapshots), while some were removed (for instance regarding SPIN1 for which many previous panels did not seem to be as important in the revised manuscript). Any of these data could be added back if the referees consider them to be important.

Reviewer #1:

Remarks to the Author:

In the manuscript entitled “Transcriptional repression accounts for RNA:DNA hybrid accumulation at DNA Double Strand breaks”, the authors carefully examined the accumulation of RNAPII and RNAPIII recruitment, as well as the distribution and strand-specificity of RNA:DNA hybrids on a genome-wide scale following the induction of multiple DSBs in both active and intergenic loci. By doing so, the authors show that RNA:DNA hybrids arise at transcribed loci undergoing DSB formation (TC-DSBs) from pre-existing RNA that hybridize with single-stranded DNA after resection at the DSB. In addition, the data indicate a role of the DNA damage kinase ATM as well as Spindlin 1, a reader of H3K4 trimethylation, in the transcriptional repression of the locus and RNA:DNA hybrid accumulation. Together, the authors propose a model where DSB-induced RNA:DNA hybrids accumulate at DSBs within actively transcribed RNAPII-enriched loci (TC-DSBs) as a result of ATM/Spindlin1 dependent alteration of RNAPII processivity. The hybrids increase short-range resection at the DSB and therefore promote RPA and RAD51 loading to initiate the HR-mediated DSB repair, however it is essential to remove the hybrids in a long-term to allow completion of the repair process and preservation of genome stability.

Overall, this is a solid piece of work and the presented data strongly support the authors' conclusions. The findings also provide an important contribution to the field given the partially contradictory studies that have been published on the role of RNA:DNA hybrids in DSB repair. The presented model can synergize with previous results and explain how the dynamics of R-loop formation during short-range versus long-range resection can at the same time differentially improve or impair efficient repair of DSBs located in actively transcribed loci. I'm in favor for publication of this impressive piece of work if the authors can address a few remaining issues stated below:

We would like to thank this reviewer for their positive comments and thorough work and understanding of our manuscript.

Major comments:

1) Along the whole manuscript, the authors discriminate between DSBs that occur on silent loci without detectable RNAPII and TC-DSBs on active loci with detectable RNAPII. What is the fraction of TC-DSBs among the 80 robustly-induced DSB sites. Based on previous publications, about half of these 80 sites can be considered in proximity of active genes and the other half in intergenic locations. Is that also true here based on the RNAPII ChIP-Seq results presented in this study? Also this would be an important information in Extended Data Figure 1c how many of the sites are classified as low/medium and high RNAPII occupancy sites.

We thank the referee for this comment and apologize for not providing this information in the original manuscript. As suggested by this referee, we have now added a supplemental Table (Supp Table 1) and a Supplemental Figure (Ext. Data Fig. 1a) detailing the positions of each AsiSI-induced DSBs relative to genomic annotations or transcriptional status defined by RNA-seq and TT_{chem}-seq (the parameters used for the classification are detailed in the Methods section). In addition, in the new version of the manuscript, we now present data obtained at TC-

DSBs *versus* silent DSBs either with individual examples (Fig. 1b; 2d; 4a; 5l; Ext. Data Fig. 4a) or heatmaps and boxplots comparing all DSBs in each category (Fig. 1c, 2b-c; 2f; Ext. Data Fig. 1b; 1e; 2c; 2e; 2f). We also show data obtained at DSBs induced in intron *versus* exon as examples (Fig. 1b; 2d; 4a; 5l; Ext. Data Fig. 4a) or globally as boxplots (Ext. Data Fig. 1g; 2g; 4b). As for the classification of high/medium/low RNAPII (now presented Ext Data Fig. 1f), the 80 DSBs were just split in three equivalent categories of nearly equal size based on RNAP II levels. This is now indicated on the plots and in the Methods section.

2) Figure 2: The authors analyze RNAPII CTD phosphorylation states by ChIP-Seq at TC-DSBs and intergenic DSB loci and find a small accumulation of S2P/S5P and Y1P at the TSS consistent with a decreased elongation rate and/or premature termination. If transcription elongation from the is affected, why do the authors not observe a decreased RNAPII occupancy in the gene body of these affected genes? Can the authors normalize with a ChIP-Seq dataset using an antibody detecting total RNAPII? This will allow the authors to calculate proper RNAPII stalling/pausing indexes on the affected genes.

Thank you for this comment. As mentioned above in the introductory paragraph, we have now substantially changed the manuscript regarding these aspects.

The revision process led to several changes in the ChIP-seq data for RNAPII and its phosphorylation forms.

1-We have strengthened our findings. by performing additional ChIP-seq experiments using other antibodies for total, S5P- and S2P-modified RNAPII (Fig. 2 of the revised manuscript).

2- In order to reinforce our findings in Fig. 1 that shows no RNAPII accrual at DSBs upon DSB induction, the phosphorylated RNAPII ChIP-seq data are now presented with a focus on silent and TC-DSBs rather than on damaged genes.

3-In light of a recent report from the Gullerova lab showing a global increase of RNAPII-Y1P following DSB induction (PMID: 38467418), we decided to modify our analysis strategy in order to focus on changes that specifically occur at DSBs and not elsewhere on the genome. For this, we scaled the data obtained before and after DSB induction in order to compensate for potential systematic differences that could take place at all genes of the genome (see Methods section). By doing so, we observed that **RNAPII levels are generally decreased at DSBs**, which is accompanied by a slight reduction in the signal for S5P-, S7P-, T1P- but not S2P-modified RNAPII (Fig. 2a-c; Ext. Data Fig. 2c). This is also observed when plotting the signal across entire damaged genes, including at the TSS, when compared to control genes (see below, Fig. 1 for reviewers). Altogether these data suggest that, when accounting for genome-wide alterations of RNAP II upon damage (which were not considered in the original version of the manuscript), there is no obvious redistribution of RNAPII that specifically takes place at damaged genes as compared to control genes. For space constraints, we have decided to remove the analyses of RNAPII profiles across damaged genes from Figure 2 in order to focus on events occurring around DSBs. The results can be included again if this referee consider this to be necessary.

4-As suggested by this reviewer, we measured the pausing index based on ChIP-seq data for either total RNAPII or using S2P-RNAPII normalized to total RNAPII. These analyses did not indicate a clear and consistent change in the pausing index following DSB induction (Fig. 2 for reviewers) and were therefore not included in the revised manuscript.

Fig. 1 for reviewers

Figure 1: DSB induction does not cause a redistribution of RNAPII or its phosphorylated forms on damaged genes.
Average profiles of RNAPII and its modified forms on damaged and control genes in the absence or presence of DSB induction

Fig. 2 for reviewers

Fig. 2: DSB induction does not strongly alter RNAPII pausing.

- Pausing index was calculated using total RNAPII ChIP-seq as the Log₂ ratio of RNAPII signal on +500bp around TSS divided by the signal on the rest of the gene body
- Pausing index were calculated using three replicates of RNAPII S2P normalized against total RNAPII, as the Log₂ ratio of RNAPII signal on +500bp around TSS divided by the signal on the rest of the gene body

However, we now provide a large amount of new data in the revised manuscript indicating that DSBs induce elongation defects at damaged genes that is linked to the eviction of SPT5 and the recruitment of the PAF1 complex, two factors previously reported to regulate RNAPII promoter-proximal pause (PPP) release.

We first measured nascent transcription by TT_{chem}-Seq to accurately assess RNA Polymerase activity following DSB induction. This convincingly indicated that transcriptional elongation is decreased at damaged genes post DSB induction (Fig. 2g; Ext. Data Fig. 2f-g). To directly measure if this elongation defect was due to reduced RNAPII velocity in the body of these

damaged genes, we further performed **DRB/TT_{chem}-seq** (as developed by the lab of J. Svejstrup PMID: 31915390). Our data show that while transcription elongation is reduced (Fig. 2g), the RNAPII velocity on damaged gene bodies does not seem to be affected (Fig. 5c).

We also obtained further insights into the mechanism allowing DSB-induced transcriptional repression. We found that this reduction in elongation is linked to the **eviction of SPT5** (Fig. 2i-j) and the **recruitment of PAF1** (Fig. 5e-f; Ext Data Fig. 5i). Of interest, these two factors have been previously reported to regulate RNAPII PPP release. Indeed, while acute degradation of SPT5 triggers defective release of RNAPII from the PPP (PMID: 34480849, PMID: 34534457), acute degradation of PAF1 triggers the opposite effect, *i.e* the unscheduled escape from promoter-proximal pausing (PMID:35973425, PMID: 28860207; PMID: 35363521). Interestingly, RNAPII promoter-proximal pausing was recently suggested to function as a key quality control checkpoint for transcriptional integrity (PMID: 39504960).

Thus, altogether, the new data included in the revised manuscript **provide evidence that DSB-induced transcriptional repression occurs largely by interfering with the regulatory step taking place at promoter-proximal pausing rather than through a decrease in RNAPII speed** and suggest that TC-DSBs induce the activation of this newly described transcriptional quality checkpoint.

3) Figure 3i. The authors use depletion of SETX as one of the major R-loop helicases to accumulate RNA:DNA hybrid levels and observe increased resection by END-Seq. The authors should confirm that SETX was indeed depleted and RNA:DNA hybrids accumulated under the used conditions to support this conclusion.

We thank the referee for this comment. We have now included a western blot (Ext Data. Fig. 3l) showing the depletion of SETX using siRNA in DivA cells. We also show by qDRIP-seq that SETX depletion triggers an increase of DSB-induced RNA:DNA hybrids (Ext Data Fig. 3m).

4) Figure 4b. The authors only perform statistical tests on the difference between -DSB and +DSBs on R-loops formed on the template strand. However, a small (but maybe also significant) increase is observed on R-loops formed on the non-template strand. The authors should at least discuss this in the manuscript.

We thank the referee for this important comment. Indeed, we can detect an increase of RNA:DNA hybrids on “non-template” strand of some annotated genes (currently Fig. 4c which replaces Fig. 4B in the original version). However, closer inspection at individual locations showed that, in several instances, RNA:DNA hybrid induced on this apparently “non-template strand” also involves pre-existing RNAs, mainly corresponding to PROMoter uPstream Transcripts (PROMPTs, which can occur in both sense and antisense directions with respect to the downstream gene PMID: 19056938) rather than pre-mRNA. This is now clearly mentioned this in the manuscript (line 288-290; 308-309).

Nevertheless, careful inspection of each of the 80 AsiSI-induced DSBs, allowed us to identify 4 DSBs for which the mechanism of R-loop/RNA:DNA hybrids accumulation remains unclear (Fig. 3 for reviewers). This includes DSB 742, which seems to accumulate an R-loop rather than a RNA:DNA hybrid, and DSBs 961, 964 and 199, which may have been mis-classified as

silent DSBs (see below). It is important to note that, even at these rare individual cases, R-loop accumulation was also asymmetrical and neither RNAPII/III recruitment nor nascent transcription by TT_{chem}-seq could be observed. It is thus unlikely that *de novo* transcription from DSBs account for R-loop formation at these DSBs.

Fig. 3 for reviewer

Fig. 3:
(a) Genomic tracks showing signals for RNAPII, RNAPIII, RNA-seq, TT_{chem}-seq, and qDRIP-seq at 2 TC-DSBs. Left : TC-DSB 473 represents a typical example of a DSB-induced RNA:DNA hybrid formed 1-with respect to the polarity of the pre-mRNA underlying genes (BCKDHA gene, transcribed from left to right) and 2- is stabilized on the non-resected strand. Right : TC-DSB 742 shows a DSB-induced increase in qDRIP-seq signal on the Crick strand upstream of the DSB. Thus, it is an R-loop, rather than an RNA:DNA hybrid that accumulate at this DSB given that the DNA moiety is the strand lost upon resection.
(b) Left: Heatmap representing qDRIP-seq signal at TC-DSBs and silent DSBs. 3 DSBs classified as silent (DSB961, DSB964 and DSB199) display increased signals following DSB induction. Right: Genomic tracks showing signals for RNAPII, RNAPIII, RNA-seq, TT_{chem}-seq, and qDRIP-seq at DSB961, DSB964 and DSB199. DSB 961 is in fact lowly transcribed, which explains the presence of RNA:DNA hybrid at this "silent" DSB. DSB 964 is located in an enhancer (our own unpublished data in U2OS, and Encode annotation) and DSB199 in a regulatory element (ReMap database, PMID: 29126285). The formation of DSB-induced R-loops/RNA:DNA hybrids at these specific regions of the genome may possibly entail a different mechanism. Yet as for all other DSBs, RNA:DNA hybrids/R-loops accumulate assymetrically and no RNAPII/III recruitment is detected.

5) Figure 5d/e. The authors analyze cDNA levels at different DSB loci after SPIN1 knockdown or MS31 treatment, a specific inhibitor of SPIN1/H3K4me3 interaction. To what extent does SPIN1 knockdown or MS31 treatment on its own (under -DSB condition) affect cDNA levels of the investigated loci? This information is lost due to the double-normalization. Additionally, the authors should provide evidence that MS31 treatment is indeed blocking the interaction with H3K4me3 as expected, for example by ChIP-qPCR of SPIN1 with and without MS31 treatment at TSS sites.

We thank the referee for this comment. The effect of MS31, SPIN1 and PAF1 depletion before DSB induction is shown below (Fig. 4 for reviewer). For space constraints, these data have not been included in the revised manuscript, but this can be done if necessary.

As for the control of MS31 treatment, we show by ChIP-qPCR that MS31 treatment blocked the recruitment of SPIN1 at TSS, while leaving H3K4me3 unaffected (Ext. Data. Fig.5o).

Fig. 4 for reviewers

Fig. 4: Effect of MS31 treatment and SPIN1, PAF1 and SKI8 siRNA on gene expression previous DSB induction by 4OHT
a. Effect of PAF1 or SKI8 depletion in the absence of damage compared to siCtrl (all normalized to TBP)
b. Effect of SPIN1 depletion in the absence of damage compared to siCtrl (all normalized to RPLP0)
c. Effect of MS31 treatment in the absence of damage compared to non treated (all normalized to RPLP0)

6) Figure 6h. The model postulates that SETX is the responsible R-loop helicase to remove the hybrids that would otherwise become toxic. However, no strong evidence is provided that SETX would be the major player here, considering the fact that a plethora of other DEAD-box RNA helicases have now been described to show R-loop helicase activities and in vivo functions. The authors should adjust the model and discuss this in the manuscript.

In the revised version, we now show that SETX depletion does have a consequent impact on DSB-induced RNA:DNA hybrids (Ext Data Fig. 3m). We also now state in the caption of Fig. 6f presenting the model that SETX may act together with other RNA:DNA helicases to remove hybrids and allow HR repair.

Minor comments:

1) Extended Data Figure 5l. The presented ChIP is done from n=1 biological replicates. It would be good to provide additional biological replicates to strengthen their conclusions.

We have now removed Extended data Fig. 5l, given the amount of new data which have been added to the revised manuscript. However, the figure with biological replicates can be found below (Fig. 5 for reviewers)

Fig. 5 for reviewers

Fig. 5: Effect of MS31 treatment on γ H2AX level
ChIP against γ H2AX was performed in DivA cells upon the indicated treatment in three, biologically independent experiments.

Reviewer #2:

Remarks to the Author:

The authors investigated the mechanisms by how DNA:RNA hybrids are generated in the vicinity of DNA double-strand breaks (DSBs). Multiple previous studies have reported that DSBs induce the formation of an R-loop composed of ssDNA and DNA:RNA hybrids. To date, there are two distinct models regarding the origin of RNA in the R-loop, i.e., de novo RNA synthesis by RNA pol2 or pol3 which is recruited in response to DSBs, or the use of pre-existing RNA synthesized before DSB induction. Although the existence of DNA:RNA hybrids near DSBs is evident, the origin of the RNA in these hybrids has been debated. In this study, the authors demonstrated no recruitment of RNA polymerases in response to DSB induction using the AsiSI-based assay. In addition, the authors found that RNA:DNA hybrids are generated in correspondence with the direction of transcription, proposing a model in which the RNA in the DNA:RNA hybrid has already been synthesized by ongoing transcription prior to DSB induction. Furthermore, the authors showed that the DNA:RNA hybrid formed on the ssDNA overhang generated by resection and the hybrid potentiated CtIP/MRE11-dependent resection, followed by RPA recruitment. Finally, the authors showed that Spindlin 1 was recruited to DSBs and promoted transcriptional repression, resulting in the formation of DNA:RNA hybrids. Overall, the study is well-designed, and the proposed model is refined and plausible. In addition, although sequence-based assays require sophisticated skills and knowledge, the obtained data must be highly reliable because the authors' group has expertise in the analysis of DSB repair using NGS technology in DivA cells. Nevertheless, because the model proposed by the authors conflicts with the other model from other groups supporting the model of de novo RNA synthesis, the release of this paper will have a significant impact on the field of DNA damage response. Therefore, the word choice for data interpretation must be made cautiously. Overall, I am supportive of the model in this study; however, the authors should address the concerns listed below.

First of all, we would like to thank the referee for their work and thorough understanding of our study, as well as for their comment on the model. We have now added a considerable amount of new data to further strengthen our conclusions and model and have carefully edited the manuscript to avoid overinterpretation.

1. The major concern is that the AsiSI-based assay is likely unable to detect a single molecule at a DSB site and a repair event that occurs over a period of seconds to minutes. The authors stated that the DivA assay could identify the transient binding of NHEJ factors. However, because XRCC4 forms filaments at DSB sites, its detection may be easier than the other NHEJ factors. Can the authors detect Ku70/80 and DNA-PKcs recruitment using ChIP-seq?

We thank the referee for this comment. We have previously reported the recruitment of DNA ligase 4 in DivA cells (PMID: 30270107) and the lab of Tanya Paull was also able to detect DNA-PKcs at AsiSI-induced breaks by ChIP-seq (PMID: 37717054). We show Fig. 6 for reviewer that we are able to consistently detect XRCC4 and Lig4 at both TC-DSBs and silent DSBs, while unable to detect RNAPII at silent DSBs.

Fig. 6: Recruitment of XRCC4, LIG4 and RNAPII at TC-DSB and silent DSBs
ChIP-seq signal of XRCC4 (PMID: 24658350), Lig4 (PMID: 30270107) and total RNAPII (this study) at N=65 TC-DSBs and N=15 silent DSBs.

Ionizing radiation (IR) can generate DSBs in less than 1 s. If the recruitment of RNAP per break is within a minute or less and there is only a single molecule per break, such a transient reaction may not be detected by ChIP-seq. In addition, the disadvantage of the AsiSI-based assay is that it requires ~4 h to introduce DSBs, and the timing of DSB induction must be heterogeneous at each locus and in each cell. Thus, the sensitivity of the assay in terms of the number of molecules and transient events was still lower than that of the live imaging-based assays. I am positive about the model proposed in this study; however, I strongly suggest that the authors tone down this statement throughout the manuscript. If the authors insist on the model with minimal modifications, further control experiments may be required.

We agree with the referee's assessment that the AsiSI DSB induction system combined to ChIP approach may indeed fail to detect some very transient events and discussed this point in the revised manuscript (lines 446-457). We also performed additional work to further strengthen our results (as also mentioned in the response to the following points). We now present data in the revised manuscript showing that RNAPII or RNAPIII recruitment is not observed by ChIP-qPCR at either etoposide- or CRISPR/Cas9-induced breaks (Fig. 1e, 1g, Ext Data Fig. 1k). Thus, our data now argue against *de novo* RNA polymerase recruitment at sites of damage in three very different DSB induction models (each coming with their own advantages and

drawbacks). In agreement, we were unable to detect nascent *de novo* transcription at DSBs using TT_{chem}-seq which measures newly synthesized RNA with a short 15 minutes metabolic labeling pulse (Fig. 2e-f).

Thus, while we do not exclude that RNA Polymerase recruitment may occur in a very transient manner, we **consider it unlikely to account for R-loop formation at DSBs**. It is also important to note that in any case the literature supporting *de novo* RNA polymerase recruitment at DSBs remains sparse and is subject to several important technical limitations. Foundational studies reporting RNAPII recruitment at DSBs relied on ChIP-qPCR at I-SceI- and I-PpoI-induced DSBs (PMID: 31570834 and PMID: 29180822) which thus display similar potential drawbacks to AsiSI-induced DSBs, which were rightly pointed out by this referee (reduced sensitivity compared to live imaging and heterogeneity of DSB induction in the cell population). While the reason behind the discrepancy between these studies and our work remain unclear, it is important to note that very high %input values were reported for these ChIP-qPCR experiments which, in addition, were almost identical for many different antibodies (PMID: 31570834). This is quite unusual in ChIP-qPCR experiments and thus calls for caution. Similarly, ChIP-qPCR experiments showing RNAPIII accrual following DSB induction at AsiSI-induced DSBs (Fig11 in PMID: 33626331) were analyzing AsiSI sites showing no evidence of any cleavage in our hands (see Ext Data Fig.1n of the revised manuscript, please note that cleavage efficiency was not controlled in PMID: 33626331). Moreover, while RNA polymerase(s) recruitment was also reported using microscopy, this was mostly at laser-induced DNA damage which generate a broad array of damage at a very high density, which can be an important limit when investigating very localized events. In addition, most studies used an antibody against RNAPII phosphorylated on Ser5, so the observed accumulation could also be explained by more subtle changes in RNAPII dynamics on damaged DNA.

By contrast, previous work from the Greenberg laboratory failed to detect RNAPII recruitment but rather observed a specific loss of elongating RNAPII (Ser2P) at FokI-induced DSB by microscopy (PMID: 20550933). In agreement with this and our data, a recent article demonstrated unambiguously that DSB induction in active genes by CRISP/Cas9-induced leads to *in-cis* transcriptional repression of damaged genes, occurring within minutes, and a strong decrease of RNAPII (measured by ChIP-seq and ChIP-qPCR) 1hour after DSB induction (PMID: 38954517).

Altogether, we consider our data to be in line with several other reports which used a variety of approaches and DSB induction methods and remain confident that *de novo* RNAPII recruitment is unlikely to be the predominant pathway accounting for RNA:DNA hybrid accumulation at DSBs.

Nevertheless, we modified the text throughout the manuscript, and we hope that the referee will find that our statements on this aspect are sufficiently toned down in the revised manuscript.

2. In this study, all the data were obtained using the DIVA assay. I do not think that the repair process of a restriction enzyme can be generalized because, as the authors described in the Discussion, AsiSI continuously digests DNA until mutations in the recognition sequence for AsiSI are generated. Does CRISPR-Cas9-based digestion show similar results? DSBs generated by CRISPR-Cas9 may fail to form RNA:DNA hybrids because of the presence of gRNA. If possible, the authors should confirm the critical findings at other TC-sites using a CRISPR-Cas9-based assay.

We would like to thank the referee for this comment and insightful advice. As suggested, we performed analysis of RNAPII, RNAPIII and RNA:DNA hybrids accumulation at Cas9-induced DSBs (Fig. 1d). We found that neither RNAPII (Fig. 1e) nor RNAPIII (Fig. 1g) accumulate at DSBs induced by Cas9 in a promoter, an intron or an intergenic region. However, RNA:DNA hybrids did accumulate at Cas9-induced DSBs if induced within a transcribed gene (Fig. 4d) but not in a non-transcribed intergenic locus (Fig. 4d, sgRNA3).

Thus, induction of DSBs using the CRISPR/Cas9 system yield similar results to those obtained at AsiSI-DSBs, *i.e.*, a lack of RNA polymerases recruitment at any type of location and an accumulation of RNA:DNA hybrids at DSBs induced in transcriptionally active loci. These experiments allow to strengthen our previous findings and give more weight to our interpretation regarding the requirement for pre-existing RNA molecules in the formation of DSB-induced RNA:DNA hybrids.

3. In Figure 6, the authors used etoposide, which induces DSBs by inhibiting topoisomerase II activity. Because topo II activity is associated with transcription, some DSBs must be TC-DSBs. However, indeed, most DSBs are repaired by NHEJ but not HR (PMID:24316220). Although the authors refer to a paper showing the involvement of SETX in etoposide-induced DSBs, it is unclear how many etoposides induced DSBs are associated with transcription.

As mentioned above, a major concern is that the conclusion of this study is based on the results of the DivA assay. If etoposide is available as a TC-DSB inducer, the authors should confirm the key findings, such as RPA, RAD51, and gH2AX foci, by etoposides. In addition, identification of the BRCA1 foci will be informative in this study.

We thank the referee for this comment. Previous work (PMID: 31202576) showed that etoposide-induced DSB displayed a strong bias toward transcribed regions and mainly promoters (Promoters 55%, Enhancers 16%, Gene bodies 12%, Intergenic regions 17%).

In order to address this point, we performed END-seq to identify etoposide-induced DSB in U2OS cells. We further analyzed RNAPII/III recruitment by ChIP-qPCR at several etoposide-induced DSBs, selecting 3 DSBs induced in intergenic loci and 3 TC-DSBs (Ext Data Fig. 1j). As mentioned above, RNAPII/III recruitment was not observed at any of these etoposide-induced DSBs and RNAPII rather decreased at etoposide-induced TC-DSBs (Ext Data Fig. 1k). These results are similar to those obtained for AsiSI- or CRISPR/Cas9-induced DSBs.

Additionally, we now also show that, as for AsiSI-induced DSB, interfering with DSB-induced transcriptional repression and RNA:DNA hybrid accumulation, for example using the SPIN1 inhibitor MS31, decreases RAD51 and BRCA1 foci assembly following etoposide treatment (Ext Data Fig. 6a).

4. In Figure 3, the authors concluded that DSB-induced RNA:DNA hybrids form on the ssDNA overhang generated by end resection (as stated in the Discussion section). However, this process is not included in the model in Figure 6h. This should be clarified. As an alternative possibility, does failure to initiate resection promote hybrid degradation? The authors may not need to perform additional experiments to answer this question; however, the proposed mechanisms should be carefully spelled out in a revised manuscript.

We have tried to clarify the model which is now presented Fig. 6f. We propose that once the DSB is induced, there is a dynamic equilibrium between the canonical dsDNA structure and the unstable R-loop (represented by the double arrow on the model). This unstable R-loop will displace the DNA strand that is not hybridized with the RNA and may therefore further favor the action of endonucleases which would initiate resection by removing the flap (as suggested by enhanced RPA binding on the side of the R-loop in Fig. 3h-i). This cleavage will lead to the stabilization of the resulting RNA:DNA hybrids (since the “competing” second DNA strand has been removed). This is in agreement with the fact that MRN inhibition and CtIP depletion decrease RNA:DNA hybrids at DSBs (Fig. 3e). As suggested by the referee, failure to initiate resection may provide more opportunities to resolve the unstable R-loop at DSBs but whether R-loops or RNA:DNA hybrids are preferred substrates for the resolution machinery remains an open question which we decided not to address in an already complex model. We made our best to improve the discussion related to this model, and hope this referee finds this new version clearer.

5. In Figure 3, the authors used mirin, which is an inhibitor of MRE11 3'-5' exonuclease activity. Since the direction of the exonuclease is opposite (resection undergoes 5'-3' direction), it is unclear how MRE11 3'-5' exonuclease is involved in this process. The authors should clarify the requirement of MRE11 endonuclease activity using an inhibitor, PFM01, or MRE11 mutant.

Following the appropriate suggestion of this referee, we measured DSB-induced R-loop by DRIP-qPCR after PFM01 and mirin treatment. Both reduced RNA:DNA hybrids levels at DSBs (Fig. 3e) strengthening the link between resection and RNA:DNA hybrids stability.

6. The authors' group previously uncovered transcription-coupled DSB repair pathway choices using the DivA assay. In this study, the authors probably used TC-DSB sites that do not recruit NHEJ factors. In contrast, several recent studies have shown that NHEJ factors bind to the DSB ends before resection (PMID:31934630; PMID:36917982). The authors almost ignored the fact that NHEJ factors are present in DSBs undergoing HR. Or NHEJ is not recruited at TC-DSB sites at any time points? The authors should mention the interaction between NHEJ factors and the TC-DSB/hybrid. Importantly, if the DivA assay fails to detect NHEJ factors at TC-DSB sites, RNAP may be also undetectable?

We apologize for the lack of clarity in the original manuscript which made the reviewer feel we were ignoring that NHEJ factors can be recruited before resection takes place.

Actually, we previously showed that NHEJ proteins (such as XRCC4 and Lig4) are recruited at **all** AsiSI-induced DSBs, *i.e.*: at TC-DSBs and silent DSBs (PMID: 24658350; PMID: 30270107, and see also the above Fig. 6 for referee in response to point #1). Thus, the TC-DSBs analyzed here do recruit NHEJ factors. Yet, while all DSBs can recruit NHEJ factors, only the subclass of DSB falling in transcribed regions (TC-DSBs) can also recruit HR proteins (*via* mechanisms previously reported by our lab and by other labs). The presence of both NHEJ and HR factors at these breaks was discussed previously (for instance in the results section and/or discussion of PMID: 24658350; PMID: 30270107; PMID: 26586426, or in reviews, such as PMID: 28363678). The data presented here now suggest that nearby RNAPII and preexisting RNA molecule could be an additional mechanism by which TC-DSB are channeled to HR

repair (by favoring resection initiation). However, we did not intend to imply that RNA:DNA hybrids do not play any role in NHEJ. We have now amended the manuscript to cite studies suggested by the referee as well as others regarding the interplay between HR and NHEJ factors at DSBs (PMID: 37777505; PMID: 36794853; PMID:31934630; PMID:36917982; PMID: 36868227). More specifically, we mention in the discussion that RNA:DNA hybrids may recruit Ku70/80 which could contribute in MRN recruitment and nickase activity (line 516-517).

7. As shown in Figure 6h, SPIN1 is located on the left side but not on the right side of the DSB. However, as shown in Figure 5 and Extended data Figure 5d, SPIN1 is recruited to both sides of the DSBs. The authors should clarify this point further.

Given that we focus a lot more on the aspects of transcriptional repression in the new version of this manuscript, we have removed the figures showing SPIN1 recruitment at DSB, to rather focus on its distribution at damaged genes (which is similar as most DSBs induced in D_{IV}A are located in the vicinity of the TSS of genes). We have now modified the model accordingly.

8. In Extended data Figure 5l-m, the authors show a decreased γ H2AX signal in MS31-treated cells. Is this result simply explained by the reduction in the number of DSBs induced by AsiSI? The change of chromatin structure may affect the efficiency of DNA cleavage by restriction enzymes. If the number of DSB induction is not affected, ATM activity is reduced in SPIN1 depleted or MS31-treated cells. Therefore, ATM activity should be confirmed, for example by monitoring ATM autophosphorylation after IR.

Given the change of focus in the revised version of the manuscript, we have removed this figure. However, our analyses indicates that ATM autophosphorylation induced by DSBs is not reduced following MS31 treatment (Fig. 7 for reviewers). This actually suggests that DSB induction is similar in MS31 treated and untreated D_{IV}A cells. The reason for decreased γ H2AX is yet unclear, and would deserve further investigation.

However, of importance, we now also show in Fig. 6a that MS31 reduces RAD51 and BRCA1 foci at **etoposide-induced DSBs**, indicating that the effect of MS31 is not limited to AsiSI-induced DSBs.

Fig. 7 for reviewers

Fig. 7: MS31 treatment does not reduce ATM autophosphorylation.

9. As shown in Figure 3, RPA recruitment (i.e., resection) was observed on both sides. In contrast, hybrids were observed on the click strand on the right side. How was resection initiated on the left side? Additional experiments are not required; however, the left-side DNA in the model figure of Figure 6 should be amended for clarity because the DNA on both sides must be resected. The current version is confusing.

We agree with this reviewer that the model in the original manuscript was confusing and we apologize for this. Indeed, we did not intend to imply that resection takes place only on one side. Bidirectional resection is now clearly pictured on the model and we describe “R-loop-induced DNA end resection” as an alternative resection initiation pathway.

10. In the manuscript, the authors use the term “short-range resection.” This term should be defined in earlier of the manuscript. If MRE11 endonuclease activity is required for this process, “initiation of resection” may be a better choice of words.

Thank you for this comment. Indeed, MRE11 endonuclease activity is required, since PFM01 treatment triggered a decrease in RNA:DNA hybrids (Fig. 3e). We now mostly refer to initiation of resection, rather than short-range resection.

Reviewer #3:

Remarks to the Author:

In the manuscript entitled: “transcriptional repression account for RNA:DNA hybrid accumulation at DNA double strand breaks” Lesage and co-authors have performed a omics-based tour de force to answer one of the big debates in the TC-DSB field, whether Pol II is recruited at DSBs and what the cellular function to the DNA damage response could be. In this study the authors were unable to identify recruitment of Pol II at DSB sites, both in transcribed or non-transcribed loci. Interestingly, the authors revealed the presence of RNA:DNA hybrids at DSBs only in transcribed genes. These RNA:DNA hybrids arise in a non-symmetrical manner, suggesting that these are caused by the hybridization of nascent RNA with the 3’overhang upon resection. Finally the authors identify that Spindlin 1 promotes RNA:DNA hybrids at DSBs, as a consequence of ATM mediated transcriptional shut-down at DSBs in transcribed loci.

This study addresses the issue whether Pol II is actively recruited at DSB-sites in a convincingly manner, which is an important issue for the field, together with the additional insights provided this manuscript is a strong candidate for publication in NCB, given that the concerns indicated below are addressed in a convincingly manner.

We would like to thank this referee for their comments on our manuscript and their thorough work and understanding of our study.

Major comments:

- Figure 1B shows that in this cellular system used to elegantly induce DSBs, on the majority of DSB sites an accumulation of Pol II is observed. This is in it self a striking observation, this could for example be caused by the fact that these DSB are predominately localized near transcriptional pause sites, or that these DSB site have a unique chromatin structure resulting in more Pol II. This high Pol II abundancy could explain why no Pol II accumulation could be observed upon DSB induction. As one of the strong points of this manuscript is to show that no

Pol II is accumulated at DSBs, the authors should show that their model is also relevant for all types of DSBs and use alternative methods to induce DSBs and confirm that main results.

We thank the referee for this comment. Indeed, most of cleaved AsiSI sites are located in promoters or in the 5' end of genes (88%, PMID: 30270107, see also PMID: 28561034), which are indeed regions prone to accumulate initiating and/or paused RNAPII, as rightly pointed by this referee. Note that this specific localization of TC-DSBs in 5' and promoters is actually a very good model given that topoisomerase II-induced DSBs also mostly lie within 5' and promoters (PMID: 31202576). We have now clarified the position of each AsiSI-induced DSB relative to genomic annotations and transcriptional status of the corresponding locus before damage (Sup Table 1 and Ext Data Fig. 1a, see also reply to referee # 1 point 1).

We now present new data directly comparing **TC-DSBs** (*i.e.* located mostly in promoters/5' end of genes and enriched for RNA PII before damage, N=65) and **silent DSBs** (*i.e.* DSB induced in either intergenic loci or silent genes, N=15). We show that we are **unable to detect a recruitment of RNAPII or RNAPIII at either TC-DSBs or silent DSBs** (*i.e.* regardless of RNA polymerase levels before DSB induction). For full legibility, we have now included heatmaps (Ext Data Fig. 1e, Ext Data Fig. 2c) and box plots (Fig. 1c, Fig. 2c) as well as several examples (Fig. 1b, Fig. 2a). Furthermore, and as suggested by the referee, we now also present additional ChIP-qPCR data **showing that RNAPII and RNAPIII cannot be detected at CRISPR/Cas9- and etoposide-induced DSBs induced in active or inactive loci** (Fig. 1d-e; Fig. 1g; Ext Data Fig. 1j-k) (see also response to reviewer 2, second part of point #1). These additional data and experiments allowed to **generalize our findings of a lack of RNA polymerase recruitment at different types of DSBs** and greatly improved the manuscript.

- How efficient is the AsiSI mediated DSB induction, in what percentage of cells is a break induced, is at a high percentage of AsiSI sites in every cell a break induced? This is crucial info, as if this is only in a low percentage of cells, this might explain why not a Pol II recruitment by ChIP could be detected.

We thank the referee for this comment. While the efficiency of cleavage can vary between individual DSBs (see for example Fig S1B in PMID:30270107), we now show that DSB 526 is cleaved in about 25% of cells (using ddPCR with primers across the DSB to measure DNA cleavage, Ext Data Fig. 3h). We consider that this should not be drastically different to studies describing RNAPII recruitment at I-SceI and I-Ppo-induced DSBs (PMID: 31570834 and PMID: 29180822) and should therefore be sufficient to detect RNAPII recruitment, even if transient. As a comparison, we can easily detect XRCC4 and Lig4 transient binding in similar conditions (PMID: 30270107 and Fig. 5 for reviewer in response to Point1 of Referee 2). Moreover, we also did **not detect RNAPII recruitment at AsiSI-induced DSBs by ChIP-qPCR** which is often considered more sensitive. In addition, and as already mentioned above, we now provide new data showing **no recruitment of RNAPII/III at CRISPR/Cas9 and etoposide-induced DSBs located in both active or inactive loci**. Together with our **TT_{chem}-seq data showing no nascent transcription at DSBs** and our observations strongly suggesting that the formation of the vast majority of **DSB-induced RNA:DNA hybrids is asymmetric and strongly dependent on the presence of pre-existing RNA molecules and resection**, we are fairly confident in our conclusion that, even if *de novo* RNAPII recruitment occurs very transiently (and thus falls below our detection threshold), it is not a prevalent mechanism

accounting for RNA:DNA hybrids accumulation as previously proposed (PMID: 30560944). We modified the discussion accordingly and hope that the referee will find it appropriate.

- The authors suggest that DSB's result in a decreased elongation speed, which results in the accumulation of RNA:DNA hybrids at break sites, however this is mainly based on correlation (e.g. ATM inhibition). The authors should show that indeed the decreased elongation speed is the cause for RNA:DNA hybrids, for example using slowly elongating RNA Pol II mutants.

We would like to thank the referee for this comment which prompted us to conduct more experiments which we believe to have considerably improved the manuscript.

To answer this, we first attempted experiments using RNAPII mutants that were reported to display slow or fast processivity (PMID: 25452276) as suggested by this referee. Unfortunately, such experiments were really difficult to interpret due to a high variability in RNAPII expression from one experiment to the next and the difficulty to calibrate the relative expression levels between RNAPII mutants (Fig. 8. for reviewers see WT, FAST, SLOW RNAPII)

Fig. 8 for Reviewers

Fig. 8.
a. Examples of 3 independent experiments of DRIP-qPCR after transfection of Rbp1-WT or Rbp1 mutants (fast or slow elongating). This shows the variability between the experiments.
b. Examples of Western blots of 4 independent experiments showing the variability in the levels between each transfected plasmids (WT, FAST or SLOW).

Protocol
 Transfection of DiVA cells with plasmids encoding Rbp1-WT, Rbp1-fast or -slow mutants. After 24h, treat with alpha amanitin for 24h. Then, add OHT to induce DSB for 4h. Cells were collected for DRIP-qPCR or Western blot analyses.
 Note that there was always a lot of cell death due to the transfection of the plasmids despite having tested different plasmid concentrations.

To investigate RNAPII elongation and velocity, we thus turned to TT_{chem}-seq and DRB/TT_{chem}-seq (PMID: 31915390). TT_{chem} seq revealed that elongation is reduced at genes that are damaged by AsiSI (Fig. 2e-h). Yet, DRB/TT_{chem}-seq, which allows to measure the progression of RNAPII (Ext Data Fig. 5d-e), did not indicate an alteration of RNAPII velocity across damaged genes (Fig. 5c).

Thus, we set to better understand the mechanisms that trigger this DSB-induced reduction in transcriptional elongation without a detectable decrease in RNAPII velocity on damaged gene bodies. We therefore investigated the contribution of various candidate factors involved in

regulating the RNAPII transcription cycle and found that while SPT5 is evicted from damaged genes, PAF1 is recruited (see also reply to Referee # 1 point 2). Interestingly, both proteins were previously reported to be involved in the control of promoter-proximal pausing release. On one side, SPT5 allows the RNAPII to escape from the PPP, and its acute degradation triggers the degradation of arrested RNAPII (PMID:34534457, PMID:34480849). On another side PAF1 acute degradation triggers the aberrant release of RNAPII into the gene body (PMID:35973425, PMID:35363521, PMID: 28860207).

Altogether, the new data included in the revised manuscript indicate that DSB-induced transcriptional repression occurs largely by interfering with the regulatory step that takes place during normal RNAPII promoter-proximal pausing via the eviction of SPT5 and the aberrant recruitment of PAF1 rather than through a decrease in RNAPII speed. We have refocused the manuscript on these aspects and hope the referee will find our data sufficiently convincing.

- Additionally, if Pol II elongation rate are severely downregulated, as suggested by the authors, further downstream of the break site a decrease in Ser2 modified Pol II is expected, the genes will not elongate that fast through the gene body. However, this decreased Ser2 ChIP signal at the end of genes upon DSB induction is not detected. The slower transcription rate, or increased termination rate should be more convincingly shown.

As mentioned in response of referee 1, we have now considerably changed this section. In the revised version of the manuscript, we rather show the profile of RNAPII at DSBs rather than on damaged genes (Fig. 2) to emphasize the fact that neither total nor modified RNAPII increase at DSBs. We also convincingly show a decrease in elongation at damaged genes by measuring nascent transcripts by TT_{chem-seq} (Fig. 2g; Ext. Data Fig. 2f-g). In agreement with this defect in elongation on damaged genes observed by TT_{chem-seq}, RNAPIIS2P does slightly decrease on damaged gene bodies compared to control genes (Fig. 1 for referees in response to referee # 1, point 2). As mentioned above, this figure has been removed from the revised version but could be included if the referee finds this to be necessary.

- The authors propose a model in which the nascent RNA of a Pol II molecule that has passed the AsiSI site before DSB induction will hybridize back to the resected DNA. This is an interesting model. However, I wonder how frequent this event would be. When we assume that splicing happens co-transcriptional in an efficient manner, intron will be removed rapidly. Therefore, only nascent RNA that has just been synthesized, and thus with Pol II very closely located to the break site, could be the source of the RNA:DNA hybrid. Since not every gene is actively transcribed at a given moment in a single cell, and that genes are relatively long with maybe only 1 or a few Pol II molecules engaged, the change that the nascent RNA hybridized with the resected DNA seems very small.

We thank the referee you for appropriately raising this point.

There are several reasons which may explain why we detect hybridization of the newly transcribed RNA molecule more frequently/efficiently that would be otherwise anticipated.

1-In the DivA system or upon etoposide treatment, most of the TC-DSBs are induced in promoters and 5' end of the genes (see above point 1, as well as reply to referee # 2 point 3 and

Supp Table 1). At these regions, RNAPII is either initiating, paused or progressing slowly, which likely gives more opportunities to the pre-mRNA or PROMPT to form R-loop and RNA:DNA hybrids. With the new data included in the revised manuscript, we now favor a model in which DSB induction triggers an exacerbated promoter-proximal pausing, further increasing this likelihood for pre-existing RNA molecules (pre mRNA or PROMPT), to hybridize back to the dsDNA at the vicinity of the break. This was not part of the original manuscript but we did our best to make it clearer in the revised manuscript (Ext Data Fig. 1, Fig.6f and its caption, and discussion section).

2-It is also important to note that R-loop accumulation is not observed at all TC-DSBs (only about 40% of DSBs, see Extended. Data Fig. 3e), suggesting that some loci are more predisposed to the formation of such structures following DSB induction. This could indeed be explained by local chromatin environment, the presence of secondary structures and DNA sequence motifs or, as suggested by the referee, by differences in the average density of transcribing RNAPII or the lifetime/stability of the R-loop forming RNA moiety (intron vs exon). Yet, we could not find any strong differences in R-loops/hybrids formation at DSB induced in intron versus exon, either when analyzing our DRIP-seq data in DlvA cells (Ext Data Fig. 4b) or using CRISPR/Cas9-induced DSB (Fig. 4e). However, we noted that R-loop/RNA:DNA hybrids formation following DSB induction seems correlated with the amount of R-loop detected before DSB induction (Fig. 9 for reviewers). This indicates that somehow these loci display the propensity to accumulate R-loop before any damage (which could be due to decreased RNAPII local velocity, increased pausing, or specific sequences at these loci) which is exacerbated following DSB induction.

3- We also consider the possibility that in cases when RNAPII and preexisting RNA are not present very closely to the break site (for example between 2 rounds of transcription), this break may be quickly repaired by NHEJ (as we previously described for breaks in inactive loci PMID: 24658350, PMID: 30270107). This fast NHEJ repair would allow another round of DSB induction which could now occur with nearby RNAPII and RNA, providing a new opportunity for R-loop and RNA:DNA hybrid formation. The resulting R-loop containing structure would be more complex to repair and thus persist for longer, making its detection easier. This discussion was not included in the manuscript due to space constraints but this can be done if this referee considers it to be necessary.

Thus, while further studies are required to fully understand why some loci are more prone to accumulate R-loops following TC-DSB than others, the above data indicate that hybrids positive TC-DSBs are likely sites that exhibit particular RNAPII behavior, accounting for the elevated R-loop level before damage. On the whole, 1) our new data about the fact that DSB-induced transcriptional repression likely involves the regulatory step that takes place at PPP, 2) the fact that DSB-induced R-loops/hybrids accumulate downstream of repression and 3) our clarification of the model (Fig.6f and its caption) will hopefully make the manuscript clearer in that respect.

Fig. 9 for reviewers

o Would this suggest that Pol II is loaded on the DSB, explaining the RNA:DNA hybrids formed ?

As mentioned in the foreword before the point-by-point response to each referee, and in response to referee #2 point 1, we are quite confident regarding the absence of *de novo* RNA Pol recruitment which we report here and which is in line with other reports (for example PMID: 20550933 and PMID: 38954517). Even if more work will be needed to fully disentangle the process(es) leading to R-loops and RNA:DNA hybrids formation at TC-DSBs, our data indicate that *de novo* RNAPII loading and transcription unlikely constitute a prevalent mechanism for their formation in most contexts and would at best be a mechanism occurring very rarely.

o Are more hybrids detected when the DSB is generated in an exon vs an intron, exons will not be co-transcriptionally removed and thus a higher chance of hybridizing.

As mentioned above, we could not find any strong differences in R-loop/hybrids formation at DSB induced in intron versus exon, either when analyzing our DRIP-seq data in DivA cells (Ext Data Fig. 4b) or using CRISPR/Cas9-induced DSB (Fig. 4e).

o Is there a increased DRIP-seq signal if transcription initiation is inhibited shortly after DSB induction, in this scenario Pol II will be near the end of genes and

Unfortunately, we were unable to perform the suggested experiment by inhibiting transcription shortly after DSB induction. In the DivA system, this would lead to a reduction of the AsiSI enzyme levels and thus a decrease in DSB induction during the 4h of OHT treatment, which will make this experiment difficult to interpret. This experiment is also difficult to envisage with CRISPR/Cas9-induced DSBs for which it is not as easy to control the timing of induction and thus the kinetic of R-loops and RNA:DNA hybrids formation. Of note, we previously reported that transcription inhibition prior to DSB induction leads to a decrease in HR repair

and a switch to NHEJ (PMID: 24658350), which is in line with the model described suggesting a link between preexisting RNA, R-loops and RNA:DNA hybrids formation and resection. While we apologize for not being able to propose a good alternative for this experiment, we sincerely hope that the new data and analysis provided in the revised manuscript will be convincing enough for this referee.

- Along this line, how frequent is the presence of the RNA:DNA hybrids at break sites? Does this happen at every break, or only in a very minor fraction. This is an important question to address as this will provide info on how relevant this phenomenon is. The RPA ChIP data also suggests that only a minor fraction of DSBs has hybrid, the RPA ChIP signal seems very at the exact same genomic regions as where the hybrids are detected, while it is expected that RPA would bind the resected DNA, but not the RNA:DNA hybrids.

To answer the point related to the prevalence of DSB-induced RNA:DNA hybrids accumulation, we compared our qDRIP-seq data with data generated from the lab of F. Chedin estimating the occurrence of R-loop at several loci using bisulfite sequencing (PMID: 32681515). By comparing the qDRIP-seq or DRIP-qPCR signals at such previously characterized loci (*RPL13A*, *RPS24*) with the level of hybrids that accumulate at a given TC-DSB (DSB 526), together with our ddPCR assay estimating that this site is cleaved in about 25% of cells, we concluded that DSB-induced RNA:DNA hybrids accumulate on more than 50% of the cleaved molecules. This is now mentioned in the manuscript (Extended Data Fig. 3f-h; lines 217-225). We sincerely hope that this referee will find this data convincing.

Regarding RPA recruitment, we would like to clarify that indeed we found that RPA ChIP-seq signal does extend over several kbs in a strand-specific manner and symmetrically on both side of the DSB (similar to PMID: 33651987). We apologize if this was not clear in the original manuscript and thank the referee for pointing this out. We now show heatmaps for all DSBs after both 4h and 24h of DSB induction (Ext Data. Fig3, j right panel). However, closer to the DSB (~ 200bp) we can see a bias of RPA accumulation on the side of the break which accumulates the hybrid. We interpreted this as the possibility that hybrids contribute to an alternative mode of resection initiation which would generate more frequent ssDNA at the site of hybrid and thus more RPA binding to ssDNA (see model Fig. 6f). In agreement, previous studies have suggested that XPG, CtIP and MRE11 may be involved in a R-loop dependent, alternative mechanisms of resection (PMID: 32375052; PMID: 35108530; PMID: 30245011). It is likely that such asymmetric non-canonical initiation of resection would commit this TC-DSB to HR repair, leading to resection on the other end (where no hybrids is formed) and subsequent symmetric and bidirectional accumulation of RPA.

- The authors state the RNA:DNA hybrids are formed in cis, however what is the proof that the RNA:DNA hybrids are not formed by RNA transcribed by the other allele? To rule out such a scenario experiments in haploid conditions should be executed.

We thank the referee for this comment. To answer this question, we performed DRIP qPCR after induction of a Cas9 DSB in the HAP1 haploid cell line. Our data indicate that in this condition, RNA:DNA hybrids are still able to accumulate after DSB induction (Ext Data Fig. 4f), arguing against hybridization of an RNA molecule produced in *trans*.

Minor comments:

- Fig. 4C, D and 6B should be shown for other genes as well, or DRIP-seq should be performed.
- Fig. Ext. 4C, DRIP-PCR should be shown for these genes as well

We thank the referee for this comment. Unfortunately, we did not manage to establish a proper DRIP-qPCR assay allowing to detect reproducible signals at these sites. However, DRIP-seq was performed in the condition of SPIN1 depletion.

- Is the DSB induction affected by the treatments or knockdowns used, e.g. ATM inhibition, MS31 or NELFE depletion?

We thank the referee for this comment. Our previous work (PMID: 26586426) indicated that ATM inhibition does not alter DSB induction by AsiSI. We now report that MS31 treatment displayed the same effects on etoposide- and AsiSI-induced DSBs (*i.e.* decreased RAD51 and BRCA1 foci formation Extended Data Fig. 6a, increased cell survival Extended Data Fig. 6c) indicating that the effect observed is not due to changes in DSB induction by AsiSI. We have not analyzed DSB induction upon NELF-E depletion. As immunofluorescence is more indicative of DSB signaling than DSB induction efficiency *per se*, this would require time consuming assays such as END-seq or our previously described cleavage assay (PMID: 20360682; PMID: 24658350). While we apologize for not having performed this experiment, we sincerely hope that the new data and analysis provided in the revised manuscript will be convincing enough for this referee. Alternatively, and given the new orientation of the revised manuscript, we could also remove the NELF-E data.